# ✳ BIOCODER: A BENCHMARK FOR BIOINFORMATICS CODE GENERATION WITH CONTEXTUAL PRAGMATIC KNOWLEDGE

## ABSTRACT

Our new abstract is as follows: Pre-trained large language models have significantly improved code generation. As these models scale up, there is an increasing need for the output to handle more intricate tasks and to be appropriately specialized to particular domains. Here, we target bioinformatics due to the amount of specialized domain knowledge, algorithms, and data operations this discipline requires. We present BIOCODER, a benchmark developed to evaluate large language models (LLMs) in generating bioinformatics-specific code. BIOCODER spans a broad spectrum of the field and covers cross-file dependencies, class declarations, and global variables. It incorporates 1026 Python functions and 1243 Java methods extracted from GitHub, along with 253 examples from the Rosalind Project, all pertaining to bioinformatics. Using topic modeling we show that overall coverage of the included code is representative of the full spectrum of bioinformatics calculations. BIOCODER incorporates a fuzz-testing framework for evaluation. We have applied it to evaluate many models including InCoder, CodeGen, CodeGen2, SantaCoder, StarCoder, StarCoder+, InstructCodeT5+, GPT-3.5, and GPT-4. Furthermore, we finetuned StarCoder, demonstrating how our dataset can effectively enhance the performance of LLMs on our benchmark (by >15% in terms of Pass@K in certain prompt configurations and always >3%). The results highlight two key aspects of successful models: (1) Successful models accommodate a long prompt (> 2600 tokens) with full context, for functional dependencies. (2) They contain specific domain knowledge of bioinformatics, beyond just general coding knowledge. This is evident from the performance gain of GPT-3.5/4 compared to the smaller models on the benchmark (50% vs up to 25%).

## 1 INTRODUCTION

Large language models (LLMs) have shown great success in code generation (Chen et al., 2021; Chowdhery et al., 2022; Chen et al., 2023; Barke et al., 2023; Li et al., 2023). The landscape of existing coding benchmarks for large language models is largely populated with simple functions, often limited to a handful of lines (Chen et al., 2021; Austin et al., 2021b; Du et al., 2023; Wong et al., 2023). Combined with a significant lack of closed-domain datasets across diverse fields, this landscape highlights the need for a more robust benchmarking system. Although domain-specific datasets, such as DS1000 (Lai et al., 2022) for data science, have emerged, they fall short of adequately addressing specific tasks in fields like bioinformatics. Open-domain alternatives, including

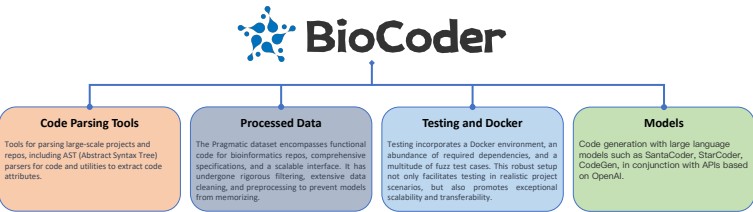

Figure 1: Overview of our contribution in BIOCODER.

HumanEval (Chen et al., 2021), MBPP (Austin et al., 2021b), and APPS (Hendrycks et al., 2021), offer entry-level programming tasks, but their utility is limited as they lack the ability to test more niche, domain-specific code blocks. This shortfall is largely due to a lack of appropriate fine-tuning and context (Muennighoff et al., 2023b); therefore, a more comprehensive, encompassing approach to benchmarking is clearly needed.

To bridge these gaps, we introduce BIOCODER (see Figure 1), a benchmark for code generation incorporating 2269 bioinformatics-specific coding problems. Our BIOCODER benchmark mainly targets bioinformatics data analysis, which includes tasks such as managing various biological data formats, understanding processing workflows, and utilizing APIs of various packages. This area encapsulates the majority of daily tasks a bioinformatician encounters in data analysis. Note, however, that BIOCODER also touches upon parts of writing bioinformatics software: when tool development intersects with data analysis (see Appendix o for more details with the topic modeling and statistics regarding the overall topic coverage of the dataset). Further expanding the scope of BIOCODER, we included an additional 253 questions from the Rosalind project. This project specializes in generating Python functions addressing key bioinformatics topics such as genetic sequencing and DNA/RNA analysis. BIOCODER assures the inclusion of all potential external packages and code that could be utilized by the generated program. This consideration extends to the recognition that real-world functions often necessitate managing multiple external function calls and global variable usage; hence, we included all potentially required class declarations in the input. Lastly, we performed ablation studies to determine whether the models are strictly memorizing the solutions rather than being proficient at generating code (see Appendix n).

The key highlights of our work can be outlined as follows:

(1) We create a new high-quality dataset for code generation, curated from 1,720 bioinformatics repositories referenced in peer-reviewed bioinformatics articles. We meticulously processed the data, rephrasing more detailed text descriptions, as well as associated comments and specifications, including considerations needed in coding. (2) We provide an extendable parsing tool that can extract all pertinent information associated with the target function in expansive projects. (3) We offer a library for code LLMs, similar to Bui et al. (2023), furnishing a seamless interface for both training and inferencing in code generation tasks. (4) We provide a fuzzer testing tool capable of scaling to handle substantial datasets. Our benchmark results, derived from 1000 iterations, are particularly reliable, indicating the Pass@K rate.

## 2 RELATED WORK

BIOCODER is a code generation benchmark designed for challenging, practical bioinformatics scenarios, offering an extensible testing framework for evaluating the performance of LLMs. We provide a brief overview of the related work in both code generation models and benchmarks.

| Benchmark | Num | Language | Data Statistics | | | | | Scenario |
|---|---|---|---|---|---|---|---|---|
| | | | Test | P.C. | P.L. | C.C. | *C.L.* | |
| HumanEval (2021) | 164 | Python | 7.8 | 450.6 | 13.7 | 180.9 | 6.8 | Code Exercise |
| MBPP (2021a) | 974 | Python | 3.1 | 78.6 | 1.0 | 181.1 | 6.7 | Code Exercise |
| APPS (2021) | 5,000 | Python | 21.0 | 1743.4 | 41.6 | 473.8 | 21.4 | Competitions |
| DS-1000 (2022) | 1,000 | Python | 1.6 | 879.1 | 31.6 | 137.4 | 5.0 | Data Science |
| HumanEval-X (2023b) | 164* | Multi. | 7.8 | 468.4 | 15.5 | 264.6 | 12.1 | Multilingual |
| NumpyEval (2022b) | 101 | Python | 3.5 | 222.9 | 7.0 | 29.9 | 1.1 | Public Library |
| TorchDataEval (2022a) | 50 | Python | 1.1 | 329.0 | 8.6 | 50.7 | 1.3 | Private Library |
| BioCoder (public set) | 460 | Multi. | 1000 | 10465.6 | 243.5 | 706.8 | 26.2 | Bioinformatics |
| BioCoder (hidden set) | 2,269 | Multi. | 1000 | 12296.7 | 298.8 | 919.5 | 26.2 | Bioinformatics |
| BioCoder (similar set) | 460 | Multi. | 1000 | 9885.6 | 240.8 | 767.5 | 26.8 | Bioinformatics |

Table 1: Comparison of the statistics of BIOCODER to previous benchmarks. **Num** is the benchmark size. **Test** refers to the average amount of test cases. **P.C.** and **P.L.** indicate the average number of characters and lines in each prompt respectively, and **C.C.** and **C.L.** indicate the average number of characters and lines in the original code solutions. This table is derived from Zan et al. (2023), please refer to Zan et al. (2023) for a more comprehensive survey.

## 2.1 CODE GENERATION WITH LLMS

LLMs have truly demonstrated astounding performances across various domains (Askell et al., 2021; Bai et al., 2022; Biderman et al., 2023; Bommasani et al., 2022; Gao et al., 2022; Patil et al., 2023; Xu et al., 2023; Qin et al., 2023; Zhang et al., 2023a). And LLMs trained with code data have shown promising results in generating code, exhibiting impressive zero-shot performance on several benchmarks (Zhang et al., 2023b; Olausson et al., 2023; Li et al., 2023; Fried et al., 2023; Wang et al., 2021; Allal et al., 2023). A proven strategy to improve model performance involves increasing both the model parameters and the volume of training data (Radford et al., 2019; Brown et al., 2020; Mitchell et al., 2023), while a lot of large-scale LLMs have been developed (Chowdhery et al., 2022; Thoppilan et al., 2022; Hoffmann et al., 2022). These models have proven their code generation prowess (Brown et al., 2020; Chen et al., 2021; OpenAI, 2023), and the field has also seen the release of several open-source code LLMs, such as bilingual GLM-130B (Zeng et al., 2022), CodeGeeX-13B (Zheng et al., 2023a), OctoPack (Muennighoff et al., 2023a), WizardCoder (Luo et al., 2023), SantaCoder (Allal et al., 2023), and StarCoder (Li et al., 2023). Salesforce's CodeGen (Nijkamp et al., 2023b;a), Huawei's PanguCoder (Christopoulou et al., 2022; Shen et al., 2023), Meta's LLaMA (Touvron et al., 2023), and CMU's InCoder model (Fried et al., 2022) also contribute to the field. To adopt code LLMs in real scenarios, researchers have further explored methods to integrate dependencies of relevant code in the prompt (Shrivastava et al., 2023; Zhang et al., 2023a).

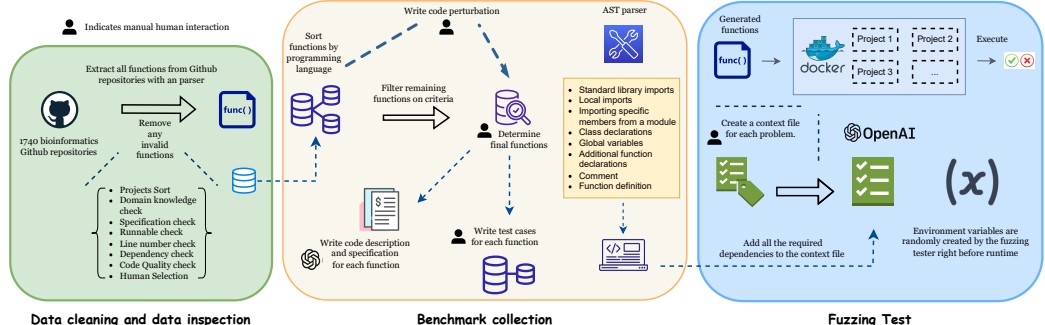

Figure 2: **A diagram** of the BIOCODER construction process involving custom GitHub repository cleaning, parsing, and function selection, as well as context and test case creation and a massively dockerized testing framework.

## 2.2 CODE GENERATION DATASETS AND BENCHMARKS

Early work on code generation benchmarks used lexical exact match, data flow, and abstract syntax tree (AST) methods. However, these measures proved to be unreliable due to their sensitivity to inconsequential differences in the generated code. In response, execution-based evaluation approaches have become more prevalent (Chen et al., 2021; Athiwaratkun et al., 2023; Li et al., 2022; Wang et al., 2022b; Lai et al., 2022; Khlaaf et al., 2022). These approaches execute tests on the generated code to verify its functional correctness, ensuring unbiased evaluations irrespective of implementation method or style variations.

As a result, the field of code generation has seen a burgeoning number of execution-based benchmarks (Table 1) (Yuan et al., 2023; Lee et al., 2023; Pan et al., 2023; Wong et al., 2023; Zan et al., 2023), each presenting unique properties in terms of size, language coverage (Orlanski et al., 2023), complexity (Du et al., 2023; Zhuo, 2023), and practical applicability (Yu et al., 2023). For instance, HumanEval (Chen et al., 2021) and MBPP (Austin et al., 2021b) are frequently used code generation benchmarks that consist of 164 and 974 simple Python functions respectively, representing a small sample size. These benchmarks also overlook the multi-language coding scenarios gap, which is partially bridged by benchmarks like HumanEval-X (Zheng et al., 2023b) and MCoNaLa (Wang et al., 2023b). See Zan et al. (2023) for a more comprehensive survey on the previous benchmarks of code generation.

However, all datasets discussed above share the same shortcoming of only benchmarking generic functions, rather than domain-specific ones. DS-1000 (Lai et al., 2022) represents a more domain-specific dataset, featuring 1,000 data science workflows extracted from Python functions. Li et al. (2023) reported that the performance on HumanEval and MBPP benchmarks do not invariably align

|  | Public | | | Hidden | | | Similar | | |
|---|---|---|---|---|---|---|---|---|---|
|  | PY | JAVA | Overall | PY | JAVA | Overall | PY | JAVA | Overall |
| **Avg. Comment Lines** | 4.96 | 2.66 | 4.40 | 8.77 | 4.90 | 6.65 | 5.75 | 3.14 | 5.12 |
| **Avg. Tokens of G.T.** | 189.25 | 106.54 | 169.28 | 353.67 | 107.88 | 219.02 | 216.62 | 100.92 | 188.68 |
| **Avg. Lines of G.T.** | 24.30 | 11.10 | 21.11 | 43.28 | 12.19 | 26.25 | 26.50 | 10.32 | 22.59 |
| **Avg. Parameters of G.T.** | 2.39 | 1.70 | 2.23 | 2.92 | 1.25 | 2.00 | 2.48 | 1.10 | 2.15 |
| **Avg. Classes/Function Decl.** | 20.25 | 2.52 | 15.97 | 19.45 | 32.96 | 26.85 | 20.20 | 1.16 | 15.60 |
| **Avg. Global Variables** | 1.90 | - | - | 2.26 | - | - | 1.87 | - | - |
| **Avg. Imports** | 11.91 | 1.52 | 9.40 | 10.37 | 5.00 | 7.43 | 11.63 | 1.16 | 9.10 |

Table 2: Summary statistics for the BIOCODER dataset. **G.T.** stands for the ground truth function. "Public data" represents datasets with test cases. "Hidden data" encompasses a wider array of intricate issues. "Similar data" is a subset of the hidden data, mimicking the distribution of the public data (Appendix z).

with those on DS-1000 benchmark. This discrepancy underscores the need for benchmarks that more accurately emulate real-world, domain-specific code generation.

In addition, the context supplied greatly influences the performance of existing LLMs (Wang et al., 2022a). While DS-1000 includes eight packages, it fails to fully reflect a typical coding environment. This gap is partially bridged through benchmarks such as CoderEval (Yu et al., 2023), which incorporate some dependencies and function calls; however, these benchmarks are rudimentary in nature, and once again consist primarily of domain-agnostic functions. As LLMs continue to develop, however, we are now beginning to see repository-level benchmarks that provide a high amount of context, but these remain new and untried, such as RepoBench (Liu et al., 2023).

Our work shares common ground with CoderEval. Both our approach and CoderEval can evaluate models beyond the simple generation of standalone functions. Given the necessity to handle context-dependent code, both methodologies employ Docker-based testing. However, our approach contrasts with that of CoderEval by placing a specific emphasis on bioinformatics. We ensure each function demands a certain level of domain expertise in bioinformatics by a combination of automatic filtering, GPT-assisted filtering, and manual inspection (Appendix t). Moreover, our dataset surpasses the scale of CoderEval, which only consists of 230 functions from 43 Python projects and 230 methods from 10 Java projects. In contrast, we source 2,522 functions from over two thousand repositories, offering a broader and more challenging context for code generation tasks. We further compare our benchmark to CoderEval in Appendix h.

## 3  THE BIOCODER BENCHMARK

### 3.1  DATASET FILTERING

Our dataset begins with an initial web scrape of 1,743 bioinformatics-adjacent GitHub repositories (see Figure 2). Specifically, we used the list of 1740 bioinformatics-adjacent repositories in Russell et al. (2018) as the initial base for BIOCODER, which contains a curated list of 1720 bioinformatics repositories from the literature. The collection includes code in languages such as C, C++, PHP, Python, R, Ruby, SQL, Perl, Java, Matlab, and C#, although for now, we only explore Python and Java, with plans to scale up to other languages in the future. Our decision to include Java and Python was based on an empirical investigation into the prevalence of different programming languages across bioinformatics repositories, for a more detailed discussion, please refer to Appendix q.

Those repositories were then filtered based on popularity and community ratings, as well as a manual round of review, resulting in 28 high-quality, highly domain-specific repositories that are commonly used in the field of bioinformatics. After determining the set of 28 high-quality, highly domain-specific repositories, we then wrote separate custom Python and Java parsers to automatically parse all the selected GitHub repositories. These parsers generated an AST of each code file in each repository and then scraped all the relevant data, including function content, function signature, important imports, and cross-file dependencies for each function in each code file. After parsing all repositories, we were left with a large set of over 20,000 Python functions and over 50,000 Java functions. Given the large baseline of functions, we initiated two rounds of automatic filtering (see Appendix t), resulting in a final count of 1,026 Python functions and 1,243 Java functions (Table 2). More details on the filtering process are in Appendix t.

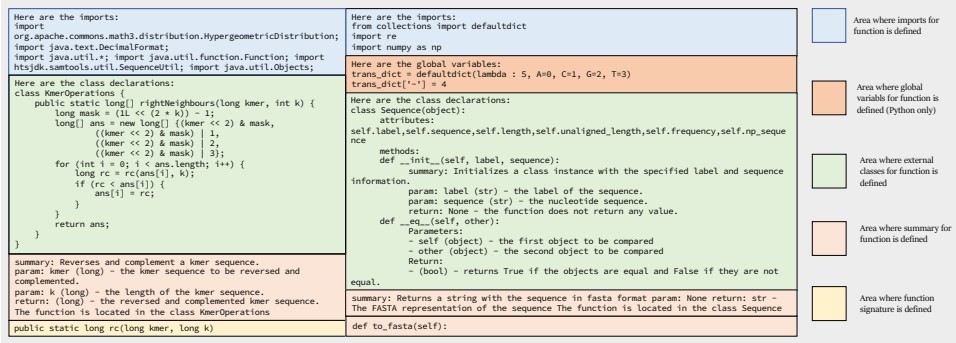

Figure 3: Sample prompts for code generation. Our prompts follow the same general outline. First, imports are declared at the top of the prompt, then global variables (if there are any), then function declarations, then class dependencies, and finally, our actual instruction regarding the function to be generated.

## 3.2 BENCHMARK CONSTRUCTION

**BIOCODER-PY and BIOCODER-JAVA** For each function that passes all rounds of filtering in Section 3.1, we manually wrote custom code context, inclusive of necessary imports, cross-file dependencies, and pertinent fuzzer test cases (explained in more detail in Section 3.4). We then crafted custom prompts based on the parsed function data and function summaries, ensuring the inclusion of any necessary imports and cross-file dependencies (see Appendix r). Imports and classes are predefined and included in the context because, as we are testing function-level code generation, we are not prompting the model nor expecting the model to generate the classes it needs to pass the tests. Instead, we are testing the model's ability to extract the pertinent imports and classes from the context to use in the generated function. More prompt statistics can be found in Table 7. Finally, we presented the model with a prompt to generate the function, offering the function signature as a starting point. Examples of the different prompt types can be found in Appendix c. Prompts were generated partly using GPT-3.5, as GPT-3.5 was used to generate function summaries for all the functions in the public dataset. These function summaries were used as part of the prompt in order to efficiently describe the functions. More details about this method are in the Appendix f. An example of the resulting prompt is in Figure 3.

**BIOCODER-ROSALIND.** To compile the Rosalind portion of the benchmark, we began by scraping the problem descriptions from the Rosalind website, identifying problems with available solutions, and gathering all possible solutions. Subsequently, we developed a custom scraper to assemble ten test cases for each Rosalind problem. Using these test cases, we crafted a script to automatically assess whether the available solutions successfully ran against the collected test cases.

Solutions that were successfully executed against all test cases formed the 'golden code' (producing the correct outputs when run with the test cases) section of the Rosalind benchmark. Each Rosalind benchmark context is custom-made, incorporating the scraped test cases and injecting them into the generated code. The prompts for the Rosalind problems are constructed using the scraped problem descriptions, supplemented with a brief section outlining the context into which the generated code would be integrated. This rigorous filtering process resulted in 253 functions meeting all our criteria. Selected examples for the Rosalind dataset are shown in Appendix d. Statistics of token counts, comment lines per function, and parameters per function can be found in Appendix b.

## 3.3 METRIC

We use the Pass@K metric to measure the functional accuracy (Chen et al., 2021; 2022; Cassano et al., 2023). The metric Pass@K evaluates the efficiency of a model's code generation ability. Specifically, for a given model, this metric quantifies the probability that the model can solve a particular problem. A problem is deemed "solve" if, among the k-function samples produced by the model, at least one sample passes all the provided test cases. The mathematical estimation of Pass@K for a particular

problem is articulated as follows: $\text{Pass@K} := \underset{\text{Problems}}{\mathbb{E}} \left[ 1 - \frac{\binom{n-c}{k}}{\binom{n}{k}} \right]$, where $n$ is the number of samples generated by the model, and $c$ is the number of samples that pass all test cases (Chen et al., 2021).

## 3.4 TESTING FRAMEWORK

Our testing framework starts with a manual review of selected functions, leading to the creation of a context file and a golden code for each problem (see Appendix r), as discussed in 3.2.

For Python and Java functions, in the context file, we employ a custom syntax to indicate the insertion points for custom randomly generated test cases. Through this syntax, we cater to four types of random generation: integers, floats, strings, and Boolean values. During the runtime, each of these insertion points is replaced with language-specific code to insert a dynamically generated test case. The tester can be run for any number of iterations, depending on the number of fuzzer tests desired.

For Rosalind functions, the process is simpler and more efficient as the functions are less complex. The golden code's output is generated and cached ahead of time. During testing, the tester executes the generated code within the corresponding context, and the output of this execution is compared with the cached golden code output.

For every fuzzer test case and Rosalind test case, we ran the golden output against itself, to ensure that the golden output passes each test with one hundred percent reliability. Furthermore, to ensure system security and test reliability, we ran our tests in Docker environments. We constructed a system using Amazon Web Services, coordinating tasks across multiple nodes to accelerate the process without compromising the validity of the results. After creating a generalized Docker image, equipped with all necessary Python requirements, we summarized our testing framework in Appendix l. We also addressed potential concerns about testing issues due to changes in packages in Appendix y.

## 4 MODELS AND RESULTS

To test BIOCODER, we opted to benchmark StarCoder-15B (Li et al., 2023), StarCoder+-15B (Li et al., 2023), InCoder (Fried et al., 2023), SantaCoder (Allal et al., 2023), CodeGen (6B-mono and 16B-mono) (Nijkamp et al., 2023b), CodeGen2-7B (Nijkamp et al., 2023a), InstructCodeT5+ (Wang et al., 2023a), GPT3.5-Turbo and GPT-4 (OpenAI, 2023) through Azure OpenAI Service. Full details of the model context lengths and model sizes can be found in Table 3.

In order to target specific performance characteristics, we came up with hundreds of variations of the prompt. We chose three goals: test the performance of models with extraneous context, without extraneous context, and without any context. These goals allow us to better analyze failure reasons and the effectiveness of our context-driven approach. After careful experimentation, we came up with the prompt type shown in Figure 3, which we call *Summary at Bottom*. Following the instruction paradigm of some of the considered models, we test a version with the summary moved to the top, along with the text "# Here is an instruction. Complete the function using the required context". To test without extraneous context, we used human annotators to manually determine the required context, and use the structure of the *Summary at Top* prompt. The remaining prompt explanations can be found in Appendix i.

Aiming to accurately represent the performance of the LLM outputs, we implemented basic correction mechanisms to rectify minor syntax and style errors that did not impact functionality. For instance, all StarCoder outputs were appended with a post-script. Consequently, each LLM output was passed through these correction mechanisms before being sent to the testing framework for evaluation (see Table 4 and 5). Furthermore, to empirically evaluate the hypothesis regarding the efficacy of smaller, specialized LLMs in closed-domain code generation, as opposed to large open-domain pre-trained models like GPT-3.5 and GPT-4, we also fine-tuned StarCoder and documented the resulting performance. Our aim is to use StarCoder as a representative sample of currently popular models. Due to computing restraints, we are unable to fine-tune all the models but we also encourage the contribution from the broader community. We ran inference on HPC clusters with 8x A100 GPUs.

The results in Table 4 and Table 5 align with our initial hypothesis, which proposed that larger models would likely outperform their smaller counterparts. However, the significant performance gap between GPT-3.5, GPT-4, and all other code-generation models was surprising. This stark contrast underscores the crucial role of both the dataset size and parameter size of the base models in

| Model | Context limit | # Parameters |
|---|---|---|
| InCoder (Fried et al., 2023) | *2048* | *6B* |
| SantaCoder (Allal et al., 2023) | *2048* | *1.1B* |
| StarCoder (Li et al., 2023) | *8192* | *15.5B* |
| StarCoderPlus (Li et al., 2023) | *8192* | *15.5B* |
| InstructCodeT5+ (Wang et al., 2023a) | *2048* | *16B* |
| CodeGen-6B (Nijkamp et al., 2023b) | *2048* | *6B* |
| CodeGen-16B (Nijkamp et al., 2023b) | *2048* | *16B* |
| CodeGen2 (Nijkamp et al., 2023a) | *2048* | *7B\** |
| GPT-3.5-Turbo | *8192* | *Unknown* |
| GPT-4 | *8192* | *Unknown* |

Table 3: Context length limits and sizes of different code LLMs.

accomplishing closed-domain code generation prompts. Java performance went up a lot since the structure is a bit more similar between the training set and testing set. This stark contrast underscores the crucial role of both the dataset size and parameter size of the base models in accomplishing closed-domain code generation prompts. Interestingly, despite the rudimentary nature of our fine-tuning on StarCoder, the results still highlighted a significant improvement compared with the non-fine-tuned model. This stark contrast in performance bolsters our original assertion: achieving success in closed-domain tasks can be realized either through large open-domain LLMs, or via fine-tuning smaller models. These smaller models could potentially achieve comparable performance but with significantly reduced computational and memory requirements. Furthermore, Table 4 demonstrates that the performance of models improves with the inclusion of dependencies in prompts, indicating that including dependencies is an important part of promoting. Without additional training, ChatGPT models performed notably better than other models. Their performance underscores the crucial role of both the dataset scale and model size. That said, the performance of other models (e.g. StarCoder) could be improved by fine-tuning.

## 5 ANALYSIS AND DISCUSSION

Looking more closely at the results in Table 4, it is clear that the larger models with more parameters generally perform better than the smaller models. GPT-4 model dwarfs the other models in this study in both size and performance. However, it is clear that BIOCODER remains a challenge as GPT-3.5 and GPT-4, the best model, only achieved an accuracy of slightly under 60%.

Looking at the other models, it is interesting to note that while InstructCodeT5+, CodeGen, and CodeGen2 are all larger than InCoder and SantaCoder, they perform far worse. This is likely due to the former being trained for single-line completions rather than function completion. Furthermore, InstructCodeT5+, CodeGen, and CodeGen2 have relatively small context limits (Mikolov et al., 2013; MOI et al., 2022), which likely hurts their performance. As for the remaining model, SantaCoder notably performs impressively well for being only a roughly 1B parameter model, which is an indication of aggressive fine-tuning on Python code.

We also note that the context length limit has a very large impact on how different models perform on different prompts. Except for GPT-3.5 and GPT-4, models performed the best on the *Summary Only* prompt style, likely because of its shorter length. Summary-only prompts are shortened prompts utilized across all our LLM models to ensure that context-limited LLMs still receive all the information necessary to potentially generate functions. Within the summary-only prompts, we optimized our prompts to only contain the absolute minimum of necessary information, without including much of the additional context that provides detail regarding the functionality of other dependencies. Looking at Figure 3, which contains the complete prompt structure, summary-only prompts would reduce the class declarations to only their declarations and one sentence describing their output and input. This is especially pronounced for InCoder and SantaCoder, as they both have small context limits of 2,048 tokens. Their Pass@K performance for Python decreases dramatically when switching from short *Summary Only* prompts to longer *Summary at Top/Bottom*.

| Model | Prompt | Java | | | | Python | | | |
|---|---|---|---|---|---|---|---|---|---|
| | | Pass@1 | Pass@5 | Pass@10 | Pass@20 | Pass@1 | Pass@5 | Pass@10 | Pass@20 |
| InCoder-6B | *Summary at Top* | 0 | 0 | 0 | 0 | 0.828 | 2.016 | 3.006 | 4.459 |
| | *Uncommented* | 0 | 0 | 0 | 0 | 0.032 | 0.159 | 0.318 | 0.637 |
| | *Summary Only* | 0 | 0 | 0 | 0 | 1.688 | 5.320 | 8.332 | 12.006 |
| | *Necessary Only* | 0 | 0 | 0 | 0 | 0.032 | 0.159 | 0.318 | 0.637 |
| SantaCoder-1.1B | *Summary at Top* | 0 | 0 | 0 | 0 | 0.637 | 1.338 | 1.844 | 2.548 |
| | *Uncommented* | 0 | 0 | 0 | 0 | 0.287 | 0.764 | 0.955 | 1.274 |
| | *Summary Only* | 0 | 0 | 0 | 0 | 2.965 | 9.848 | 14.227 | 18.181 |
| | *Necessary Only* | 0 | 0 | 0 | 0 | 0.032 | 0.159 | 0.318 | 0.637 |
| StarCoder-15.5B | *Summary at Top* | 0 | 0 | 0 | 0 | 3.694 | 13.197 | 19.359 | 24.554 |
| | *Uncommented* | 0 | 0 | 0 | 0 | 0.318 | 1.062 | 1.591 | 2.548 |
| | *Summary Only* | 0 | 0 | 0 | 0 | 4.682 | 15.225 | 21.200 | 27.166 |
| | *Necessary Only* | 0 | 0 | 0 | 0 | 0.127 | 0.603 | 1.123 | 1.911 |
| StarCoder-15.5B (finetuned) | *Summary at top* | 0 | 0 | 0 | 0 | 5.818 | 16.562 | 21.091 | 27.048 |
| | *Uncommented* | 0 | 0 | 0 | 0 | 3.312 | 9.073 | 12.574 | 17.536 |
| | *Summary Only* | 0.200 | 1.000 | 2.000 | 4.000 | 7.295 | 20.838 | 26.143 | 39.570 |
| | *Necessary Only* | 3.300 | 12.097 | 19.545 | 30.000 | 0.597 | 1.173 | 1.813 | 2.611 |
| StarCoder+ | *Summary at Top* | 0 | 0 | 0 | 0 | 2.675 | 9.133 | 14.019 | 19.650 |
| | *Uncommented* | 0 | 0 | 0 | 0 | 0.510 | 0.955 | 1.274 | 1.911 |
| | *Summary Only* | 1.300 | 5.031 | 8.042 | 12.000 | 2.548 | 8.279 | 12.864 | 18.057 |
| | *Necessary Only* | 0 | 0 | 0 | 0 | 0.127 | 0.457 | 0.609 | 0.637 |
| InstructCodeT5+ | *All prompt types* | 0 | 0 | 0 | 0 | 0 | 0 | 0 | 0 |
| CodeGen-6B-mono | *Summary at Top* | 0 | 0 | 0 | 0 | 0.637 | 0.637 | 0.637 | 0.637 |
| | *Uncommented* | 0 | 0 | 0 | 0 | 0 | 0 | 0 | 0 |
| | *Summary Only* | 0 | 0 | 0 | 0 | 0.637 | 0.637 | 0.637 | 0.637 |
| | *Necessary Only* | 0 | 0 | 0 | 0 | 0 | 0 | 0 | 0 |
| CodeGen-16B-mono | *Summary at Top* | 0 | 0 | 0 | 0 | 0.637 | 0.637 | 0.637 | 0.637 |
| | *Uncommented* | 0 | 0 | 0 | 0 | 0 | 0 | 0 | 0 |
| | *Summary Only* | 0 | 0 | 0 | 0 | 0.637 | 0.637 | 0.637 | 0.637 |
| | *Necessary Only* | 0 | 0 | 0 | 0 | 0 | 0 | 0 | 0 |
| CodeGen2-7B | *Summary at Top* | 0 | 0 | 0 | 0 | 0.637 | 0.637 | 0.637 | 0.637 |
| | *Uncommented* | 0 | 0 | 0 | 0 | 0.510 | 0.637 | 0.637 | 0.637 |
| | *Summary Only* | 0 | 0 | 0 | 0 | 0.860 | 2.494 | 3.962 | 6.242 |
| | *Necessary Only* | 0 | 0 | 0 | 0 | 0 | 0 | 0 | 0 |
| GPT-3.5-Turbo | *Summary at Top* | 4.100 | 7.235 | 8.989 | 11.600 | 22.771 | 33.461 | 36.551 | 39.490 |
| | *Uncommented* | 6.300 | 11.563 | 14.436 | 18.000 | 11.019 | 19.075 | 21.680 | 24.204 |
| | *Summary Only* | 17.400 | 33.199 | 37.878 | 42.000 | 24.682 | 33.997 | 37.132 | 40.127 |
| | *Necessary Only* | 43.500 | 52.582 | 53.995 | 55.400 | 28.758 | 39.529 | 44.029 | 47.771 |
| GPT-4 | *Summary at top* | 1.100 | 5.500 | 11.000 | 22.000 | 10.701 | 25.500 | 32.910 | 39.490 |
| | *Uncommented* | 6.367 | 11.234 | 15.897 | 18.562 | 12.654 | 20.129 | 24.387 | 27.932 |
| | *Summary Only* | 19.483 | 24.721 | 29.634 | 2.543 | 13.172 | 24.578 | 28.394 | 31.938 |
| | *Necessary Only* | **45.011** | **55.350** | **57.616** | **60.000** | **38.439** | **48.491** | **50.619** | **52.229** |

Table 4: Zero-shot and finetuned performance with five prompt versions of BIOCODER. For detailed explanations of prompt versions see Appendix i. For all settings, we performed trials twice for Pass@K. Results are expressed in percentages. We only finetuned StarCoder for 2000 steps, all others are zero-shot results. Additional results can be found in Appendix u.

As shown by the scatterplots in Appendix k, on models with an average Pass@K score of at least 2%, there is an inverse relationship between the number of tokens in the prompt and the Pass@K score. Furthermore, for models such as SantaCoder and GPT models, the performance fell sharply after around 500 tokens. Despite this, model performance can not only be attributed to prompt length. We can see that even though the "Necessary Only prompts are relatively shorter when compared to the "Summary at Top" or "Uncommented" prompts, the Pass@k performance of the "Uncommented"

| Model | Rosalind | | | | Model | Rosalind | | | |
|---|---|---|---|---|---|---|---|---|---|
| | P@1 | P@5 | P@10 | P@20 | | P@1 | P@5 | P@10 | P@20 |
| InCoder | 0.020 | 0.099 | 0.198 | 0.395 | InstructCodeT5+ | 0.059 | 0.296 | 0.593 | 1.186 |
| SantaCoder | 0.158 | 0.658 | 1.075 | 1.581 | CodeGen | 0.692 | 2.088 | 3.055 | 3.953 |
| StarCoder | 0.534 | 2.042 | 3.228 | 4.743 | CodeGen2 | 0.059 | 0.296 | 0.593 | 1.186 |
| StarCoderPlus | 0.356 | 1.313 | 1.978 | 2.767 | GPT-3.5 Turbo | 23.671 | 31.953 | 36.702 | 40.725 |
| StarCoder (FT) | 1.623 | 3.109 | 5.328 | 7.036 | GPT-4 | **24.308** | **39.551** | **44.864** | **50.198** |

Table 5: Performance of Rosalind. P@1 denotes Pass@1, etc. Numbers represent the Pass@K in the form of percentages. For all settings, n=20, and all models use the *Description* prompt.

| Failure/Success | Count | Percent (%) |
|---|---|---|
| Mismatched output | 8661 | 4.567 |
| Invalid syntax | 117665 | 62.038 |
| Runtime error | 55351 | 29.184 |
| Time out | 4 | 0.002 |
| Successfully Passed | 7982 | 4.209 |
| Total Testcases | 189663 | 100 |

Table 6: Aggregated Error Distribution Across All Models

| Prompt | Mean | Median | STDev |
|---|---|---|---|
| Java | 2278.82 | 2599.00 | 1331.81 |
| Python | 2790.75 | 2194.00 | 2539.79 |
| Rosalind | 564.49 | 509.00 | 286.47 |
| Overall | 1510.66 | 812.50 | 1882.80 |

Table 7: Prompt Tokens Distribution

prompts is in fact worse for many of the models. For more analysis regarding this and prompt structure in general, please refer to Appendix aa.

Focusing on Java's performance, it is clear that most of the publicly available LLMs have not been fine-tuned for Java, resulting in the near 0% Pass@K values. Finally, Rosalind's performance results in Table 5 are roughly in line with Python's performance in Table 4.

Table 6 provides an overview of the error statistics collected from our test runs. For more information about what each error means, see Appendix s. We also have error statistics per model in Appendix v Looking at Appendix s, it appears that the models struggle the most at writing code that will successfully compile or run. The fact that the number of samples of generated code that produced wrong output is relatively small compared to the code that failed to compile or run indicates that if the model is able to generate code that is able to be run, then that code is generally accurate and doesn't produce the wrong output. Therefore, it seems that models have the most trouble generating syntactically correct code rather than understanding the logic required to complete the problems outlined in the prompts. Further discussion on the results of each model can be found in Appendix j.

Despite these challenges, we firmly believe that this dataset holds pivotal importance for benchmarking future models, especially ones with larger context limits, like GPT-4-32k and Claude2.

# 6 CONCLUSIONS AND FUTURE WORK

Our study underscores the challenges in code generation, emphasizing the shortcomings of current models in the face of complex tasks. We present a highly challenging natural language to code tasks, providing input rich with dependencies and imports. Existing models struggle to comprehend the application of these imported toolkits or functions contained in other files. Our tasks are marked by extensive input and a high level of specialization. These programs are closer to real-world scenarios, requiring professional-level code-writing skills, rather than merely catering to beginners. This suggests that the code in question can typically be produced only by professional programmers.

As a novel benchmark within the field of bioinformatics, there remains a multitude of areas for future exploration. Currently, we have covered most of the existing models (at the time, August 2023). Additionally, we will move beyond function-level code generation as current models do not have the capacity of the token to generate file-sized code. We included only a few well-established bioinformatics repositories; in future work, without a manpower constraint, we could include additional repositories that span more niche sub-studies of bioinformatics and span across other languages.

## 7    ETHICS STATEMENT

We understand the importance of discussing the potential impacts of our specific work. In the context of our benchmark, one potential concern is the accuracy of the benchmark across all data points. There is a risk that the benchmark may optimize incorrect outputs, which users might then use to test their LLMs. This concern is especially significant in research settings deploying incorrect code could lead to inaccurate conclusions, initiating a snowball effect of misinformation. Beyond the immediate implications for LLMs and research outcomes, our benchmark and dataset potentially could be misused. For example, malicious users might use these data to train models that generate harmful biomedical experiments, such as designing dangerous molecules. While our intention is to advance knowledge and use it in a beneficial manner, there must be a level of caution and responsibility in employing the benchmark and dataset we provide.

## 8    REPRODUCIBILITY

Our dataset, benchmark, Docker images, corresponding prompts, and test scripts essential for the reproduction of our work can be accessed through the anonymous GitHub `https://anonymous.4open.science/r/BioCoder-86BD/`.

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

## A   HYPERPARAMETER SETTINGS

For all models, we used the following hyperparameter settings:

- top_k=50
- top_p=0.95
- temperature=0.7
- early_stopping=True
- num_return_sequences=1
- do_sample=True

## B   FUNCTION STATISTICS

Below are our distributions of comment lines, token lengths, and the number of parameters per function across our entire dataset. Generally, our number of parameters and comments are left-skewed, while our prompt lengths, while left-skewed, had a much more even distribution.

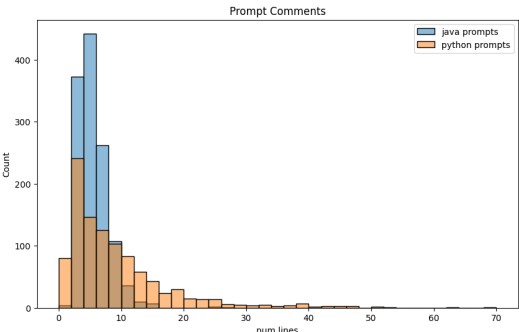

Figure 4: Comment lines per Function Distribution

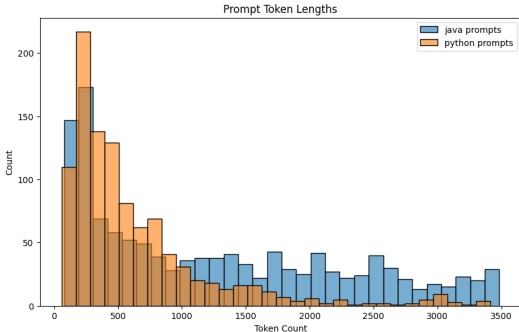

Figure 5: Prompt Token Length Distribution

Below is the distribution of the number of output tokens, averaged across either Python or Java. The vast concentration of the generations is concentrated at 0-250, as those are the minimum reserved tokens for generations if the prompts are too large. Across each language, we utilized nine models to generate outputs.

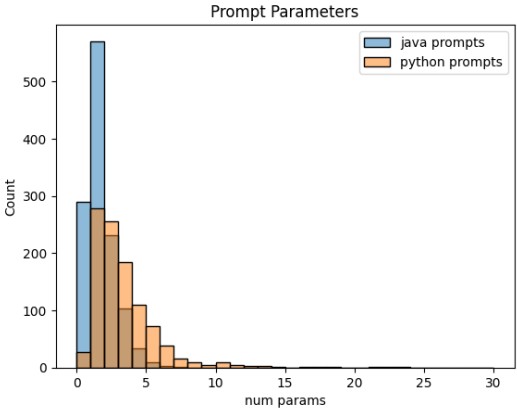

Figure 6: Parameters per Function Distribution

Figure 7: The distribution of the number of models' output tokens.

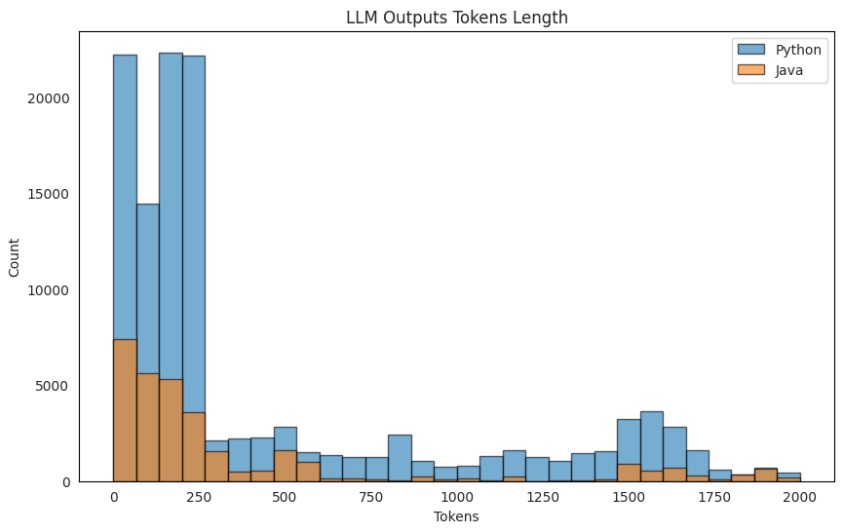

## C    PYTHON EXAMPLES

### C.1    PYTHON EXAMPLE 1

#### C.1.1    FULL PROMPT

```
1 #This is in python
2 #Write a function called "unpipe_name" that takes in a string parameter
      called "name". The function takes a string containing multiple gene
      names separated by pipes, and returns a single gene name devoid of
      duplicates and pipe characters. The function checks if there are any
      duplicates, and removes any meaningless target names. If there are
      multiple gene names present, the function takes the longest name as
      the final name. If there are any ambiguous gene names, the function
      logs a warning and selects the longest name as the final name. The
      function should also import "cnvlib.params".
3 #
4 #def unpipe_name(name):
5 #
6 #Here are the imports:
7 #import numpy as np
```

```
8  #import logging
9  #from . import params
10 #from skgenome import tabio
11 #Here are the global variables:
12 #MIN_REF_COVERAGE = -5.0
13 #MAX_REF_SPREAD = 1.0
14 #NULL_LOG2_COVERAGE = -20.0
15 #GC_MIN_FRACTION = 0.3
16 #GC_MAX_FRACTION = 0.7
17 #INSERT_SIZE = 250
18 #IGNORE_GENE_NAMES = '-', '.', 'CGH'
19 #ANTITARGET_NAME = 'Antitarget'
20 #ANTITARGET_ALIASES = ANTITARGET_NAME, 'Background'
21 #Here are the class declarations:
22 #Here are the additional function declarations:
23 #def do_import_picard(fname, too_many_no_coverage):
24 #    summary: Reads a file in 'picardhs' format, processes the data, and
          returns a modified dataframe.
25 #    param: fname (string) - the file name/path to be read in 'picardhs'
          format.
26 #    param: too_many_no_coverage (int) - if the number of bins with no
          coverage is greater than this value, a warning message is logged.
          Default is 100.
27 #    return: garr (pandas dataframe) - a modified dataframe with added
          columns 'gene' and 'log2' based on the original dataframe read from
          the input file.
28 #def unpipe_name(name):
29 #    summary: Remove duplicate gene names and pipe characters from a given
           string.
30 #    param: name (str) - the string containing the gene names.
31 #    return: new_name (str) - the cleaned gene name string.
32 #def do_import_theta(segarr, theta_results_fname, ploidy):
33 #    summary: A function for importing theta results and estimating copy
          number and log2 ratios of segments.
34 #    param: segarr (numpy array) - array of segments
35 #    param: theta_results_fname (str) - name of theta results file
36 #    param: ploidy (int) - ploidy of genome (default is 2)
37 #    return: generator of numpy arrays - array of segments with estimated
          copy number and log2 ratios.
38 #def parse_theta_results(fname):
39 #    summary: Parses THetA results into a data structure with NLL, mu, C,
          and p* columns.
40 #    param: fname (str) - name of the file to parse the results from
41 #    return: (dict) - a dictionary containing the NLL, mu_normal,
          mu_tumors, C, and p* values
42 #Here are the comments and the specs:
43 #Write a function called "unpipe_name" that takes in a string parameter
      called "name". The function takes a string containing multiple gene
      names separated by pipes, and returns a single gene name devoid of
      duplicates and pipe characters. The function checks if there are any
      duplicates, and removes any meaningless target names. If there are
      multiple gene names present, the function takes the longest name as
      the final name. If there are any ambiguous gene names, the function
      logs a warning and selects the longest name as the final name. The
      function should also import "cnvlib.params".
44 #def unpipe_name(name):
```

### C.1.2 SMALL PROMPT

```
1 Write a function with the following specs:
2 --specs begin here--
3 #Write a function called "unpipe_name" that takes in a string parameter
     called "name". The function takes a string containing multiple gene
     names separated by pipes, and returns a single gene name devoid of
```

```
      duplicates and pipe characters. The function checks if there are any
      duplicates, and removes any meaningless target names. If there are
      multiple gene names present, the function takes the longest name as
      the final name. If there are any ambiguous gene names, the function
      logs a warning and selects the longest name as the final name. The
      function should also import "cnvlib.params".
4  param: name (str) - the string containing the gene names.
5  return: new_name (str) - the cleaned gene name string.
6  --specs end here--
7  Note the function will be embedded in the following context
8  --context begins here--
9  import random
10 import hashlib
11 import numpy as np
12 import skimage
13 import skimage.measure
14 import scipy.ndimage
15 import os
16 import logging
17 from functools import wraps
18 from scipy import stats
19 import sys
20 import math
21 IGNORE_GENE_NAMES = ("-", ".", "CGH")
22 <<insert solution here>>
23 def main():
24     string1 = <|string|>
25     string2 = 'CGH'
26     name=f'{string1}|{string2}'
27     print(unpipe_name(name))
28 // context continues
29 --context ends here--
30 Make sure to only generate the function and not any of the context. Make
      sure you are generating valid, runnable code. Begin your solution
      with:
31 def unpipe_name(name):
32 MAKE SURE TO INDENT THE BODY OF YOUR FUNCTION BY A TAB
```

### C.1.3  NO COMMENT PROMPT

```
1  This is in python
2  Write a function called "unpipe_name" that takes in a string parameter
      called "name". The function takes a string containing multiple gene
      names separated by pipes, and returns a single gene name devoid of
      duplicates and pipe characters. The function checks if there are any
      duplicates, and removes any meaningless target names. If there are
      multiple gene names present, the function takes the longest name as
      the final name. If there are any ambiguous gene names, the function
      logs a warning and selects the longest name as the final name. The
      function should also import "cnvlib.params".
3
4  def unpipe_name(name):
5
6  Here are the imports:
7  import numpy as np
8  import logging
9  from . import params
10 from skgenome import tabio
11 Here are the global variables:
12 MIN_REF_COVERAGE = -5.0
13 MAX_REF_SPREAD = 1.0
14 NULL_LOG2_COVERAGE = -20.0
15 GC_MIN_FRACTION = 0.3
16 GC_MAX_FRACTION = 0.7
```

```
17  INSERT_SIZE = 250
18  IGNORE_GENE_NAMES = '-', '.', 'CGH'
19  ANTITARGET_NAME = 'Antitarget'
20  ANTITARGET_ALIASES = ANTITARGET_NAME, 'Background'
21  Here are the class declarations:
22  Here are the additional function declarations:
23  def do_import_picard(fname, too_many_no_coverage):
24      summary: Reads a file in 'picardhs' format, processes the data, and
            returns a modified dataframe.
25      param: fname (string) - the file name/path to be read in 'picardhs'
            format.
26      param: too_many_no_coverage (int) - if the number of bins with no
            coverage is greater than this value, a warning message is logged.
            Default is 100.
27      return: garr (pandas dataframe) - a modified dataframe with added
            columns 'gene' and 'log2' based on the original dataframe read from
            the input file.
28  def unpipe_name(name):
29      summary: Remove duplicate gene names and pipe characters from a given
             string.
30      param: name (str) - the string containing the gene names.
31      return: new_name (str) - the cleaned gene name string.
32  def do_import_theta(segarr, theta_results_fname, ploidy):
33      summary: A function for importing theta results and estimating copy
            number and log2 ratios of segments.
34      param: segarr (numpy array) - array of segments
35      param: theta_results_fname (str) - name of theta results file
36      param: ploidy (int) - ploidy of genome (default is 2)
37      return: generator of numpy arrays - array of segments with estimated
            copy number and log2 ratios.
38  def parse_theta_results(fname):
39      summary: Parses THetA results into a data structure with NLL, mu, C,
            and p* columns.
40      param: fname (str) - name of the file to parse the results from
41      return: (dict) - a dictionary containing the NLL, mu_normal,
            mu_tumors, C, and p* values
42  Here are the comments and the specs:
43  Write a function called "unpipe_name" that takes in a string parameter
        called "name". The function takes a string containing multiple gene
        names separated by pipes, and returns a single gene name devoid of
        duplicates and pipe characters. The function checks if there are any
        duplicates, and removes any meaningless target names. If there are
        multiple gene names present, the function takes the longest name as
        the final name. If there are any ambiguous gene names, the function
        logs a warning and selects the longest name as the final name. The
        function should also import "cnvlib.params".
44  def unpipe_name(name):
```

### C.1.4   REFERENCE CODE

```
1   def unpipe_name(name):
2       """Fix the duplicated gene names Picard spits out.
3
4       Return a string containing the single gene name, sans duplications
        and pipe
5       characters.
6
7       Picard CalculateHsMetrics combines the labels of overlapping
        intervals
8       by joining all labels with '|', e.g. 'BRAF|BRAF' -- no two distinct
9       targeted genes actually overlap, though, so these dupes are redundant
        .
10      Meaningless target names are dropped, e.g. 'CGH|FOO|-' resolves as '
        FOO'.
```

```
11    In case of ambiguity, the longest name is taken, e.g. "TERT|TERT
      Promoter"
12    resolves as "TERT Promoter".
13    """
14    if '|' not in name:
15        return name
16    gene_names = set(name.split('|'))
17    if len(gene_names) == 1:
18        return gene_names.pop()
19    cleaned_names = gene_names.difference(IGNORE_GENE_NAMES)
20    if cleaned_names:
21        gene_names = cleaned_names
22    new_name = sorted(gene_names, key=len, reverse=True)[0]
23    if len(gene_names) > 1:
24        logging.warning('WARNING: Ambiguous gene name %r; using %r', name
      ,
25            new_name)
26    return new_name
```

### C.1.5    ANNOTATION PROMPTS

```
1  #This is in python
2  #Write a function called "unpipe_name" that takes in a string parameter
      called "name". The function takes a string containing multiple gene
      names separated by pipes, and returns a single gene name devoid of
      duplicates and pipe characters. The function checks if there are any
      duplicates, and removes any meaningless target names. If there are
      multiple gene names present, the function takes the longest name as
      the final name. If there are any ambiguous gene names, the function
      logs a warning and selects the longest name as the final name. The
      function should also import "cnvlib.params".
3  #
4  #def unpipe_name(name):
```

### C.1.6    CONTEXT

```
1  import random
2  import hashlib
3  import numpy as np
4  import skimage
5  import skimage.measure
6  import scipy.ndimage
7  import os
8  import logging
9  from functools import wraps
10 from scipy import stats
11 import sys
12 import math
13 IGNORE_GENE_NAMES = ("-", ".", "CGH")
14 <<insert solution here>>
15 def main():
16     string1 = <|string|>
17     string2 = 'CGH'
18     name=f'{string1}|{string2}'
19     print(unpipe_name(name))
20 if __name__ == "__main__":
21     main()
```

## C.2    PYTHON EXAMPLE 2

### C.2.1    FULL PROMPT

```
1  #This is in python
2  #write a function called "UnifyLevels" that takes three parameters:
       baseLevel (a constant array of integers representing the base levels)
       , addonLevel (a constant array of integers representing the
       additional levels), and windowSize (an integer representing the
       window size). The function should merge the two lists of breakpoints,
        but drop addonLevel values that are within windowSize of baseLevel
       values. The function should return an array of integers called
       joinedLevel. If addonLevel is an empty array, the function should
       return baseLevel. The output should be sorted and of dtype=np.int_.
3  #
4  #def UnifyLevels(baseLevel, addonLevel, windowSize):
5  #
6  #Here are the imports:
7  #from scipy import stats
8  #import math
9  #import numpy as np
10 #import logging
11 #import pandas as pd
12 #Here are the global variables:
13 #Here are the class declarations:
14 #Here are the additional function declarations:
15 #def segment_haar(cnarr, fdr_q):
16 #    summary: Segment CNVkit data using HaarSeg algorithm
17 #    param: cnarr (CopyNumArray) - binned, normalized copy ratios
18 #    param: fdr_q (float) - false discovery rate q-value
19 #    return: segarr (CopyNumArray) - CBS data table as a CNVkit object
20 #def one_chrom(cnarr, fdr_q, chrom):
21 #    summary: This function segments copy number data for a single
       chromosome using the HaarSeg algorithm.
22 #    param: cnarr (pandas.DataFrame) - a dataframe with columns '
       chromosome', 'start', 'end', 'log2', and 'probes'.
23 #    param: fdr_q (float) - the false discovery rate threshold for
       segmenting the copy number data.
24 #    param: chrom (str) - the chromosome to segment.
25 #    return: table (pandas.DataFrame) - a dataframe with columns '
       chromosome', 'start', 'end', 'log2', 'gene', and 'probes',
       representing the segmented copy number data.
26 #def variants_in_segment(varr, segment, fdr_q):
27 #    summary: Generates a table of variant segments based on allele
       frequencies
28 #    param: varr (object) - variant data
29 #    param: segment (object) - genomic segment data
30 #    param: fdr_q (float) - false discovery rate threshold
31 #    return: table (object) - pandas DataFrame with segmented data
32 #def haarSeg(I, breaksFdrQ, W, rawI, haarStartLevel, haarEndLevel):
33 #    summary: Perform segmentation on a 1D array of log-ratio values
       according to the HaarSeg algorithm.
34 #    param: I (array) - A 1D array of log-ratio values, sorted according
       to their genomic location.
35 #    param: W (array) - Weight matrix, corresponding to quality of
       measurement, with values :math:'1/(\\sigma^2)'. Must have the same
       size as I.
36 #    param: rawI (array) - The minimum between the raw test-sample and
       control-sample coverages (before applying log ratio, but after any
       background reduction and/or normalization). These raw red / green
       measurments are used to detect low-value probes, which are more
       sensitive to noise. Used for the non-stationary variance compensation
       . Must have the same size as I.
37 #    param: breaksFdrQ (float) - The FDR q parameter. This value should
       lie between 0 and 0.5.
38 #    param: haarStartLevel (int) - The detail subband from which we start
       to detect peaks.
39 #    param: haarEndLevel (int) - The detail subband until which we use to
       detect peaks.
```

```
40 #    return: dict - Returns a dictionary containing the start and end
         points of each segment and the mean value of each segment.
41 #def FDRThres(x, q, stdev):
42 #    summary: Calculates the False Discovery Rate (FDR) threshold.
43 #    param: x (unknown type) - array of values.
44 #    param: q (unknown type) - a floating-point number.
45 #    param: stdev (unknown type) - a floating-point number representing
         the standard deviation.
46 #    return: T (unknown type) - a floating-point number representing the
         FDR threshold.
47 #def SegmentByPeaks(data, peaks, weights):
48 #    summary: Average the values of the probes within each segment.
49 #    param: data (array) - the probe array values
50 #    param: peaks (array) - Positions of copy number breakpoints in the
         original array
51 #    param: weights (None or array) - optional array of weights of same
         length as the data array
52 #    return: segs (array) - array of segment values obtained by averaging
         the values of the probes within each segment.
53 #def HaarConv(signal, weight, stepHalfSize):
54 #    summary: Convolve haar wavelet function with a signal, applying
         circular padding.
55 #    param: signal (array of floats) - signal to be convolved.
56 #    param: weight (array of floats) - optional weights for the steps of
         the convolution.
57 #    param: stepHalfSize (int) - half size of the step to be used in the
         convolution.
58 #    return: array (array of floats) - of floats, representing the
         convolved signal.
59 #def FindLocalPeaks(signal):
60 #    summary: Finds local maxima on positive values, local minima on
         negative values.
61 #    param: signal (const array of floats): an array of floating point
         numbers
62 #    return: peakLoc (array of ints): Locations of extrema in ‘signal‘
63 #def UnifyLevels(baseLevel, addonLevel, windowSize):
64 #    summary: Merge two lists of breakpoints and drop addonLevel values
         that are too close to baseLevel values.
65 #    param: baseLevel (const array of ints) - a list of integers
         representing the base level.
66 #    param: addonLevel (const array of ints) - a list of integers
         representing the addon level.
67 #    param: windowSize (int) - an integer representing the maximum
         distance between addonLevel and baseLevel values.
68 #    return: joinedLevel (array of ints) - a sorted array of integers
         representing the merged list of breakpoints.
69 #def PulseConv(signal, pulseSize):
70 #    summary: Convolve a pulse function with a signal applying circular
         padding to the signal for non-stationary variance compensation.
71 #    param: signal (const array of floats) - the signal to be convolved.
72 #    param: pulseSize (int) - the size of the pulse function.
73 #    return: array of floats - the convolved signal.
74 #def AdjustBreaks(signal, peakLoc):
75 #    summary: Improve localization of breaks in a signal by adjusting peak
         locations.
76 #    param: signal (const array of floats) - the signal to adjust the peak
         locations in.
77 #    param: peakLoc (const array of ints) - the current peak locations in
         the signal.
78 #    return: newPeakLoc (array of ints) - the adjusted peak locations in
         the signal.
79 #def table2coords(seg_table):
80 #    summary: Returns x and y arrays for plotting with the help of the
         input segment Table
```

```
81  #    param: seg_table (list) - a list of tuples containing start, size,
        and value.
82  #    return: x (list) - a list of x-coordinates for plotting.
83  #    return: y (list) - a list of y-coordinates for plotting.
84  #Here are the comments and the specs:
85  #write a function called "UnifyLevels" that takes three parameters:
        baseLevel (a constant array of integers representing the base levels)
        , addonLevel (a constant array of integers representing the
        additional levels), and windowSize (an integer representing the
        window size). The function should merge the two lists of breakpoints,
         but drop addonLevel values that are within windowSize of baseLevel
        values. The function should return an array of integers called
        joinedLevel. If addonLevel is an empty array, the function should
        return baseLevel. The output should be sorted and of dtype=np.int_.
86  #def UnifyLevels(baseLevel, addonLevel, windowSize):
```

### C.2.2  SMALL PROMPT

```
1  Write a function with the following specs:
2  --specs begin here--
3  #write a function called "UnifyLevels" that takes three parameters:
       baseLevel (a constant array of integers representing the base levels)
       , addonLevel (a constant array of integers representing the
       additional levels), and windowSize (an integer representing the
       window size). The function should merge the two lists of breakpoints,
        but drop addonLevel values that are within windowSize of baseLevel
       values. The function should return an array of integers called
       joinedLevel. If addonLevel is an empty array, the function should
       return baseLevel. The output should be sorted and of dtype=np.int_.
4  param: baseLevel (const array of ints) - a list of integers representing
       the base level.
5  param: addonLevel (const array of ints) - a list of integers representing
        the addon level.
6  param: windowSize (int) - an integer representing the maximum distance
       between addonLevel and baseLevel values.
7  return: joinedLevel (array of ints) - a sorted array of integers
       representing the merged list of breakpoints.
8  --specs end here--
9  Note the function will be embedded in the following context
10 --context begins here--
11 import random
12 import hashlib
13 import numpy as np
14 import skimage
15 import skimage.measure
16 import scipy.ndimage
17 import os
18 import logging
19 from functools import wraps
20 from scipy import stats
21 import sys
22 import math
23 <<insert solution here>>
24 def main():
25     np.random.seed(<|int;range=0,1000|>)
26     baseLevel = np.random.randint(20, size=(10))
27     np.random.seed(<|int;range=0,1000|>)
28     addonLevel = np.random.randint(20, size=(10))
29     print(UnifyLevels(baseLevel, addonLevel, 3))
30 if __name__ == "__main__":
31 // context continues
32 --context ends here--
```

```
33 Make sure to only generate the function and not any of the context. Make
      sure you are generating valid, runnable code. Begin your solution
      with:
34 def UnifyLevels(baseLevel, addonLevel, windowSize):
35 MAKE SURE TO INDENT THE BODY OF YOUR FUNCTION BY A TAB
```

### C.2.3 NO COMMENT PROMPT

```
1 This is in python
2 write a function called "UnifyLevels" that takes three parameters:
      baseLevel (a constant array of integers representing the base levels)
      , addonLevel (a constant array of integers representing the
      additional levels), and windowSize (an integer representing the
      window size). The function should merge the two lists of breakpoints,
       but drop addonLevel values that are within windowSize of baseLevel
      values. The function should return an array of integers called
      joinedLevel. If addonLevel is an empty array, the function should
      return baseLevel. The output should be sorted and of dtype=np.int_.
3
4 def UnifyLevels(baseLevel, addonLevel, windowSize):
5
6 Here are the imports:
7 from scipy import stats
8 import math
9 import numpy as np
10 import logging
11 import pandas as pd
12 Here are the global variables:
13 Here are the class declarations:
14 Here are the additional function declarations:
15 def segment_haar(cnarr, fdr_q):
16     summary: Segment CNVkit data using HaarSeg algorithm
17     param: cnarr (CopyNumArray) - binned, normalized copy ratios
18     param: fdr_q (float) - false discovery rate q-value
19     return: segarr (CopyNumArray) - CBS data table as a CNVkit object
20 def one_chrom(cnarr, fdr_q, chrom):
21     summary: This function segments copy number data for a single
      chromosome using the HaarSeg algorithm.
22     param: cnarr (pandas.DataFrame) - a dataframe with columns '
      chromosome', 'start', 'end', 'log2', and 'probes'.
23     param: fdr_q (float) - the false discovery rate threshold for
      segmenting the copy number data.
24     param: chrom (str) - the chromosome to segment.
25     return: table (pandas.DataFrame) - a dataframe with columns '
      chromosome', 'start', 'end', 'log2', 'gene', and 'probes',
      representing the segmented copy number data.
26 def variants_in_segment(varr, segment, fdr_q):
27     summary: Generates a table of variant segments based on allele
      frequencies
28     param: varr (object) - variant data
29     param: segment (object) - genomic segment data
30     param: fdr_q (float) - false discovery rate threshold
31     return: table (object) - pandas DataFrame with segmented data
32 def haarSeg(I, breaksFdrQ, W, rawI, haarStartLevel, haarEndLevel):
33     summary: Perform segmentation on a 1D array of log-ratio values
      according to the HaarSeg algorithm.
34     param: I (array) - A 1D array of log-ratio values, sorted according
      to their genomic location.
35     param: W (array) - Weight matrix, corresponding to quality of
      measurement, with values :math:'1/(\\sigma^2)'. Must have the same
      size as I.
36     param: rawI (array) - The minimum between the raw test-sample and
      control-sample coverages (before applying log ratio, but after any
      background reduction and/or normalization). These raw red / green
```

```
        measurments are used to detect low-value probes, which are more
        sensitive to noise. Used for the non-stationary variance compensation
        . Must have the same size as I.
37      param: breaksFdrQ (float) - The FDR q parameter. This value should
        lie between 0 and 0.5.
38      param: haarStartLevel (int) - The detail subband from which we start
        to detect peaks.
39      param: haarEndLevel (int) - The detail subband until which we use to
        detect peaks.
40      return: dict - Returns a dictionary containing the start and end
        points of each segment and the mean value of each segment.
41  def FDRThres(x, q, stdev):
42      summary: Calculates the False Discovery Rate (FDR) threshold.
43      param: x (unknown type) - array of values.
44      param: q (unknown type) - a floating-point number.
45      param: stdev (unknown type) - a floating-point number representing
        the standard deviation.
46      return: T (unknown type) - a floating-point number representing the
        FDR threshold.
47  def SegmentByPeaks(data, peaks, weights):
48      summary: Average the values of the probes within each segment.
49      param: data (array) - the probe array values
50      param: peaks (array) - Positions of copy number breakpoints in the
        original array
51      param: weights (None or array) - optional array of weights of same
        length as the data array
52      return: segs (array) - array of segment values obtained by averaging
        the values of the probes within each segment.
53  def HaarConv(signal, weight, stepHalfSize):
54      summary: Convolve haar wavelet function with a signal, applying
        circular padding.
55      param: signal (array of floats) - signal to be convolved.
56      param: weight (array of floats) - optional weights for the steps of
        the convolution.
57      param: stepHalfSize (int) - half size of the step to be used in the
        convolution.
58      return: array (array of floats) - of floats, representing the
        convolved signal.
59  def FindLocalPeaks(signal):
60      summary: Finds local maxima on positive values, local minima on
        negative values.
61      param: signal (const array of floats): an array of floating point
        numbers
62      return: peakLoc (array of ints): Locations of extrema in 'signal'
63  def UnifyLevels(baseLevel, addonLevel, windowSize):
64      summary: Merge two lists of breakpoints and drop addonLevel values
        that are too close to baseLevel values.
65      param: baseLevel (const array of ints) - a list of integers
        representing the base level.
66      param: addonLevel (const array of ints) - a list of integers
        representing the addon level.
67      param: windowSize (int) - an integer representing the maximum
        distance between addonLevel and baseLevel values.
68      return: joinedLevel (array of ints) - a sorted array of integers
        representing the merged list of breakpoints.
69  def PulseConv(signal, pulseSize):
70      summary: Convolve a pulse function with a signal applying circular
        padding to the signal for non-stationary variance compensation.
71      param: signal (const array of floats) - the signal to be convolved.
72      param: pulseSize (int) - the size of the pulse function.
73      return: array of floats - the convolved signal.
74  def AdjustBreaks(signal, peakLoc):
75      summary: Improve localization of breaks in a signal by adjusting peak
         locations.
```

```
76    param: signal (const array of floats) - the signal to adjust the peak
        locations in.
77    param: peakLoc (const array of ints) - the current peak locations in
      the signal.
78    return: newPeakLoc (array of ints) - the adjusted peak locations in
      the signal.
79 def table2coords(seg_table):
80    summary: Returns x and y arrays for plotting with the help of the
      input segment Table
81    param: seg_table (list) - a list of tuples containing start, size,
      and value.
82    return: x (list) - a list of x-coordinates for plotting.
83    return: y (list) - a list of y-coordinates for plotting.
84 Here are the comments and the specs:
85 write a function called "UnifyLevels" that takes three parameters:
      baseLevel (a constant array of integers representing the base levels)
      , addonLevel (a constant array of integers representing the
      additional levels), and windowSize (an integer representing the
      window size). The function should merge the two lists of breakpoints,
       but drop addonLevel values that are within windowSize of baseLevel
      values. The function should return an array of integers called
      joinedLevel. If addonLevel is an empty array, the function should
      return baseLevel. The output should be sorted and of dtype=np.int_.
86 def UnifyLevels(baseLevel, addonLevel, windowSize):
```

### C.2.4 REFERENCE CODE

```
1 def UnifyLevels(baseLevel, addonLevel, windowSize):
2    """Unify several decomposition levels.
3
4    Merge the two lists of breakpoints, but drop addonLevel values that
      are too
5    close to baseLevel values.
6
7    Parameters
8    ----------
9    baseLevel : const array of ints
10   addonLevel : const array of ints
11   windowSize : int
12
13   Returns
14   -------
15   joinedLevel : array of ints
16
17   Source: HaarSeg.c
18   """
19   if not len(addonLevel):
20       return baseLevel
21   joinedLevel = []
22   addon_idx = 0
23   for base_elem in baseLevel:
24       while addon_idx < len(addonLevel):
25           addon_elem = addonLevel[addon_idx]
26           if addon_elem < base_elem - windowSize:
27               joinedLevel.append(addon_elem)
28               addon_idx += 1
29           elif base_elem - windowSize <= addon_elem <= base_elem +
      windowSize:
30               addon_idx += 1
31           else:
32               assert base_elem + windowSize < addon_elem
33               break
34       joinedLevel.append(base_elem)
35   last_pos = baseLevel[-1] + windowSize if len(baseLevel) else -1
```

```
36      while addon_idx < len(addonLevel) and addonLevel[addon_idx] <=
        last_pos:
37          addon_idx += 1
38      if addon_idx < len(addonLevel):
39          joinedLevel.extend(addonLevel[addon_idx:])
40      return np.array(sorted(joinedLevel), dtype=np.int_)
```

### C.2.5 CONTEXT

```
1  import random
2  import hashlib
3  import numpy as np
4  import skimage
5  import skimage.measure
6  import scipy.ndimage
7  import os
8  import logging
9  from functools import wraps
10 from scipy import stats
11 import sys
12 import math
13 <<insert solution here>>
14 def main():
15     np.random.seed(<|int;range=0,1000|>)
16     baseLevel = np.random.randint(20, size=(10))
17     np.random.seed(<|int;range=0,1000|>)
18     addonLevel = np.random.randint(20, size=(10))
19     print(UnifyLevels(baseLevel, addonLevel, 3))
20 if __name__ == "__main__":
21     main()
```

## D    ROSALIND EXAMPLES

The Rosalind dataset consists of 253 problems from the bioinformatics training website rosalind.info. For each of the 253 problems that make up the Rosalind portion of BIOCODER, we created both a prompt that we fed into the LLMs as well as a golden code solution that we either wrote ourselves or found on public GitHub repositories.

For each problem, to create the prompt, we first scraped the problem's page on the Rosalind website to obtain a problem description. This problem description would then make up the first section of the prompt. For instance, for the Rosalind problem ba1a, here is the scraped problem description:

```
1  # This is the first problem in a collection of "code challenges" to
        accompany Bioinformatics Algorithms: An Active-Learning Approach by
        Phillip Compeau & Pavel Pevzner.
2  # A k-mer is a string of length k.
3  # We define Count(Text, Pattern) as the number of times that a k-mer
        Pattern appears as a substring of Text.
4  # For example,
5  # $\textit{Count}(\text{ACA}\color{green}\textbf{ACTAT}\color{black}\text
        {GCAT}\color{green}\textbf{ACTAT}\color{black}\text{CGGGA}\color{
        green}\textbf{ACTAT}\color{black}\text{CCT}, {\color{green}\textbf{
        ACTAT}}) = 3$.
6  # We note that Count($\text{CG}\color{green}\textbf{ATATA}\color{black}\
        text{TCC}\color{green}\textbf{ATA}\color{black}\text{G}$, $\color{
        green}\textbf{ATA}$) is equal to 3 (not 2) since we should account
        for overlapping occurrences of Pattern in Text.
7  # To compute Count(Text, Pattern), our plan is to "slide a window" down
        Text, checking whether each k-mer substring of Text matches Pattern.
        We will therefore refer to the k-mer starting at position i of Text
        as Text(i, k). Throughout this book, we will often use 0-based
        indexing, meaning that we count starting at 0 instead of 1. In this
        case, Text begins at position 0 and ends at position |Text| - 1 (|
        Text| denotes the number of symbols in Text). For example, if Text =
        GACCATACTG,
8  # then Text(4, 3) = ATA. Note that the last k-mer of Text begins at
        position |Text| - k, e.g., the last 3-mer of GACCATACTG starts at
        position 10 - 3 = 7. This discussion results in the following
        pseudocode for computing Count(Text, Pattern).
9  # PatternCount(Text, Pattern)
10 # count <- 0
11 # for i <- 0 to |Text| - |Pattern|
12 # if Text(i, |Pattern|) = Pattern
13 # count <-s count + 1
14 # return count
15 # Implement PatternCount
16 # Given: {{DNA strings}} Text and Pattern.
17 # Return: Count(Text, Pattern).
18 # Sample Dataset
19 # GCGCG
20 # GCG
21 # Sample Output
22 # 2
```

From the sample problem description above you can see that a problem description generally consists of an explanation of the problem at hand, as well as a sample test case. The second section of the prompt consists of a custom block of text that prompts the LLM to generate its code to fit the context that generated Rosalind code will be run in. For instance, for the same Rosalind problem ba1a, here is the custom block of text that makes up the second section of the prompt:

```
1      #write the solve() function to solve the above problem
2  #Do NOT print anything
3  #Do NOT make comments
4  #Do NOT call the main() function.
5  #Use 4 spaces for tabs.
6  #input_str: a string
```

```
7  #return output: another string
8
9
10 def main():
11     with open("input.txt", "r") as f:
12         output = solve(f.read())
13     print(output)
14
15 # Write your solution here
16 # Begin with: def solve(input_str):
```

From the example above, you can see that the custom block of text consists of a prompt telling the LLM to generate a `solve` function as well as the context that the generated code will be run in, namely the `main` function that is included in the custom text. Each prompt consists of these two sections, i.e., the first section of the prompt will include a problem description consisting of an explanation of the problem as well as a sample test case, and the second section of the prompt will consist of custom prompting text that tells the LLM to generate a `solve` function and provides the context in which the Rosalind code will be run.

For each of the 253 problems that make up the Rosalind portion of BIOCODER, we also either found online or wrote custom golden code solutions. Each golden code solution is meant to be run in the custom context, following the `main` and `solve` structure that is described in every Rosalind prompt. For instance, here is the golden code solution for the Rosalind problem ba1a:

```
1  # Compute the Number of Times a Pattern Appears in a Text
2
3  def substrings(text, size):
4      for i in range(len(text) - size + 1):
5          yield text[i : i + size]
6
7  def pattern_count(text, pattern):
8      return sum(pattern == x for x in substrings(text, len(pattern)))
9
10 def solve(input_str):
11     text, pattern = input_str.splitlines()
12
13     return pattern_count(text, pattern)
```

As you can see, a golden code solution consists of the `solve` function requested by the corresponding prompt, as well as any additional helper functions that may be called (possibly recursively) by the `solve` function. For more information on the Rosalind portion of the BIOCODER dataset, three detailed examples of Rosalind prompts and golden code are provided below to further illustrate the differences between the code and the prompt.

### D.1 EXAMPLE 1

The given input is a problem description in the field of bioinformatics, namely the Reverse Complement Problem. The problem is related to the DNA sequences, which are strings containing the characters 'A', 'T', 'G', and 'C'. Each of these characters represents a nucleotide in the DNA.

The problem requires model to find the reverse complement of a given DNA string. The reverse complement of a DNA string is obtained by replacing each nucleotide in the string with its complement ('A' with 'T', 'T' with 'A', 'C' with 'G', and 'G' with 'C'), and then reversing the entire string. For example, the reverse complement of "GTCA" is "TGAC."

The problem description includes a sample dataset (DNA string) and its corresponding output, which is its reverse complement. The problem also specifies the format for the solution: a Python function named 'solve', which takes a DNA string as an input and returns its reverse complement as output. The function should not print anything or include any comments.

```
1
2  Reverse Complement Problem:
3  In DNA strings, symbols 'A' and 'T' are complements of each other, as are
       'C' and 'G'.
```

```
4  Given a nucleotide p, we denote its complementary nucleotide as p. The
       reverse complement of a DNA string Pattern = p1âĂępn is the string
       Pattern = pn âĂę p1 formed by taking the complement of each
       nucleotide in Pattern, then reversing the resulting string.
5
6  For example, the reverse complement of Pattern = "GTCA" is Pattern = "
       TGAC".
7
8
9  Problem:
10 Find the reverse complement of a DNA string.
11 Given: A DNA string Pattern.
12 Return: Pattern, the reverse complement of Pattern.
13
14 Sample Dataset:
15 AAAACCCGGT
16
17 Sample Output:
18 ACCGGGTTTT
19
20 write the solve() function to solve the above problem.
21
22 Do NOT print anything
23 Do NOT make comments
24 Do NOT call the main() function.
25 Use 4 spaces for tabs.
26
27 input_str: a string
28 return output: another string
29
30 def main():
31     with open("input.txt", "r") as f:
32         output = solve(f.read())
33     print(output)
34
35
36 Write your solution here:
37 Begin with: def solve(input_str):
```

The given sample solution to the problem is written in Python. It defines two functions: 'revcomp' and 'main'. The 'revcomp' function computes the reverse complement of a DNA string by first reversing the string and then replacing each nucleotide with its complement using Python's 'str.maketrans' method. The 'main' function reads a DNA string from a file, computes its reverse complement using the 'revcomp' function, and then prints the result.

The ground-truth code is a solution to the problem presented in the 'Rosalind' platform, a platform that provides a collection of bioinformatics problems. This platform is known for helping students and researchers learn bioinformatics by solving the computational problems it presents.

```
1  Find the Reverse Complement of a String
2
3
4  def revcomp(seq):
5      return seq[::-1].translate(str.maketrans("ACGT", "TGCA"))
6
7
8  def main(file):
9      seq = open(file).read().splitlines()[0]
10     print(revcomp(seq))
```

## D.2   EXAMPLE 2

The input here consists of several parts.

The initial portion discusses the Hardy Weinberg Principle and Mendel's laws, which are used in genetics to predict how genes will distribute throughout a population over time. The Hardy Weinberg Principle assumes that a population is large and remains in genetic equilibrium, meaning the frequency of each allele (a version of a gene) remains constant across generations, under certain conditions.

The problem statement is presented: if we know the proportion of homozygous recessive individuals (those with two copies of the recessive allele) for each of several genetic factors in a population, can we predict the probability that a randomly selected individual carries at least one copy of the recessive allele for each factor? The task is to write a Python function named 'solve' to solve this problem.

A sample dataset is given, represented as an array of floating-point numbers, each of which indicates the proportion of homozygous recessive individuals for a specific genetic factor in the population. The 'solve' function will receive this array as a string and should return another string representing the solution array.

The final portion of the input is a sample Python code that reads the dataset from a file, calls the 'solve' function to compute the solution, and then prints the solution. However, the user is instructed not to print anything, not to write comments, and not to call this 'main' function in your solution. The user is also instructed to use four spaces for indentation.

```
1  Genetic Drift and the Hardy-Weinberg Principle Mendel's laws of
       segregation and independent assortment are excellent for the study of
        individual organisms and their progeny, but they say nothing about
       how alleles move through a population over time.
2
3  Our first question is: when can we assume that the ratio of an allele in
       a population, called the allele frequency, is stable? G. H. Hardy and
        Wilhelm Weinberg independently considered this question at the turn
       of the 20th Century, shortly after Mendel's ideas had been
       rediscovered. They concluded that the percentage of an allele in a
       population of individuals is in genetic equilibrium when five
       conditions are satisfied: The population is so large that random
       changes in the allele frequency are negligible. No new mutations are
       affecting the gene of interest; The gene does not influence survival
       or reproduction, so natural selection is not occurring; Gene flow, or
        the change in allele frequency due to migration into and out of the
       population, is negligible. Mating occurs randomly with respect to the
        gene of interest. The Hardy-Weinberg principle states that if a
       population is in genetic equilibrium for a given allele, then its
       frequency will remain constant and evenly distributed through the
       population. Unless the gene in question is important to survival or
       reproduction, Hardy-Weinberg usually offers a reasonable enough model
        of population genetics. One of the many benefits of the Mendelian
       theory of inheritance and simplifying models like Hardy-Weinberg is
       that they help us predict the probability with which genetic diseases
        will be inherited, so as to take appropriate preventative measures.
       Genetic diseases are usually caused by mutations to chromosomes,
       which are passed on to subsequent generations. The simplest and most
       widespread case of a genetic disease is a single gene disorder, which
        is caused by a single mutated gene. Over 4,000 such human diseases
       have been identified, including cystic fibrosis and sickle-cell
       anemia. In both of these cases, the individual must possess two
       recessive alleles for a gene in order to contract the disease. Thus,
       carriers can live their entire lives without knowing that they can
       pass the disease on to their children. The above introduction to
       genetic equilibrium leaves us with a basic and yet very practical
       question regarding gene disorders: if we know the number of people
       who have a disease encoded by a recessive allele, can we predict the
       number of carriers in the population?
4
5  Problem:
6
7  To model the Hardy-Weinberg principle, assume that we have a population
       of $N$ diploid individuals. If an allele is in genetic equilibrium,
       then because mating is random, we may view the $2N$ chromosomes as
```

```
        receiving their alleles uniformly. In other words, if there are $m$
        dominant alleles, then the probability of a selected chromosome
        exhibiting the dominant allele is simply $p = \frac{m}{2N}$. Because
        the first assumption of genetic equilibrium states that the
        population is so large as to be ignored, we will assume that $N$ is
        infinite, so that we only need to concern ourselves with the value of
         $p$.

Given: An array $A$ for which $A[k]$ represents the proportion of
        homozygous recessive individuals for the $k$-th Mendelian factor in a
         diploid population. Assume that the population is in genetic
        equilibrium for all factors.

Return: An array $B$ having the same length as $A$ in which $B[k]$
        represents the probability that a randomly selected individual
        carries at least one copy of the recessive allele for the $k$-th
        factor.

Sample Dataset
0.1 0.25 0.5

write the solve() function to solve the above problem

Do NOT print anything
Do NOT make comments
Do NOT call the main() function.

Use 4 spaces for tabs.

input_str: a string
return output: another string

def main():
    with open("input.txt", "r") as f:
        output = solve(f.read())
    print(output)

Write your solution here
Begin with: def solve(input_str):
```

In the output portion, a Python function afrq is presented, which takes an array of proportions of homozygous recessive individuals for each genetic factor in a population and returns an array of probabilities that a randomly selected individual carries at least one copy of the recessive allele for each factor. The main function uses this afrq function to solve the problem based on the input data. It takes the input data from a file, applies the afrq function to compute the solution, and then prints the solution in the form of a space-separated string of floating-point numbers, each rounded to three decimal places.

```
# Counting Disease Carriers

from math import sqrt
from .helpers import Parser

def afrq(a):
    """Counting Disease Carriers"""
    return [2 * sqrt(x) - x for x in a]

def main(file):
    b = afrq(Parser(file).floats())
    print(*[round(x, 3) for x in b])
```

D.3 EXAMPLE 3

This Python script is designed to find the most frequent k-mers (substrings of length k) in a given DNA sequence (a string), while allowing for a specified number of mismatches. This problem is referred to as the "Frequent Words with Mismatches Problem".

The input to this problem is a string that contains two distinct parts separated by a newline:

The DNA sequence is represented as a string of characters 'A', 'C', 'G', and 'T'. Two integers, k and d, are separated by a space. The integer k refers to the length of the substrings (k-mers) of interest, and d specifies the maximum number of mismatches that are allowed in a k-mer.

```
We defined a mismatch in ``Compute the Hamming Distance Between Two
    StringsâĂİ. We now generalize âĂFind the Most Frequent Words in a
    StringâĂİ to incorporate mismatches as well. Given strings Text and
    Pattern as well as an integer d, we define Countd(Text, Pattern) as
    the total number of occurrences of Pattern in Text with at most d
    mismatches. For example, Count1(AACAAGCTGATAAACATTTAAAGAG, AAAAA) = 4
     because AAAAA appears four times in this string with at most one
    mismatch: AACAA, ATAAA, AAACA, and AAAGA. Note that two of these
    occurrences overlap.

A most frequent k-mer with up to d mismatches in Text is simply a string
    Pattern maximizing Countd(Text, Pattern) among all k-mers. Note that
    Pattern does not need to actually appear as a substring of Text; for
    example, AAAAA is the most frequent 5-mer with 1 mismatch in
    AACAAGCTGATAAACATTTAAAGAG, even though AAAAA does not appear exactly
    in this string. Keep this in mind while solving the following problem
    .

Frequent Words with Mismatches Problem

Find the most frequent k-mers with mismatches in a string. Given: A
    string Text as well as integers k and d. Return: All most frequent k-
    mers with up to d mismatches in Text.

Sample Dataset
ACGTTGCATGTCGCATGATGCATGAGAGCT
4 1

Sample Output
GATG ATGC ATGT

write the solve() function to solve the above problem
Do NOT print anything
Do NOT make comments
Do NOT call the main() function.
Use 4 spaces for tabs.
input_str: a string
return output: another string

def main():
    with open("input.txt", "r") as f:
        output = solve(f.read())
    print(output)

Write your solution here
Begin with: def solve(input_str):
```

Here is the corresponding reference code:

```
# Find the Most Frequent Words with Mismatches in a String

from .ba1g import hamming
```

```
4 from .ba1b import count_kmers, most_frequent
5 from itertools import product
6
7 # Note, the best kmer might not be observed in our sequence. The
      simplistic
8 # method here simply checks all possible kmers (which is ~17M for k = 12)
9
10 def generate_kmers(k):
11     return ("".join(x) for x in product(["A", "C", "G", "T"], repeat=k))
12
13 def count_hamming_kmers(kmers, d, k):
14     for x in generate_kmers(k):
15         count = sum(kmers[y] for y in kmers if hamming(x, y) <= d)
16         if count > 0:
17             yield [x, count]
18
19 def main(file):
20     seq, ints = open(file).read().splitlines()
21     k, d = list(map(int, ints.split()))
22     kmers = count_kmers(seq, k)
23     hkmers = dict(count_hamming_kmers(kmers, d, k))
24     print(*most_frequent(hkmers))
```

# E ADDITIONAL MODEL INFORMATION

| Model | Parameters |
|---|---|
| InCoder (Fried et al., 2023) | *6B* |
| SantaCoder (Allal et al., 2023) | *1.1B* |
| StarCoder (Li et al., 2023) | *15.5B* |
| StarCoderPlus (Li et al., 2023) | *15.5B* |
| InstructCodeT5+ (Wang et al., 2023a) | *16B* |
| CodeGen (Nijkamp et al., 2023b) | *6B-mono\** |
| CodeGen (Nijkamp et al., 2023b) | *16B-mono\** |
| CodeGen2 (Nijkamp et al., 2023a) | *7B\** |
| GPT-3.5-Turbo | *Unknown* |
| GPT-4 | *Unknown* |

Table 8: Size of models.

We were not able to run the 16B models due to issues with scaling and sharding on the A100s.

We found that the following parameters produced the best results across all models.

- top_k=50
- top_p=0.95
- temperature=0.7
- early_stopping=True
- num_return_sequences=1
- do_sample=True

We utilized similar parameters to make model testing consistent across all tested models. This approach allowed for a more unbiased comparison of how each model performed on our benchmark. We determined these weights by averaging the recommended parameters for the above models and then used Bayesian optimization to determine the most optimal parameters.

Furthermore, note that we used the version of GPT-3.5-Turbo hosted on Azure OpenAI Service (gpt-35-turbo-0301). There may be some minor differences compared with the OpenAI Platform version.

## F  UTILIZING CHATGPT FOR PROMPT GENERATION

ChatGPT served as a pivotal component in our prompt generation, enabling us to assemble comprehensive annotations for the ground-truth functions and their dependencies. This eliminates the need for extensive manual labor that would otherwise be spent on annotation. Utilizing GPT-3.5 Turbo calls further ensures consistency across annotations, mitigating the variability that might arise from human annotators.

The prompt was constructed as follows:

```
1 Generate a prompt for the following function, such that a programmer
     could reproduce it based solely on the description. Emphasize crucial
      components like the return statement, complex function calls, and
     ambiguous variables. However, avoid line-by-line descriptions; aim to
      provide a high-level overview. The response should be in plain text,
      free from any formatting such as Markdown. Keep the length under 200
      words or 15 lines, whichever is shorter.
2 Here is the function for you to describe:
3 <function>
4 Here are the additional dependencies that may or may not be used within
     the function:
5 <dependencies>
6 Again, limit your responses to no more than 200 words or 15 lines,
     whichever comes first.
7 Start your prompt with "write a function".
```

## G  MODEL CONTEXT LIMITS

| Model | Context limit |
|---|---|
| InCoder (Fried et al., 2023) | *2048* |
| SantaCoder (Allal et al., 2023) | *2048* |
| StarCoder (Li et al., 2023) | *8192* |
| StarCoderPlus (Li et al., 2023) | *8192* |
| InstructCodeT5+ (Wang et al., 2023a) | *2048* |
| CodeGen (Nijkamp et al., 2023b) | *2048* |
| CodeGen2 (Nijkamp et al., 2023a) | *2048* |
| GPT-3.5-Turbo | *8192* |
| GPT-4 | *8192* |

Table 9: Length limits of different models.

Since the limit includes both input and output tokens, we reserved 256 tokens for the output, therefore reducing the input length limit by 256. In our best effort to continue testing with these context limits, instead of treating a failing test case immediately, we attempted to truncate the prompts such that they fit barely below the context limit of each respective model. For all models, we also tested a version with the function summary at the top. Note that for all models, the context limit was identical regardless of the parameter count.

# H CODEREVAL COMPARISON

To validate the performance of code LLMs, multiple existing benchmarks are proposed, including only cases of generating a standalone function, i.e., a function that invokes or accesses only built-in functions and standard libraries. However, standalone functions constitute only about 30% of the functions from real open-source projects. To assess a model's performance for pragmatic code generation (i.e., code generation for real settings of open source or proprietary code), CoderEval (Yu et al., 2023) proposes a benchmark named CoderEval of pragmatic code generation with generative pre-trained models.

In comparison to CoderEval, our approach is more focused on bioinformatics, as we ensure each function requires a certain level of domain knowledge in bioinformatics, as opposed to entry-level data science or software engineering tasks.

Moreover, we employ a more advanced parser, implementing a comprehensive parsing of the respective GitHub repositories. This includes the imported third-party packages and the classes that have been imported from another actual file.

We also conduct function testing at the file level, rather than the repository level, manually extracting the context at the end and running it through a framework that catches more errors than CoderEval.

Additionally, our dataset is larger than CoderEval's, which only consists of 230 functions from 43 Python projects and 230 methods from 10 Java projects, while we have data from more than two thousand sources.

CoderEval classifies tasks into six levels according to the source of dependency outside the function, while we directly use the most complex dependencies.

CoderEval employs a human-labeled version description from 13 experienced engineers, while we leverage ChatGPT to generate function descriptions. Similarly, for test cases, our use of fuzz testing allows us to scale to large data volumes without the need for intensive manual annotation.

Yet, we share some similarities with CoderEval. Both BIOCODER and CoderEval can be used to assess the performance of models beyond merely generating standalone functions. Given the need to address dependencies with context, we both establish testing based on Docker, cloning GitHub repositories and their associated import dependencies. With 'k' candidate outputs generated by different models (e.g., 20), we simulate scenarios in actual IDEs.

## I PROMPT EXPLANATIONS

Below is an explanation of the prompt types:

1. **Summary Only** We begin the prompt from these prompts, where only the summary and the function signature are contained in the prompt. The summary is added as a comment directly above the signature, which is uncommented. The summary includes nearly complete details about the task; however, it intentionally does not thoroughly explain what the context is. Therefore, this result is best treated as a baseline when compared with other prompt types. An example is shown below:

```
1  #This is in python
2  #Write a function named "planewise_morphology_closing" that
       accepts two parameters, "x_data" and "structuring_element".
       Inside the function, create a new variable named "y_data" that
       will hold the result of the morphological closing operation.
       Loop through each item in "x_data" and perform the same
       operation on each. Do this by using the "enumerate" function to
        get both the index and plane for each item. Use the "closing"
       function from the "skimage.morphology" module to perform the
       morphological closing operation on each plane using the "
       structuring_element" parameter. Finally, return the newly
       created "y_data" array that holds the result of the operation.
       Ensure that the function uses the "numpy" and "skimage.
       morphology" modules, as they are dependencies for the function.
3  #
4  #def planewise_morphology_closing(x_data, structuring_element):
5
6
```

2. **Uncommented** We add the summary back into all of the parsed context. Therefore, the entire prompt contains all imports, global variables, classes, internal class functions, etc., that were imported into the file originally. For functions over ten lines, we summarize the parameters, return type, and logic instead of including the actual function code in order to reduce the number of tokens used and eliminate data that the model would not need. Note that, by design, if these extraneous functions were used, the test would have counted as failed, because it would be extremely inefficient code. An example is shown below:

```
1  This is in python
2  Here are the imports:
3  from cellprofiler.library.modules import closing
4  from ._help import HELP_FOR_STREL
5  from cellprofiler_core.module import ImageProcessing
6  from cellprofiler_core.setting import StructuringElement
7  Here are the global variables:
8  Here are the class declarations:
9  class Closing(ImageProcessing):
10     attributes: self.structuring_element,self.function
11     methods:
12     def create_settings(self):
13         summary: creates settings for the Closing function
14         param: self (unknown) - instance of the Closing function
15         return: None
16     def settings(self):
17         summary: Returns settings from superclass with self.
       structuring_element added.
18         param: None
19         return: list - __settings__ with self.structuring_element
       appended.
20     def visible_settings(self):
21         summary: This function returns the visible settings of a
       Closing object.
22         param: self (Closing) - the Closing object whose settings
       are to be returned.
```

```
23        return: list - the visible settings of the Closing object.
24    def run(self, workspace):
25        summary: Applies morphological closing to an image in a
      workspace.
26        param: workspace (object) - the workspace containing the
      image.
27        return: None
28 Here are the additional function declarations:
29 def planewise_morphology_closing(x_data, structuring_element):
30    summary: Applies morphology closing operation to each plane in
       x_data using structuring_element and returns the resulting
      y_data.
31    param: x_data (numpy array) - 3D array containing the input
      data.
32    param: structuring_element (numpy array) - Structuring element
       used to perform the closing operation.
33    return: y_data (numpy array) - 3D array containing the result
      of closure operation on each plane in x_data.
34 Here are the comments and the specs:
35 Write a function named "planewise_morphology_closing" that accepts
       two parameters, "x_data" and "structuring_element". Inside the
       function, create a new variable named "y_data" that will hold
      the result of the morphological closing operation. Loop through
       each item in "x_data" and perform the same operation on each.
      Do this by using the "enumerate" function to get both the index
       and plane for each item. Use the "closing" function from the "
      skimage.morphology" module to perform the morphological closing
       operation on each plane using the "structuring_element"
      parameter. Finally, return the newly created "y_data" array
      that holds the result of the operation. Ensure that the
      function uses the "numpy" and "skimage.morphology" modules, as
      they are dependencies for the function.
36 def planewise_morphology_closing(x_data, structuring_element):
37
38
```

Note that it is nearly certain that each prompt will contain extraneous data that does not need to be used for the function. The goal is for the LLM to ensure it uses the correct context for the function. Note that for prompts that exceed the model's context limit, this prompt likely fails.

3. **Summary at Bottom** We generate these using the previous prompt (uncommented), and comment out all the context. Note that for prompts that exceed the model's context limit, this prompt likely fails. In addition, note that there are no results for "summary at bottom" for Java due to an incompatibility with Java syntax. We could not generate this type of prompt for Java similar to how we generated it for Python.

```
1 #Here are the imports:
2 #from cellprofiler.library.modules import closing
3 #from ._help import HELP_FOR_STREL
4 #from cellprofiler_core.module import ImageProcessing
5 #from cellprofiler_core.setting import StructuringElement
6 #Here are the global variables:
7 #Here are the class declarations:
8 #class Closing(ImageProcessing):
9 #    attributes: self.structuring_element,self.function
10 #    methods:
11 #    def create_settings(self):
12 #        summary: creates settings for the Closing function
13 #        param: self (unknown) - instance of the Closing function
14 #        return: None
15 #    def settings(self):
16 #        summary: Returns settings from superclass with self.
      structuring_element added.
17 #        param: None
```

```
18 #        return: list - __settings__ with self.structuring_element
        appended.
19 #    def visible_settings(self):
20 #        summary: This function returns the visible settings of a
        Closing object.
21 #        param: self (Closing) - the Closing object whose settings
        are to be returned.
22 #        return: list - the visible settings of the Closing object.
23 #    def run(self, workspace):
24 #        summary: Applies morphological closing to an image in a
        workspace.
25 #        param: workspace (object) - the workspace containing the
        image.
26 #        return: None
27 #Here are the additional function declarations:
28 #def planewise_morphology_closing(x_data, structuring_element):
29 #    summary: Applies morphology closing operation to each plane in
         x_data using structuring_element and returns the resulting
        y_data.
30 #    param: x_data (numpy array) - 3D array containing the input
        data.
31 #    param: structuring_element (numpy array) - Structuring element
         used to perform the closing operation.
32 #    return: y_data (numpy array) - 3D array containing the result
        of closure operation on each plane in x_data.
33 #Here are the comments and the specs:
34 #Write a function named "planewise_morphology_closing" that
        accepts two parameters, "x_data" and "structuring_element".
        Inside the function, create a new variable named "y_data" that
        will hold the result of the morphological closing operation.
        Loop through each item in "x_data" and perform the same
        operation on each. Do this by using the "enumerate" function to
         get both the index and plane for each item. Use the "closing"
        function from the "skimage.morphology" module to perform the
        morphological closing operation on each plane using the "
        structuring_element" parameter. Finally, return the newly
        created "y_data" array that holds the result of the operation.
        Ensure that the function uses the "numpy" and "skimage.
        morphology" modules, as they are dependencies for the function.
35 def planewise_morphology_closing(x_data, structuring_element):
36
```

4. **Summary at Top** These prompts are generated from the previous prompts (summary at
   bottom); however, the summary is copied at the top (moved for Java). This is intended for
   models with shorter context lengths, as when we truncated the prompt, the summary would
   still be intact, along with a portion of the context.

```
1 #This is in python
2 #Write a function named "planewise_morphology_closing" that
        accepts two parameters, "x_data" and "structuring_element".
        Inside the function, create a new variable named "y_data" that
        will hold the result of the morphological closing operation.
        Loop through each item in "x_data" and perform the same
        operation on each. Do this by using the "enumerate" function to
         get both the index and plane for each item. Use the "closing"
        function from the "skimage.morphology" module to perform the
        morphological closing operation on each plane using the "
        structuring_element" parameter. Finally, return the newly
        created "y_data" array that holds the result of the operation.
        Ensure that the function uses the "numpy" and "skimage.
        morphology" modules, as they are dependencies for the function.
3 #
4 #def planewise_morphology_closing(x_data, structuring_element):
5 #
6 #Here are the imports:
```

```
7  #from cellprofiler.library.modules import closing
8  #from ._help import HELP_FOR_STREL
9  #from cellprofiler_core.module import ImageProcessing
10 #from cellprofiler_core.setting import StructuringElement
11 #Here are the global variables:
12 #Here are the class declarations:
13 #class Closing(ImageProcessing):
14 #    attributes: self.structuring_element,self.function
15 #    methods:
16 #    def create_settings(self):
17 #        summary: creates settings for the Closing function
18 #        param: self (unknown) - instance of the Closing function
19 #        return: None
20 #    def settings(self):
21 #        summary: Returns settings from superclass with self.
       structuring_element added.
22 #        param: None
23 #        return: list - __settings__ with self.structuring_element
       appended.
24 #    def visible_settings(self):
25 #        summary: This function returns the visible settings of a
       Closing object.
26 #        param: self (Closing) - the Closing object whose settings
       are to be returned.
27 #        return: list - the visible settings of the Closing object.
28 #    def run(self, workspace):
29 #        summary: Applies morphological closing to an image in a
       workspace.
30 #        param: workspace (object) - the workspace containing the
       image.
31 #        return: None
32 #Here are the additional function declarations:
33 #def planewise_morphology_closing(x_data, structuring_element):
34 #    summary: Applies morphology closing operation to each plane in
        x_data using structuring_element and returns the resulting
       y_data.
35 #    param: x_data (numpy array) - 3D array containing the input
       data.
36 #    param: structuring_element (numpy array) - Structuring element
        used to perform the closing operation.
37 #    return: y_data (numpy array) - 3D array containing the result
       of closure operation on each plane in x_data.
38 #Here are the comments and the specs:
39 #Write a function named "planewise_morphology_closing" that
       accepts two parameters, "x_data" and "structuring_element".
       Inside the function, create a new variable named "y_data" that
       will hold the result of the morphological closing operation.
       Loop through each item in "x_data" and perform the same
       operation on each. Do this by using the "enumerate" function to
        get both the index and plane for each item. Use the "closing"
       function from the "skimage.morphology" module to perform the
       morphological closing operation on each plane using the "
       structuring_element" parameter. Finally, return the newly
       created "y_data" array that holds the result of the operation.
       Ensure that the function uses the "numpy" and "skimage.
       morphology" modules, as they are dependencies for the function.
40 #def planewise_morphology_closing(x_data, structuring_element):
41
```

5. **Necessary Only** We use a mixture of our syntax solving algorithm and hand annotation to determine exactly which objects of the code are necessary for the function to execute, and then use them in place of the original context. Note that this is very similar to the environment used for testing the functions.

```
1 Write a function with the following specs:
```

```
2  --specs begin here--
3  #Write a function named "planewise_morphology_closing" that
       accepts two parameters, "x_data" and "structuring_element".
       Inside the function, create a new variable named "y_data" that
       will hold the result of the morphological closing operation.
       Loop through each item in "x_data" and perform the same
       operation on each. Do this by using the "enumerate" function to
        get both the index and plane for each item. Use the "closing"
       function from the "skimage.morphology" module to perform the
       morphological closing operation on each plane using the "
       structuring_element" parameter. Finally, return the newly
       created "y_data" array that holds the result of the operation.
       Ensure that the function uses the "numpy" and "skimage.
       morphology" modules, as they are dependencies for the function.
4  param: x_data (numpy array) - 3D array containing the input data.
5  param: structuring_element (numpy array) - Structuring element
       used to perform the closing operation.
6  return: y_data (numpy array) - 3D array containing the result of
       closure operation on each plane in x_data.
7  --specs end here--
8  Note the function will be embedded in the following context
9  --context begins here--
10 import numpy
11 import skimage.morphology
12 import os
13 <<insert solution here>>
14 def main():
15     numpy.random.seed(<|int;range=0,100|>)
16     pixels = numpy.random.randint(2, size=(10, 10, 10))
17     structuring_element = skimage.morphology.square(3)
18     print(planewise_morphology_closing(pixels, structuring_element
       ))
19 if __name__ == "__main__":
20     main()
21 --context ends here--
22 Make sure to only generate the function and not any of the context
       . Make sure you are generating valid, runnable code. Begin your
        solution with:
23 def planewise_morphology_closing(x_data, structuring_element):
24
```

## J  MODEL ABLATION STUDY

For the ablation study, we considered two representative functions, one for Java and one for Python, and determined how the various models performed in generating the following functions. The Java function we selected is the following:

```
1  public static String getReverseComplementedSequence(SAMRecord record, int
        startIndex, int length) {
2      if (startIndex < 0) {
3          startIndex = record.getReadLength() + startIndex;
4      }
5      byte[] rangeBytes = Arrays.copyOfRange(record.getReadBases(),
        startIndex, startIndex + length);
6      SequenceUtil.reverseComplement(rangeBytes);
7      return new String(rangeBytes);
8  }
```

and the (abridged) context that the function getReverseComplementedSequence had access to is the following:

```
1  import org.apache.commons.math3.distribution.HypergeometricDistribution;
2  import java.text.DecimalFormat;
3  import java.util.*;
4  import java.util.function.Function;
5  import htsjdk.samtools.util.SequenceUtil;
6  import java.util.Objects;
7
8  class SAMRecord {
9      public byte[] arr2;
10
11     public SAMRecord(byte[] arr) {
12         arr2 = arr;
13     }
14
15     public byte[] getReadBases() {
16         return arr2;
17     }
18
19     public int getReadLength() {
20         return arr2.length;
21     }
22 }
23
24 final class Utils {
25     /**
26      * Method returns reverse complemented sequence for the part of the
        record. Can work with 3' and 5' ends
27      * (if start index < 0, then it will found the index in the end of
        sequence by adding the length of record).
28      * @param record read from SAM file to process
29      * @param startIndex index where start the sequence
30      * @param length length of pert of sequence
31      * @return reverse complemented part of record
32      */
33     <<insert solution here>>
34
35     public static String reverse(String string) {
36         return new StringBuffer(string).reverse().toString();
37     }
38
39     public static String complement(String string) {
40         final byte[] bases = htsjdk.samtools.util.StringUtil.
        stringToBytes(string);
41         complement(bases);
42         return htsjdk.samtools.util.StringUtil.bytesToString(bases);
```

```
43        }
44 }
```

We chose this as the representative Java function because it is highly bioinformatics related and the solution will require an understanding of the context around it, namely the custom SAMRecord class. Furthermore, the model will have to understand that there are many helpful utility functions available in the Utils class that can be used to help complete the function, such as the reverse and complement functions.

The Python function that we selected for this ablation study is the following:

```
1 def absolute_clonal(cnarr, ploidy, purity, is_reference_male,
     is_sample_female
2     ):
3     """Calculate absolute copy number values from segment or bin log2
     ratios."""
4     absolutes = np.zeros(len(cnarr), dtype=np.float_)
5     for i, row in enumerate(cnarr):
6         ref_copies, expect_copies = _reference_expect_copies(row.
     chromosome,
7             ploidy, is_sample_female, is_reference_male)
8         absolutes[i] = _log2_ratio_to_absolute(row.log2, ref_copies,
9             expect_copies, purity)
10    return absolutes
```

and the (abridged) context that the function absolute_clonal had access to is the following:

```
1 import random
2 import hashlib
3 import numpy as np
4 import os
5
6 def _reference_expect_copies(chrom, ploidy, is_sample_female,
     is_reference_male
7     ):
8     """Determine the number copies of a chromosome expected and in
     reference.
9     For sex chromosomes, these values may not be the same ploidy as the
10    autosomes. The "reference" number is the chromosome's ploidy in the
11    CNVkit reference, while "expect" is the chromosome's neutral ploidy
     in the
12    given sample, based on the specified sex of each. E.g., given a
     female
13    sample and a male reference, on chromosome X the "reference" value is
      1 but
14    "expect" is 2.
15    Returns
16    -------
17    tuple
18        A pair of integers: number of copies in the reference, and
     expected in
19        the sample.
20    """
21    chrom = chrom.lower()
22    if chrom in ['chrx', 'x']:
23        ref_copies = ploidy // 2 if is_reference_male else ploidy
24        exp_copies = ploidy if is_sample_female else ploidy // 2
25    elif chrom in ['chry', 'y']:
26        ref_copies = ploidy // 2
27        exp_copies = 0 if is_sample_female else ploidy // 2
28    else:
29        ref_copies = exp_copies = ploidy
30    return ref_copies, exp_copies
31
```

```
32 def _log2_ratio_to_absolute_pure(log2_ratio, ref_copies):
33     """Transform a log2 ratio to absolute linear scale (for a pure sample
       ).
34     Purity adjustment is skipped. This is appropriate if the sample is
       germline
35     or if scaling for tumor heterogeneity was done beforehand.
36     .. math :: n = r*2^v
37     """
38     ncopies = ref_copies * 2 ** log2_ratio
39     return ncopies
40
41 def _log2_ratio_to_absolute(log2_ratio, ref_copies, expect_copies, purity
       =None
42     ):
43     """Transform a log2 ratio to absolute linear scale (for an impure
       sample).
44     Does not round to an integer absolute value here.
45     Math::
46         log2_ratio = log2(ncopies / ploidy)
47         2^log2_ratio = ncopies / ploidy
48         ncopies = ploidy * 2^log2_ratio
49     With rescaling for purity::
50         let v = log2 ratio value, p = tumor purity,
51             r = reference ploidy, x = expected ploidy,
52             n = tumor ploidy ("ncopies" above);
53         v = log_2(p*n/r + (1-p)*x/r)
54         2^v = p*n/r + (1-p)*x/r
55         n*p/r = 2^v - (1-p)*x/r
56         n = (r*2^v - x*(1-p)) / p
57     If purity adjustment is skipped (p=1; e.g. if germline or if scaling
       for
58     heterogeneity was done beforehand)::
59         n = r*2^v
60     """
61     if purity and purity < 1.0:
62         ncopies = (ref_copies * 2 ** log2_ratio - expect_copies * (1 -
       purity)
63             ) / purity
64     else:
65         ncopies = _log2_ratio_to_absolute_pure(log2_ratio, ref_copies)
66     return ncopies
67
68 <<insert solution here>>
```

We chose this as the representative Python function because, like the Java function that we chose, it is both highly bioinformatics related and the solution will require an understanding of the context around it, namely the custom _reference_expect_copies and _log2_ratio_to_absolute functions, as well as their problem, will require the use of some of the general imports such as the numpy package.

1. **InCoder**: Similar to many other models, InCoder has very poor Java performance, due to the fact that it was trained and fine-tuned on a Python corpus. Therefore, InCoder achieves 0% Pass@K for the Java prompts. The main issue that InCoder runs into is that for Java, it seems to struggle to understand the objective in the prompt and fails to generate the actual function; instead, it seems to regurgitate the prompt, which will obviously result in a fail. For instance here is a sample InCoder output for the function getReverseComplementedSequence:

```
1           # Do not write any comments in your code.
2 #This is in java
3 #Write a function called "getReverseComplementedSequence" that
      takes in a SAMRecord object, an integer called "startIndex",
      and an integer called "length". The function should check if
      startIndex is less than 0, and if it is, set startIndex to the
      SAMRecord object's read length plus startIndex. The function
      should then create a byte array called "rangeBytes" that is a
```

```
          copy of a portion of the SAMRecord object's read bases starting
           from the startIndex and ending at startIndex + length. The
          function should then call SequenceUtil.reverseComplement on
          rangeBytes, and finally return a new String created from the
          rangeBytes byte array.
  4  #Here are the imports:
  5  #import org.apache.commons.math3.distribution.
          HypergeometricDistribution;
  6  #import java.text.DecimalFormat;
  7  #import java.util.*;
  8  #import java.util.function.Function;
  9  #import htsjdk.samtools.util.SequenceUtil;
 10  #import java.util.Objects;
 11  #Here are the class declarations:
 12  #class SAMRecord {
 13  #      public byte[] arr2;
 14  #      public SAMRecord(byte[] arr) {
 15  #          arr2 = arr;
 16  #      }
 17  #      public byte[] getReadBases() {
 18  #          return arr2;
 19  #      }
 20  #      public int getReadLength() {
 21  #          return arr2.length;
 22  #      }
 23  #}
 24  #class Region {
 25  #      /**
 26  #       * Chromosome name
 27  #       */
 28  #      public final String chr;
 29  #      /**
 30  #       * Region start position
 31  #       */
 32  #      public final int start;
 33  #      /**
 34  #       * Region end position
 35  #       */
 36
```

Although InCoder performs comparatively better on Python, it still struggles greatly to understand the objective of the prompt and actually generate the function. Instead, for Python code, the InCoder models seem to simply regurgitate the summary of the Python function that can be found in the prompt. For instance, here is a sample InCoder output for the function `absolute_clonal`:

```
  1          #   summary: Returns the number of copies of a chromosome
          in reference and in sample.
  2  #    param: chrom (string) - the chromosome to be evaluated
  3  #    param: ploidy (int) - the ploidy of the chromosome
  4  #    param: is_reference_male (bool) - indicates if the reference
          is male or not
  5  #    return: tuple - a pair of integers indicating the number of
          copies in the reference and in the sample.
  6  #def _log2_ratio_to_absolute(cnarr, ref_copies, expect_copies,
          purity):
  7  #    summary: Converts log2 ratio values into absolute copy number
          values.
  8  #    param: cnarr (array-like) - Copy number segments or bins
  9  #    param: ref_copies (array-like) - An array of integers
          indicating the number of copies in the reference for each
          segment or bin.
 10  #    param: expect_copies (array-like) - An array of integers
          indicating the number of copies in the expected for each
          segment or bin.
```

```
11  #     param: purity (float) - estimated tumor pur
12
```

Overall, it seems that InCoder struggles to understand longer and more complex prompts, and t struggles to understand the objective of the prompts. Often it simply defaults to regurgitating parts of the prompt that it has encountered, without generating the desired function.

2. **SantaCoder**:

   Similar to many other models, SantaCoder also has poor Java performance, due to the fact that it was trained and fine-tuned on a Python corpus. Therefore, SantaCoder achieves 0% Pass@K for the Java prompts. The main issue that SantaCoder runs into is that Java seems to struggle to understand the objective in the prompt and fails to generate the full function, only generating parts of the function. This may also be due to the smaller context limit of SantaCoder, resulting in it being unable to fully generate the functions. For instance, here is a sample SantaCoder output for the function `getReverseComplementSequence`:

```
1           #     double x = perc;
2  #     int a = (int) x;
3  #     int b = (int) (x - a);
4  #     int c = (int) (b - a);
5  #     if(c==0){
6  #         return 0;
7  #     }else if(c==1){
8  #         return 1;
9  #     }else{
10 #         return 2;
11 #     }
12 #     //return getColor(value, perc);
13
14 #     int c1 = (int) ((perc - a)/b);
15 #     int c2 = (int) ((perc - a - b)/a);
16 #     return 1 + c1 * 3 + c2 * 5;
17
18 #     int a1 = (int) (c1 / 2);
19 #     int a2 = (int) ((c1 - a1) / 2);
20 #     int a3 = (int) (c2 / 2);
21 #     int a4 = (int) ((c2 - a3) / 2);
22 #     int a5 = (int) (c1 +
23
```

   Although SantaCoder performs comparatively better on Python, especially considering its small model size, it still struggles greatly to generate many of the functions. Often due to the small context size of the SantaCoder model, thus for more complicated functions, SantaCoder is unable to generate the model. Overall, it seems that SantaCoder struggles to understand the longer and more complex prompts, and it often struggles to understand the objective of the prompts. Furthermore, the small context limit of SantaCoder limits its ability to generate more complex and more context-rich code.

3. **StarCoder**:

   Similar to many of the other models, StarCoder has very poor Java performance, due to the fact that it was trained and finetuned on a Java corpus. Therefore, StarCoder achieves 0% Pass@K for the Java prompts. However, unlike some of the other models, it seems to understand the objective of the prompt and uses the context appropriately. For instance, using the `reverseComplement` helper function correctly when generating the `getReverseComplementedSequence` function, as can be seen below:

```
1           def getReverseComplementedSequence(samRecord, startIndex,
       length):
2       if startIndex < 0:
3           startIndex = samRecord.getReadLength() + startIndex
4       rangeBytes = samRecord.getReadBases().getBytes()[startIndex :
       startIndex + length]
5       reverseComplemented = SequenceUtil.reverseComplement(
       rangeBytes)
```

```
6        return reverseComplemented.toString()
7
```

However, StarCoder, being finetuned on Python, seems to have a hard time generating proper Java syntax, and instead generates a Python function. Although StarCoder performs comparatively better on Python, it still struggles greatly to generate many of the functions. Python seems to understand the objective in the prompt and generates proper syntax, however, some of its logic is incorrect, resulting in some failed tests. For instance, here is a sample StarCoder output for the function `absolute_clonal`:

```
1        #Write your function here!
2  def absolute_clonal(cnarr, ploidy, purity, is_reference_male,
       is_sample_female):
3      absolutes = np.zeros(len(cnarr), dtype=np.float_)
4      for i, row in enumerate(cnarr):
5          chromosome, log2_value = row
6          ref_copies, expect_copies = _reference_expect_copies(
       chromosome, ploidy, is_sample_female, is_reference_male)
7          absolute_value = _log2_ratio_to_absolute(log2_value,
       ref_copies, expect_copies, purity)
8          absolutes[i] = absolute_value
9      return absolutes
10
```

4. **StarCoder+**: Unlike many of the other models, StarCoder+ has nonzero Pass@K performance on Java. This is likely because StarCoder+ is trained and finetuned on a larger corpus that consists of more languages than just Python. It seems to understand the objective of the prompt and uses the context appropriately, for instance using the `reverseComplement` helper function correctly when generating the `getReverseComplementedSequence` function, as can be seen below:

```
1        import htsjdk.samtools.util.SequenceUtil;
2  public class getReverseComplementedSequence
3  {
4  public static String getReverseComplementedSequence(SAMRecord
       samRecord, int startIndex, int length)
5  {
6  if(startIndex < 0)
7  {
8  startIndex = samRecord.getReadLength() + startIndex;
9  }
10  byte[] rangeBytes = new byte[length];
11  samRecord.getReadBases(rangeBytes, startIndex, length);
12  byte[] rcBytes = SequenceUtil.reverseComplement(rangeBytes);
13  return new String(rcBytes);
14  }
15  }
16
```

Unlike StarCoder, it seems to understand and be able to generate Java syntax.

5. **CodeGen**: The performance of CodeGen1 was not great. This could be due to a variety of factors, including the fact that CodeGen models are not intended for full multi-line function generation, or the fact that the dataset used to train CodeGen models does not contain much Java code. This is demonstrated in some of the generations, as CodeGen ended up generating Python code, despite us mentioning that this prompt is in Java, and using Java code in the context. Sometimes, it also generates the comment again, such as below:

```
1  #Note: This is not a string method.
2
```

6. **CodeGen2**: CodeGen2 outperformed CodeGen as expected in our normal test cases; however, it was unexpectedly slightly outperformed in the Rosalind benchmark.

7. **GPT-3.5**: GPT clearly performs the best out of all models tested, on both Java and Python and across all prompt types. The version we used on Azure had an 8k context size, so it had

an advantage over some of the models due to being able to understand more of the context. Furthermore, the fact that some of the function descriptions were generated by GPT might mean that it had an easier time generating code from its own description. Furthermore, given the size of the model, it is possible that some of the functions were memorized, although we tried to combat this by changing the function signature slightly in some cases. Here is an example of the GPT output:

```
def FindLocalPeaks(signal):
    peakLoc = []
    maxSuspect = None
    minSuspect = None
    for i in range(1, len(signal)-1):
        if signal[i] > 0:
            if signal[i] > signal[i-1] and signal[i] == signal[i
    +1]:
                maxSuspect = i
            elif signal[i] == signal[i-1] and signal[i] > signal[i
    +1]:
                if maxSuspect is not None:
                    peakLoc.append(maxSuspect)
                    maxSuspect = None
            elif signal[i] > signal[i-1] and signal[i] > signal[i
    +1]:
                peakLoc.append(i)
        elif signal[i] < 0:
            if signal[i] < signal[i-1] and signal[i] == signal[i
    +1]:
                minSuspect = i
            elif signal[i] == signal[i-1] and signal[i] < signal[i
    +1]:
                if minSuspect is not None:
                    peakLoc.append(minSuspect)
                    minSuspect = None
            elif signal[i] < signal[i-1] and signal[i] < signal[i
    +1]:
                peakLoc.append(i)
    return np.array(peakLoc, dtype=int)

```

It is also important to note that in every prompt, GPT-3.5 generated usable, cleaner code without comments as directed. Therefore, most test cases were run without syntax errors, but mostly logic errors.

Next, we performed a horizontal analysis on a single prompt to compare the outputs of each of the models with each other in order to understand the differences between the models and gain insight into what parts of the prompts the various models seem to understand. We applied the following problem for the horizontal analysis:

```
@property
def outlines(self):
    """Get a mask of all the points on the border of objects"""
    if self._outlines is None:
        for i, labels in enumerate(self.labels):
            if i == 0:
                self._outlines = centrosome.outline.outline(labels) != 0
            else:
                self._outlines |= centrosome.outline.outline(labels) != 0
        if self.line_width is not None and self.line_width > 1:
            hw = float(self.line_width) / 2
            d = scipy.ndimage.distance_transform_edt(~self._outlines)
            dti, dtj = numpy.where((d < hw + 0.5) & ~self._outlines)
            self._outlines = self._outlines.astype(numpy.float32)
            self._outlines[dti, dtj] = numpy.minimum(1, hw + 0.5 - d[dti,
    dtj])
    return self._outlines.astype(numpy.float32)
```

The reasons we chose this function for the horizontal analysis are the following:

1. It is a Python function, and all of the model's generally seem to be fine-tuned on Python code and thus will perform better on Python code; therefore, it would be a more fair comparison to use a Python function for this analysis.

2. Furthermore, this is also a relatively tricky function to generate as it has many moving parts and relies on many different levels of understanding of the context. First, it is important to note that this has a @property decorator and is a function that is part of a class; thus, the models must understand this and generate the @property decorator accordingly. Furthermore, the model must understand and correctly use the self._outlines, self._line_width, and self.labels instance variables. Finally, the model must understand and use correctly the highly bioinformatics-specific external library that is centrosome.

The prompt that we used for the horizontal analysis is the following "Summary Only" prompt:

```
1  #This is in python
2  #Write a function called "get_border_mask" that takes in a class object
       as an argument and returns a binary mask of all the points on the
       border of objects. The function should check if the instance variable
        "_outlines" is None and if it is, then use the labels attribute of
       the class object to generate the outlines mask using "centrosome.
       outline.outline(labels)" method. If the "_outlines" variable is not
       None, then it should iterate over "centrosome.outline.outline(labels)
       " for each i, and update the "_outlines" variable using the bitwise
       OR operator with the new mask. If the line_width attribute of the
       class object is not None and greater than 1, the function should
       calculate the distance transform using the "~self._outlines" mask and
        compute the binary mask for borders with the given line width (hw)
       using numpy.where(). Finally, the function should return the mask as
       a float32 numpy array. Make sure to import the necessary dependencies
        including "centrosome.outline" and "numpy" and also initialize the "
       _outlines" variable in the class.
3  #
4  #The function is located in the class OutlinesMixin
5  #
6  #@property
7  #def get_border_mask(self):
```

We chose the "Summary Only" format of the prompt because generally, the models performed best when the prompt was written in the "Summary Only" format. The outputs of the various models for this prompt are as follows:

1. **InCoder**:

```
1  def get_border_mask(self):
2      #If outlines is none, generate outlines using centrosome.
          outline.outline(labels) method and set outlines to the
          generated mask.
3      if self.outlines is None:
4          self.outlines = self.outline.outline(labels=self.labels)
5
6      # Get the mask using numpy
7      border_mask = np.where(self.outlines > 0, 1, 0 )
8
9      # Check if line width is greater than 1
10     if self.line_width > 1:
11         # Get distance transform
12         dt = np.abs(self.outlines - border_mask) * 255
13         dt[dt==0] = 255
14
15         # Use np.where to create border mask
16         border_mask = np.where(
17             dt > 0,
```

```
18              1,
19              0
20          )
21
22      # Return border mask as float32 numpy array
23      return border_mask.astype(np.float32)
24
```

2. **SantaCoder**:

```
1 def get_border_mask(self):
2     if self._outlines is None:
3         self._outlines = centrosome.outline.outline(self.labels)
4     outlines_mask = np.where(self._outlines, self._outlines, self.
      labels)
5     #outlines_mask = np.where(self._outlines, self._outlines, self
      ._outlines)
6     #outlines_mask = np.where(outlines_mask > 0, outlines_mask, 0)
7     return outlines_mask
8
```

3. **StarCoder**:

```
1 def get_border_mask(self):
2     if self._outlines is None:
3         self._outlines = centrosome.outline.outline(self.labels)
4     else:
5         for i in range(self._outlines.shape[0]):
6             self._outlines[i] = self._outlines[i] | centrosome.
      outline.outline(self.labels[i])
7     if self.line_width is not None and self.line_width > 1:
8         dist_trans = ndimage.distance_transform_edt(~self.
      _outlines)
9         border_mask = np.where(dist_trans <= self.line_width, 1,
      0)
10    else:
11        border_mask = self._outlines
12    return border_mask.astype(np.float32)
13
```

4. **StarCoder+**:

```
1 @property
2 def outlines(self):
3     if not hasattr(self, '_outlines'):
4         self._outlines = centrosome.outline.outline(self.labels)
5         return self._outlines
6     else:
7         for i in range(len(self.labels)):
8             self._outlines[i] = centrosome.outline.outline(self.
      labels)
9             self._outlines = self._outlines | self._outlines
10        return self._outlines
11
```

5. **InstructCodeT5+**:

```
1 def get_border_mask(self,labels):
2
3     if self._outlines is None:
4         self._outlines = outline(labels)
5
6     mask = self._outlines
7
8     if self.line_width is not None:
9         hw = self.line_width
```

```
10
11          if self.line_width >1:
12
```

6. **CodeGen**:

```
1  # Do not write any comments in your code.
2  #This is in python
3  #Write a function called "get_border_mask" that takes in a class
       object as an argument and returns a binary mask of all the
       points on the border of objects. The function should check if
       the instance variable "_outlines" is None and if it is, then
       use the labels attribute of the class object to generate the
       outlines mask using "centrosome.outline.outline(labels)" method
       . If the "_outlines" variable is not None, then it should
       iterate over "centrosome.outline.outline(labels)" for each i,
       and update the "_outlines" variable using the bitwise OR
       operator with the new mask. If the line_width attribute of the
       class object is not None and greater than 1, the function
       should calculate the distance transform using the "~self.
       _outlines" mask and compute the binary mask for borders with
       the given line width (hw) using numpy.where(). Finally, the
       function should return the mask as a float32 numpy array. Make
       sure to import the necessary dependencies including "centrosome
       .outline" and "numpy" and also initialize the "_outlines"
       variable in the class.
4  #
5  #The function is located in the class OutlinesMixin
6  #
7  #@property
8  #def get_border_mask(self):
9
```

7. **CodeGen2**:

```
1  def get_border_mask(self):
2      """
3      Computes the mask for the outlines of a morphology.
4
5      Parameters
6      ----------
7      self : morphology object
8
9      Returns
10     -------
11     border_mask :
12
```

8. **GPT-3.5**:

```
1  @property
2  def get_border_mask(self):
3      if self._outlines is None:
4          self._outlines = outline(self.labels)
5      else:
6          for i in range(1, np.max(self.labels)):
7              self._outlines |= outline(self.labels == i)
8
9      if self.line_width is not None and self.line_width > 1:
10         dt = np.array(self._outlines, dtype=np.uint8)
11         dt = np.where(dt < 0.5, 1, 0)
12         dt = ndimage.distance_transform_edt(dt)
13         dt = (dt < self.line_width).astype(float)
14         return dt
15     else:
16         return np.array(self._outlines, dtype=np.float32)
17
```

9. **GPT-4**:

```
def outlines(self):
    if self._outlines is None:
        self._outlines = centrosome.outline.outline(self.labels)
    else:
        for i in range(len(self.labels)):
            self._outlines |= centrosome.outline.outline(self.
    labels[i])

    if self.line_width is not None and self.line_width > 1:
        distances = scipy.ndimage.distance_transform_edt(~self.
    _outlines)
        hw = self.line_width / 2
        self._outlines = numpy.where(distances <= hw, 1, 0)

    return self._outlines.astype(numpy.float32)
```

In comparing the various model's outputs to the chosen prompt, it is clear that there are various levels of understanding of the prompt between the models. By far the worst performing models are the InstructCodeT5+, CodeGen, and CodeGen2 models. All of these models failed to generate a full function. This is likely due to these models being trained to do one-line completions rather than full-function generation. InstructCodeT5+ did the best out of the three, as it at least generated part of a function and showed an understanding of the self._outlines instance variable unlike both the CodeGen and CodeGen2 models. However, InstructCodeT5+ also got the function signature wrong, showing no understanding in the prompt of what the structure of the function it was trying to generate was. The CodeGen2 model did not generate any function body; however, it did generate the function signature correctly and generated part of a useful docstring for the function. Out of these three models, the original CodeGen model performed by far the worst, as it simply regurgitated the prompt, not generating any new code.

For the remaining five models, there were different levels of understanding. The two models that demonstrated the deepest levels of understanding of the prompt were the StarCoder+ and GPT-3.5 models, as they were the only two models to recognize that the get_border_mask function has an @property function decorator. Furthermore, they both showed some understanding of the external function calls that were necessary to complete the function, with Star-Coder+ calling the `centrosome.outline.outline` function correctly and GPT-3.5 using the `ndimage.distance_transform_edt` function, although not completely correctly, as the correct function call would be `scipy.ndimage.distance_transform_edt`. However, the logic that each of these models use to perform the get_border_mask function is correct, with GPT-3.5 getting closer to the intended logic. For instance, GPT-3.5 makes the `self.line_width is None` check.

The remaining three models not yet mentioned in this analysis are InCoder, SantaCoder, and Star-Coder. These three models generated complete functions unlike InstructCodeT5+, CodeGen, and CodeGen2; however, they did not include the @property function decorator, unlike the StarCoder+ and GPT-3.5 models. Out of these three "middle-of-the-pack" models, StarCoder performs especially well, as it shows understanding of the three necessary instance variables, namely `self._outlines`, `self.labels`, and `self.line_width`, and uses both the `centrosome.outline.outline` and `ndimage.distance_transform_edt` external function calls. However, like GPT-3.5, it does not quite make the ndimage external call correctly. Furthermore, the structural logic of StarCoder's code is similar to the golden code, on the same level as GPT-3.5. As for InCoder and SantaCoder, although they both generated full functions, their functions were somewhat off in their structural logic. SantaCoder performs decently, using the `centrosome.outline.outline` external package correctly; however, the function it generated is overly simple and does not meet the prompt description guidelines. As for InCoder, it uses a nonexistent `self.outlines` instance variable, instead of the intended `self._outlines` instance variable. Furthermore, it calls the nonexistent function `self.outline.outline` instead of the intended `centrosome.outline.outline` function. By contrast, InCoder writes a more involved function than SantaCoder which more closely mirrors the guidelines provided by the prompt.

# K  PROMPT LENGTH STUDY

As shown by the following scatterplots, it seems that on models with an average Pass@K score of at least 2%, there is an inverse relationship between the number of tokens in the prompt and the Pass@K score. Furthermore, for models such as SantaCoder and GPT, the performance fell sharply after around 500 tokens. This could be due to the massive amount of context "confusing" the models. When we look at the *Necessary Only* prompts, a similar trend occurs for longer contexts, but the phenomenon is not as pronounced, as on average these prompts are significantly shorter.

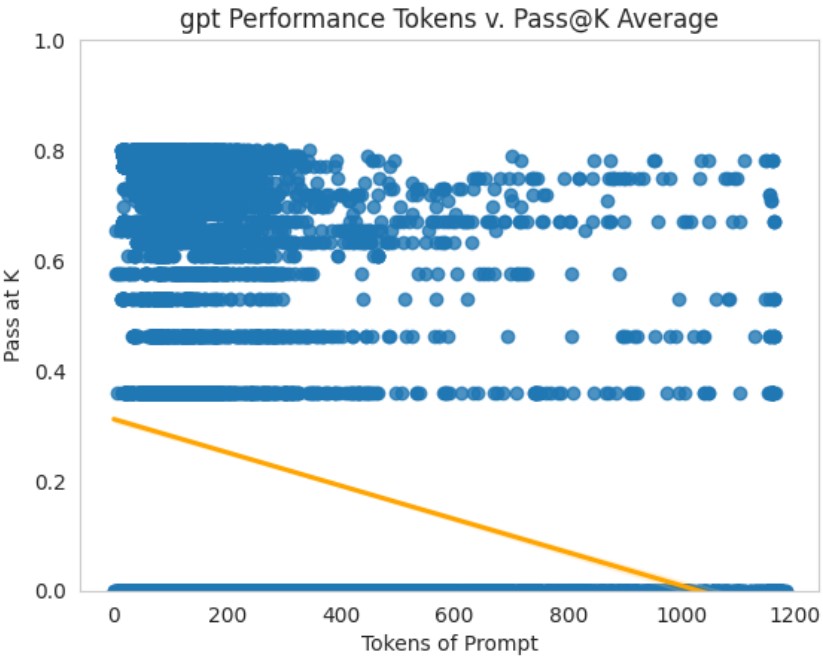

Figure 8: The scatterplots of the correlation of Pass@K and tokens of prompt. Statistics for GPT-4.

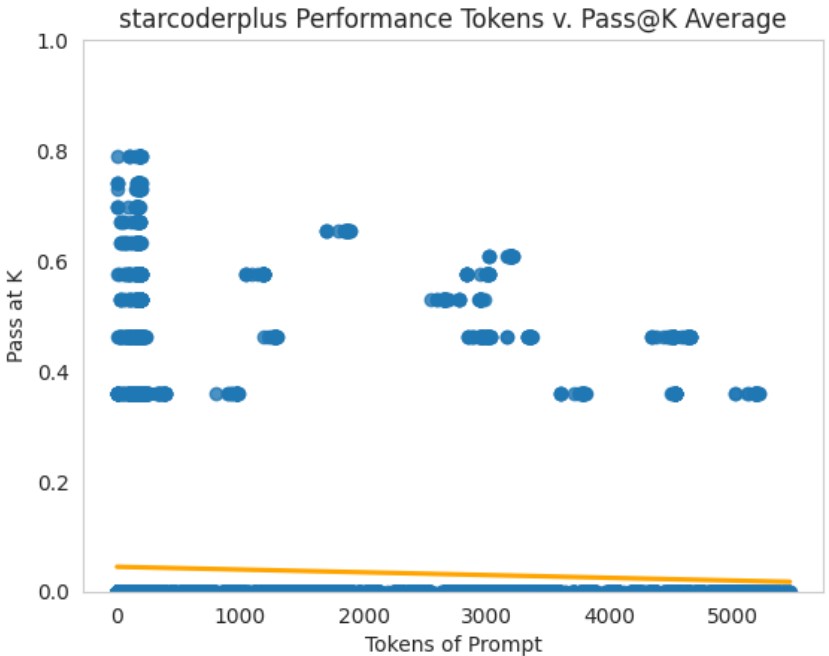

Figure 9: Statistics for Starcoderplus

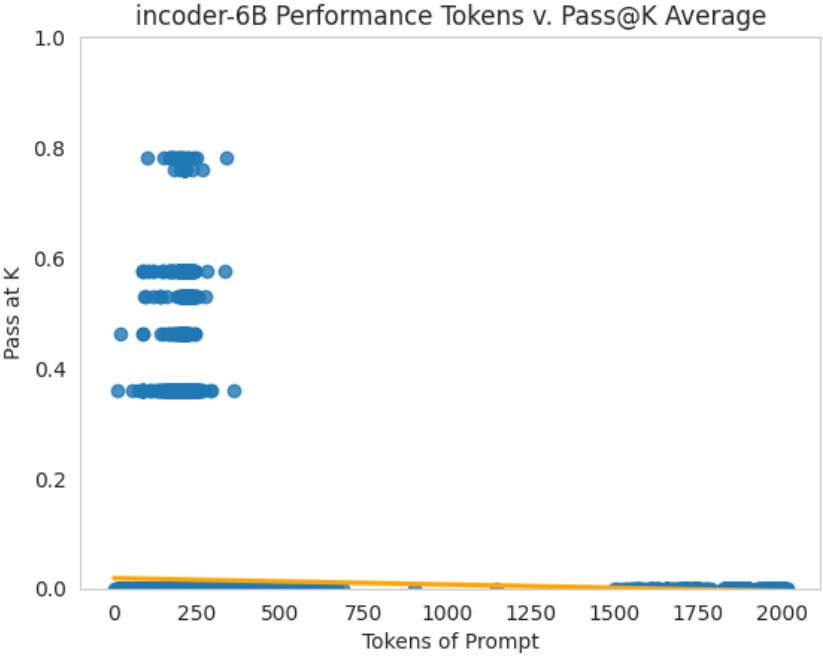

Figure 10: Statistics for Incoder

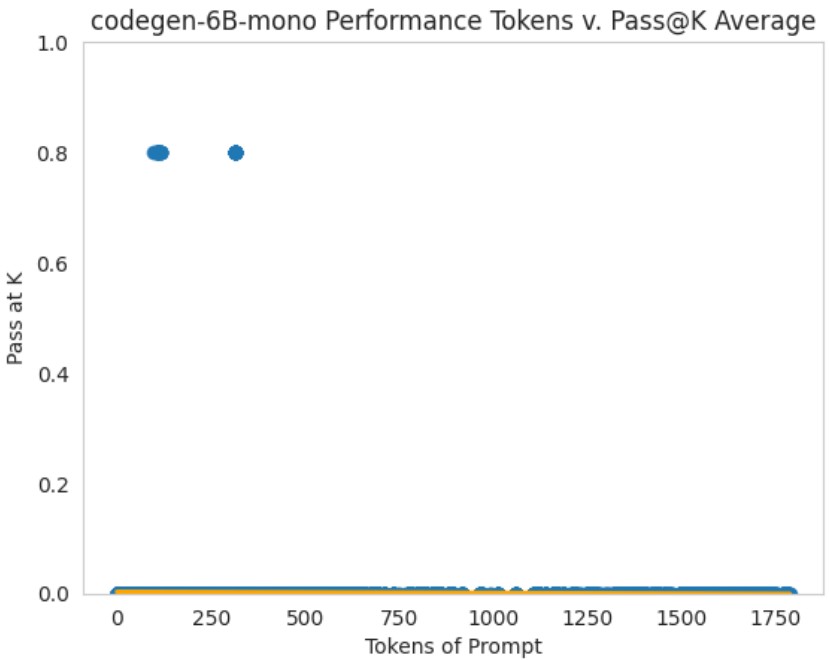

Figure 11: Statistics for CodeGen

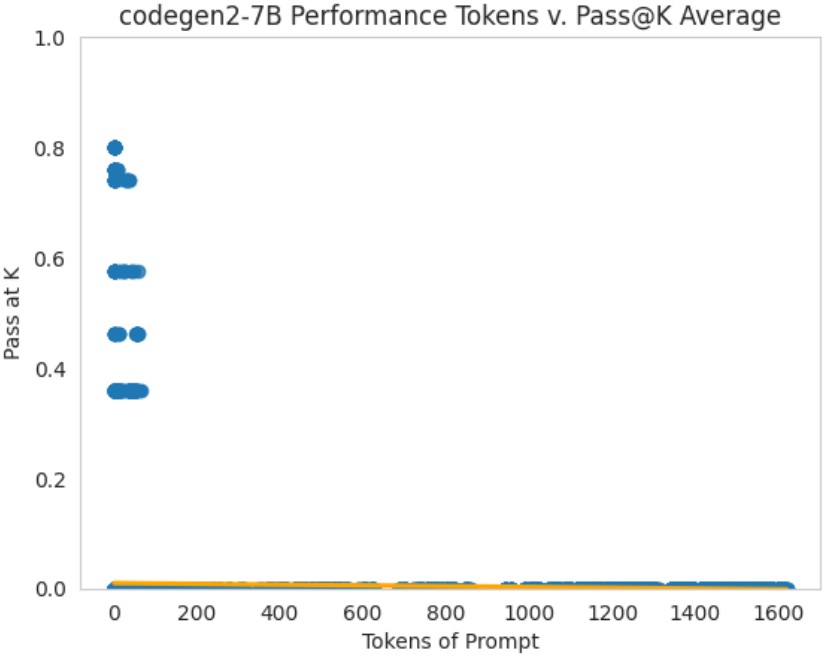

Figure 12: Statistics for CodeGen2

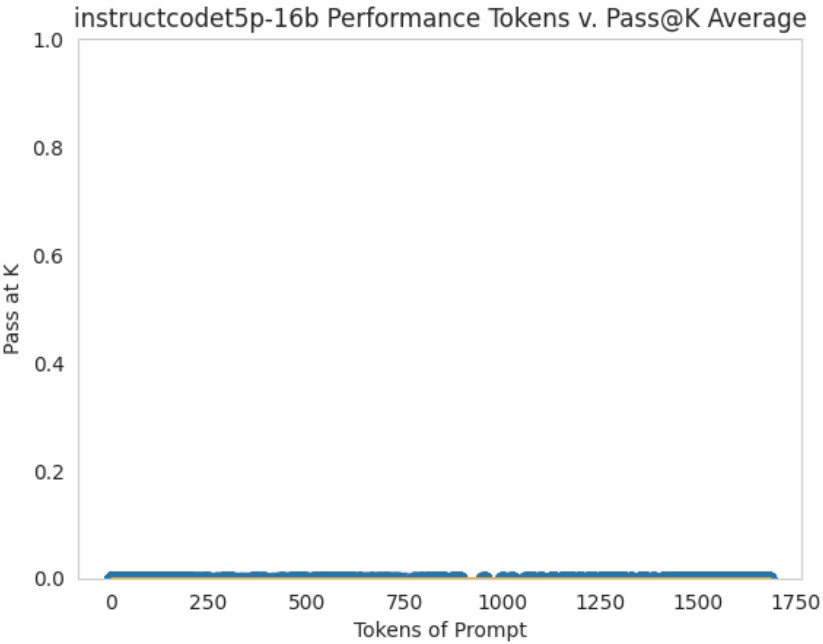

Figure 13: Statistics for InstructCodeT5+

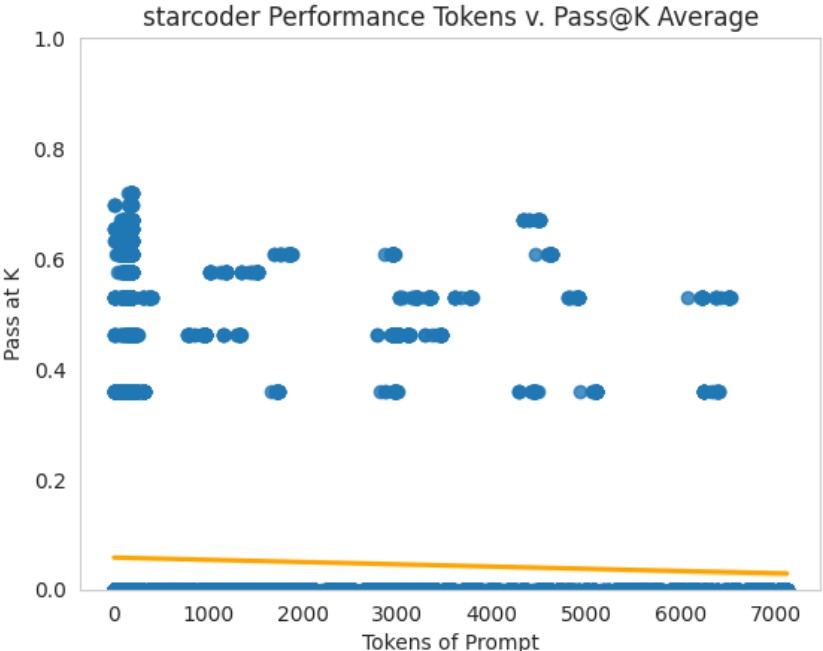

Figure 14: Statistics for Starcoder

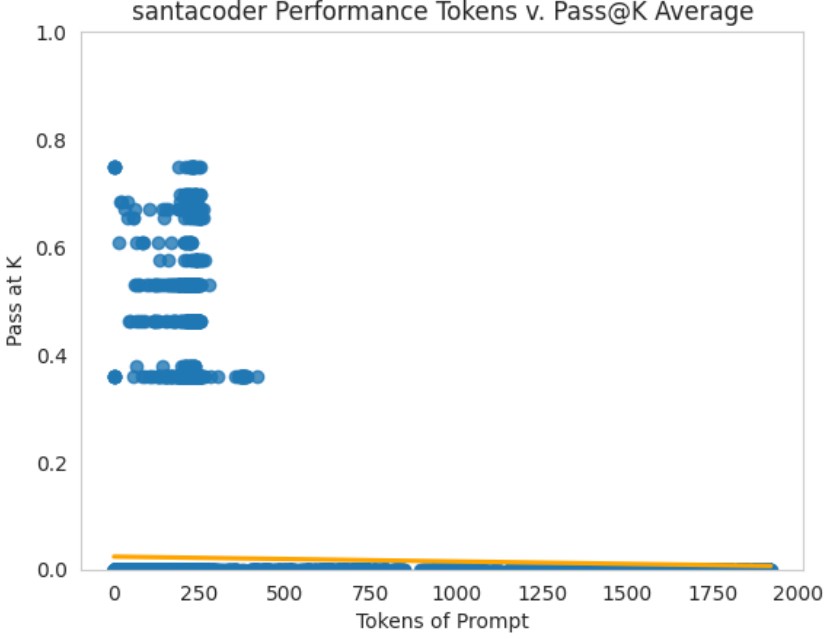

Figure 15: Statistics for Santacoder

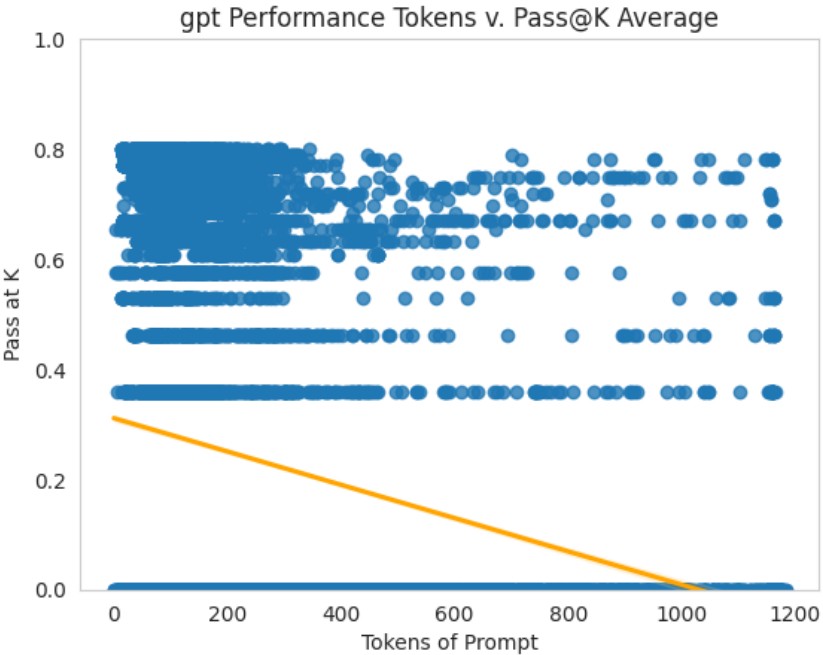

Figure 16: Statistics for GPT-3.5

## L  TESTING FRAMEWORK

An outline of our testing frame is as follows:

1. **Preprocessing:** Each generated output is cleaned and minor corrections are applied in accordance with the above steps. We append a call to the main() function at the end of the file.
2. **Container Preparation:** The output from LLMs, alongside the environment files and sample test cases, are copied into the container.
3. **Fuzzing Execution:** The test cases are executed with a memory cap of 7GB and a time limit of 60 seconds. A test is considered passed if the actual output precisely matches the expected output, for all cases tested.
4. **Analysis:** We gather and analyze the results from these executions.

Through this approach, we provide a secure and efficient testing framework, promising robustness in the evaluation of generated code.

### L.1  MOTIVATION BEHIND FUZZ TESTING

We decided to utilize concepts from fuzz testing, as fuzz testing is widely used in the industry to capture bugs, crashes, security vulnerabilities, etc. in functions. However, in these cases, they do not have access to a "correct" version of the function; instead, they are merely creating inputs to intentionally try to crash the program, find out of bounds memory accesses, etc. Our situation is unique because we have the "golden code", or the ground truth version of the code, so given an input, we definitely know what the expected output should be, which is not something that's usually available in typical fuzz testing frameworks. Therefore, our situation could be considered a mixture of both unit testing and fuzz testing.

Given this requirement, and the goal of large-scale prompt generation, we decided to implement our own framework. We set out to accomplish two things: make the annotation process a lot easier for human editors, and support our feature set that combines both elements from unit testing and elements from fuzz testing. We believe that our resulting pipeline is more intuitive than piecing together other testing frameworks, and in our annotation process it proved to make things efficient, enabling larger-scale annotation, as the goal of the paper.

Furthermore, note that while handwritten test cases would likely target edge cases of a program (eg. branch coverage, conditional coverage), the probability of our fuzz testing framework to hit all of the same edge cases is high given 1000 iterations of randomly generated inputs. This means that we can save a significant amount of time building the dataset, as we only need to write an outline of a test case, and let the framework handle the rest. Therefore, we can think of the framework as thousands of "random unit tests," with a high probability that these unit tests would include handwritten test cases, if we had written them.

In terms of variable generation, we replace the <|var_type;parameter|> syntax with random values each iteration, for an unlimited number of iterations. These parameters are modifiable, and we implemented this system to be flexible, so that we can target specific scopes for fuzz testing. We check correctness by substituting the exact same variables in the original code, and checking if the outputs of the two functions match. This indicates identical functionality with the original code.

Here is an example of random integer and numpy array generation:

```
import numpy
import skimage.morphology
import os
<<insert solution here>>
def main():
    numpy.random.seed(<|int;range=0,100|>)
    labels = numpy.random.randint(2, size=(3, 3))
    diameter = <|int;range=2,10|>
    print(fill_object_holes(labels, diameter))
if __name__ == "__main__":
    main()
```

Example of random string generation:

```
1  import random
2  [IMPORTS REDACTED FOR CONCISENESS]
3  import warnings
4  from textwrap import wrap
5  import string
6  import zlib
7  import io
8  from os.path import isfile
9  class GenipeError(Exception):
10         pass
11 _CHECK_STRING = b'GENIPE INDEX FILE'
12 def dosage_from_probs(homo_probs, hetero_probs, scale=2):
13     """Computes dosage from probability matrix (for the minor allele).
14     Args:
15         homo_probs (numpy.array): the probabilities for the homozygous
       genotype
16         hetero_probs (numpy.array): the probabilities for the
       heterozygous
17                                            genotype
18         scale (int): the scale value
19     Returns:
20         numpy.array: the dosage computed from the probabilities
21     """
22     return (homo_probs + (hetero_probs / 2)) * scale
23 <<insert solution here>>
24 def main():
25     np.random.seed(<|int;range=0,100|>)
26     prob_matrix = np.random.rand(10, 10)
27     a1 = <|string|>
28     a2 = <|string|>
29     print(maf_dosage_from_probs(prob_matrix, a1, a2))
30 if __name__ == "__main__":
31     main()
```

<|int|> denotes an integer. If left without parameters, then in one iteration of the program, this will be replaced with a random integer between INT_MIN and INT_MAX before compile time (or in this case, before the Python file is executed). There are parameters that can be passed in, that include range, even/odd, etc. Similarly, for <|string|> this generates a random ASCII string of any type. It can be further narrowed down into ASCII strings only, lowercase only, specific characters only, etc. by passing in the relevant parameters. These random inserted values can be manipulated to become part of a larger data structure, for example, a Numpy array, or a mock Python object.

When these files are executed, we replace «insert solution here» with the golden code on one iteration, and the error corrected generated code on a corresponding iteration. The fuzzing framework is designed so that the same inputs will be passed to this pair of iterations, meaning that we should be getting the same output (none of the functions have a non-deterministic component to them). Therefore this supports one aspect of the "secure" testing framework, as we have created an environment where all else is equal, except for the generated/golden code.

## M    PROMPT LENGTH ABLATION STUDY

We have conducted additional experiments to illustrate the effect of different prompts.

As a result, our study revealed that more detailed text descriptions improve code generation, particularly for Python, but they have less impact on Java. However, irrespective of the level of detail, the core structure of the generated code remains similar. Hence, ambiguity in descriptions will not dramatically influence the outcome.

Specifically, in response to concerns about the ambiguity of the Python text descriptions, we conducted a study to examine the effects of ambiguous prompts on code generation. Using ChatGPT, we generated both "short" and "long" summaries, with the former being more concise and potentially omitting some details, and the latter providing more detail to reduce ambiguity.

For the "short" summaries, we used the following prompt:

```
1 Please write a clear and succinct prompt in one paragraph directing the
      user to write the following function. In the prompt make sure to
      include sufficient details on what the following function does.
```

For the "long" summaries, we used the following prompt:

```
1 Please write a highly detailed prompt in one paragraph directing the user
       to write the following function. In the one paragraph prompt make
      sure to include all the details on what the following function does.
      If you are unsure of some. of the variable and function references
      make an educated guess.
```

Passing the above prompts into ChatGPT, we generated "short" and "long" summaries. Then, we used these summaries to construct "short" and "long" prompts.

We calculated Pass@K results on ChatGPT generations for both "short" and "long" prompts. The results are as follows:

| Prompt | Java | | | | Python | | | |
|---|---|---|---|---|---|---|---|---|
| | Pass@1 | Pass@5 | Pass@10 | Pass@20 | Pass@1 | Pass@5 | Pass@10 | Pass@20 |
| Short Summary | 34.700 | 42.313 | 43.000 | 43.800 | 25.960 | 36.371 | 39.825 | 42.484 |
| Long Summary | 36.800 | 46.412 | 49.842 | 54.000 | 38.567 | 52.419 | 56.439 | 60.510 |

Below are examples of "short" and "long" prompts:

```
1 SHORT PROMPT EXAMPLE
2 #This is in python
3 # Write a Python function called "UnifyLevels" that takes in three
      parameters: "baseLevel" (a constant array of integers), "addonLevel"
      (a constant array of integers), and "windowSize" (an integer). The
      function merges the two lists of breakpoints, but drops addonLevel
      values that are too close to baseLevel values. The function then
      returns the merged list of breakpoints as an array of integers. If
      the "addonLevel" list is empty, the function should return the "
      baseLevel" as is.
4 # Return only the code in the completion. I don't want any other comments
      . Don't say "here is your code" or similar remarks.
5 # def UnifyLevels(baseLevel, addonLevel, windowSize):
```

```
1 LONG PROMPT EXAMPLE
2 #This is in python
3 # Write a function called "UnifyLevels" that takes in three parameters: "
      baseLevel" (a list of integers), "addonLevel" (a list of integers),
      and "windowSize" (an integer). The purpose of this function is to
      merge the two lists of breakpoints (baseLevel and addonLevel), but
      drop any values from addonLevel that are too close to values in
      baseLevel. The function should first check if addonLevel is empty. If
       it is, the function should simply return baseLevel. Otherwise, the
```

```
     function should create an empty list called "joinedLevel" and an
        integer variable called "addon_idx" initialized to 0. The function
        should then iterate over each element in baseLevel using a for loop.
        Inside the loop, there should be a while loop that continues as long
        as addon_idx is less than the length of addonLevel. Within the while
        loop, the function should retrieve the element at index addon_idx
        from addonLevel and assign it to a variable called "addon_elem". Next
        , there should be three cases to handle:
4  # - If addon_elem is less than base_elem minus windowSize, the function
        should append addon_elem to joinedLevel, increment addon_idx by 1,
        and continue to the next iteration of the while loop.
5  # - If addon_elem is between base_elem minus windowSize and base_elem
        plus windowSize (inclusive), the function should increment addon_idx
        by 1 and continue to the next iteration of the while loop.
6  # - If addon_elem is greater than base_elem plus windowSize, the function
         should assert that base_elem plus windowSize is less than addon_elem
        , break out of the while loop, and move to the next iteration of the
        for loop.
7  After each iteration of the for loop, the function should append
        base_elem to joinedLevel. Once the for loop is complete, the function
         should check if there are any remaining elements in addonLevel
        starting from addon_idx. If there are, the function should extend
        joinedLevel with those elements. Finally, the function should return
        a numpy array containing the sorted elements of joinedLevel, with the
         data type set to np.int_.
8  # Return only the code in the completion. I don't want any other comments
        . Don't say "here is your code" or similar remarks.
9  # def UnifyLevels(baseLevel, addonLevel, windowSize):
```

According to our study, using more detailed prompts improved the performance, especially for Python generations. Python Pass@1 performance notably increased from 26.0% to 38.6% with more detailed prompts. However, the improvement for Java was less pronounced, with only a minor increase from 34.7% to 36.8%. Despite the difference in performance, the structural similarity between code generated from both short and long prompts remained high. Therefore, even if a prompt is ambiguous or trivial, it will not significantly affect the fundamental structure of the generated code. Consider the following short prompt and its generated example:

```
1
2  SHORT PROMPT
3
4
5  #This is in python
6  # Write a function called "FindLocalPeaks" that takes in a constant array
         of floats called "signal" as a parameter. The function should find
        the locations of local maxima on positive values and local minima on
        negative values in the "signal" array. The first and last index of
        the array should not be considered as extrema. The function should
        iterate through the array and compare each element with its adjacent
        elements to determine if it is a local peak. If a local peak is found
        , its index should be stored in an array called "peakLoc". The
        function should return the "peakLoc" array as the output. Make sure
        to include the necessary import statement for numpy.
7  # Return only the code in the completion. I don't want any other comments
        . Don't say "here is your code" or similar remarks.
8  # def FindLocalPeaks(signal):
```

```
1
2  SAMPLE GENERATION
3  import numpy as np
4  def FindLocalPeaks(signal):
5      peakLoc = []
6      for i in range(1, len(signal)-1):
7          if signal[i-1] < signal[i] > signal[i+1] and signal[i] > 0:
8              peakLoc.append(i)
```

```
 9           elif signal[i-1] > signal[i] < signal[i+1] and signal[i] < 0:
10               peakLoc.append(i)
11      return peakLoc
```

and here is the corresponding long prompt and its sample generated example:

```
1 LONG PROMPT
2 #This is in python
3 # Write a function called "FindLocalPeaks" that takes in one parameter, "
      signal", which is a constant array of floats. The function is used to
       find local maxima on positive values and local minima on negative
      values. The function excludes the first and last indices of the "
      signal" array as possible extrema. The function should return an
      array of integers, "peakLoc", which represents the locations of these
       extrema in the "signal" array. The function should follow the logic
      provided in the source code provided below, which includes conditions
       for various scenarios when comparing adjacent elements in the "
      signal" array. When a local maxima or minima is found, its location
      should be appended to the "peakLoc" array. If you encounter equal
      values while comparing adjacent elements, there are additional
      conditions that need to be followed. Remember to import the necessary
       modules and define any necessary variables.
4 # Return only the code in the completion. I don't want any other comments
      . Don't say "here is your code" or similar remarks.
5 # def FindLocalPeaks(signal):
```

```
1
2 SAMPLE GENERATION
3 def FindLocalPeaks(signal):
4     import numpy as np
5     peakLoc = []
6     for i in range(1, len(signal)-1):
7         if signal[i] > 0:
8             if signal[i] > signal[i-1] and signal[i] > signal[i+1]:
9                 peakLoc.append(i)
10        elif signal[i] < 0:
11            if signal[i] < signal[i-1] and signal[i] < signal[i+1]:
12                peakLoc.append(i)
13    return np.array(peakLoc)
```

As can be seen from the above example, using a more detailed prompt (10 lines of description vs. 7 lines of description) does not have an outsized effect on the structure of the sample generations, as the functions generated from the two different prompts are structurally similar. Therefore, the improved performance likely stems from the small details that might be omitted in the concise prompts that are relevant to functional performance.

# N  OBFUSCATION ABLATION STUDY

We conducted a study using "obfuscated prompts". The framework for this "obfuscated prompts" ablation study drew inspiration from Lachaux et al. (2020); Yang et al. (2023). The prompts used were primarily "short" GPT summaries in their original form, with function or parameter names substituted with placeholder terminology deliberate adjustment to hinder a model's ability to memorize portions of the training data.

Our analysis of the results shows that there is a performance decrease with obfuscated prompts compared to their non-obfuscated counterparts. However, this difference doesn't reach a magnitude to justify attributing ChatGPT's performance exclusively to the memorization of training data.

Below we provide an example showing a comparison between an obfuscated prompt with a non-obfuscated prompt.

```
1  SAMPLE ORIGINAL PROMPT
2  #This is in python
3  # Write a Python function called "unpipe_name" that takes in a string
      argument called "name" and returns a new string that is the single
      gene name after removing any duplications and pipe characters. This
      function is designed to fix the duplicated gene names produced by
      Picard CalculateHsMetrics, which combines the labels of overlapping
      intervals by joining all labels with '|'. If the name only consists
      of a single gene name, then that name is returned. If there are
      duplicated gene names and a meaningless target name, such as '-' or
      any name in the IGNORE_GENE_NAMES set, then they are removed. If
      there are still duplicated gene names, then the longest name is taken
      . If there are multiple equally long gene names, then one of them is
      arbitrarily selected and a warning is logged.
4
5  # Return only the code in the completion. I don't want any other comments
      . Don't say "here is your code" or similar remarks.
6  # def unpipe_name(name):
```

```
1  SAMPLE OBFUSCATED PROMPT
2  # This is in python
3  # Write a Python function called "FUNCTION" that takes in a string
      argument called "VAR0" and returns a new string that is the single
      gene VAR0 after removing any duplications and pipe characters. This
      function is designed to fix the duplicated gene names produced by
      Picard CalculateHsMetrics, which combines the labels of overlapping
      intervals by joining all labels with '|'. If the VAR0 only consists
      of a single gene VAR0, then that VAR0 is returned. If there are
      duplicated gene names and a meaningless target VAR0, such as '-' or
      any VAR0 in the IGNORE_GENE_NAMES set, then they are removed. If
      there are still duplicated gene names, then the longest VAR0 is taken
      . If there are multiple equally long gene names, then one of them is
      arbitrarily selected and a warning is logged.
4
5  # Return only the code in the completion. I don't want any other comments
      . Don't say "here is your code" or similar remarks.
6  # def FUNCTION(VAR0):
```

Notice that, by replacing specific function names and parameters with generic monikers like FUNCTION and VAR0 in our obfuscated prompts, we sought to limit the potential influence of prior training data memorization in ChatGPT's code generation. Overall, this process aimed to increase the difficulty for the model to rely on memorization as a primary means of accurate code generation, thus ensuring that the model's performance is based on interpreting the structure and requirement of the task, not on recollecting learned patterns.

We carried out Pass@K tests on the obfuscated prompts. The results were comparable to those achieved with non-obfuscated versions, although they were slightly lower, showing that obfuscation of the prompts does not significantly impede code generation performance. This implies that GPT 3.5's performance on our dataset can not be attributed solely to memorization.

| Prompt | Java | | | | Python | | | |
|---|---|---|---|---|---|---|---|---|
| | Pass@1 | Pass@5 | Pass@10 | Pass@20 | Pass@1 | Pass@5 | Pass@10 | Pass@20 |
| Short Summary | 34.700 | 42.313 | 43.000 | 43.800 | 25.960 | 36.371 | 39.825 | 42.484 |
| Obfuscated Summary | 30.700 | 40.684 | 44.250 | 47.900 | 22.771 | 32.733 | 35.872 | 38.790 |

Our experiment indeed shows that obfuscation of the prompts does not significantly impede code generation performance. Obfuscated prompts marginally underperformed in comparison to non-obfuscated ones, suggesting that memorization might play a role in ChatGPT's performance. Yet, the proximity of the results demonstrates that the model's understanding of the task described in the prompt is the primary driver of its performance.

Moreover, the fact that obfuscated and non-obfuscated short prompts yield similar generated examples serves as further evidence, testifying to the model's capacity to grasp the task's requirements and generate relevant codes even when specific cues from the prompt, such as function or parameter names, are withheld.

This finding supports the view that language models like ChatGPT generate code predominantly based on their understanding and interpretation of the task presented to them and that memorization, while present to an extent, does not appear to be the dominant factor.

Here is an example to further bolster our case, again showing that there is no data leakage. This example shows more detail on the generation:

```
1 SHORT PROMPT
2 #This is in python
3 # Write a function called '_neighbors' that takes in a two-or-three
      dimensional image and returns the number of neighbor pixels for each
      pixel of the image. The function should first pad the image with
      zeros, then create a mask where any pixel value greater than zero is
      considered a neighbor. Next, it should compute the response using a
      uniform filter and multiply it by a scaling factor. The labels are
      then computed by multiplying the response with the mask and slicing
      the padded image. Finally, the function should return the labels as
      an array of unsigned integers.
4 # Return only the code in the completion. I don't want any other comments
      . Don't say "here is your code" or similar remarks.
5 # def _neighbors(image):
```

```
1
2
3 SAMPLE GENERATION with ORIGINAL PROMPT
4
5
6 def _neighbors(image):
7     from skimage.filters import uniform_filter
8     from scipy import ndimage
9     scaling_factor = 10
10     mask = (image > 0).astype(int)
11     padded_image = np.pad(image, ((1,1), (1,1), (0,0)), 'constant',
      constant_values=0)
12     response = uniform_filter(padded_image, size=3)
13     labels = (response * scaling_factor * mask)[1:-1, 1:-1, :]
14     labels = ndimage.label(labels)[0]
15     return labels.astype(np.uint16)
16
17
```

```
1 SAMPLE OBFUSCATED PROMPT
2
3
4 # This is in python
5 # Write a function called 'FUNCTION' that takes in a two-or-three
      dimensional VAR0 and returns the number of neighbor pixels for each
```

```
       pixel of the VAR0. The function should first pad the VAR0 with zeros,
        then create a mask where any pixel value greater than zero is
       considered a neighbor. Next, it should compute the response using a
       uniform filter and multiply it by a scaling factor. The labels are
       then computed by multiplying the response with the mask and slicing
       the padded VAR0. Finally, the function should return the labels as an
        array of unsigned integers.
6  # Return only the code in the completion. I don't want any other comments
       . Don't say "here is your code" or similar remarks.
7  # def FUNCTION(VAR0):
8
9
10
```

```
1
2  SAMPLE GENERATION with OBFUSCATED PROMPT
3  def FUNCTION(VAR0):
4      import numpy as np
5      from scipy.ndimage import uniform_filter
6
7      VAR0 = np.pad(VAR0, ((1, 1), (1, 1)), mode='constant')
8      mask = VAR0 > 0
9      response = uniform_filter(mask, size=3)
10     scaling_factor = 10
11     labels = response * scaling_factor * mask * VAR0[1:-1, 1:-1]
12     return labels.astype(np.uint16)
```

The striking similarity between functions generated from both obfuscated and non-obfuscated prompts reaffirms that the model's ability to generate code is primarily based on its understanding and interpretation of the prompt and task, rather than the memorization of the training data.

Important to note, however, is the impressive resilience of the model's performance even after the obfuscation of specific functions or parameter names. This resilience indicates that the model does not rely heavily on these specific cues to generate appropriate code and can still produce functionally comparable results even when details are abstracted out.

## O   TOPIC ANALYSIS

The repositories that we used for this study came from a larger set of 1,720 bioinformatics repositories that were constructed in Russell et al. (2018). We manually selected 28 high-quality repositories from this set. Each repository is a codebase of one bioinformatics journal article. We used latent Dirichlet allocation (LDA) to infer topics for abstracts of the articles citing each repository in the main dataset.

From the LDA model, we identified terms that were primarily associated with a single topic. We chose a model with eight topics due to its maximal coherence of concepts within the top topic-specialized terms. We manually assigned a label to each of the eight topics that capture a summary of the top terms. We then classified each article abstract into one or more topics. The eight topics in the LDA model are described below.

1. Cancer and epigenetics: Cancer and epigenetics refer to the study of heritable changes in gene expression that do not involve changes to the underlying DNA sequence but can lead to the development and progression of cancer. These epigenetic modifications, which include DNA methylation, histone modifications, and small RNA-associated gene silencing, can turn genes on or off, influencing cancer susceptibility, initiation, progression, and response to treatment. Understanding these processes can aid in developing targeted therapies and better diagnostic tools for cancer patients.

2. Proteomics and microscopy: Proteomics and microscopy refer to the integration of proteomic analyses (the large-scale study of proteins, their structures, and functions) with advanced microscopy techniques. This integration allows scientists to visualize and quantify the spatial and temporal distribution of proteins within cells and tissues. By combining the detailed molecular information from proteomics with the high-resolution imaging from microscopy, researchers can gain deeper insights into cellular processes, protein-protein interactions, and the structural organization of cellular components.

3. Variant calling: Variant calling is a process in bioinformatics where sequence data (often from next-generation sequencing) are analyzed to identify differences, or variants, between a given sample and a reference genome. These variants can include single-nucleotide polymorphisms, insertions, deletions, and more. Identifying these variants is crucial for understanding genetic diversity, disease susceptibility, and personalizing medical treatments based on an individual's genetic makeup.

4. Genetics and population analysis: Genetics and population analysis in bioinformatics refers to the study of the genetic composition of populations and how it changes over time. This involves analyzing large datasets of genetic information to understand patterns of genetic variation, inheritance, and evolution within and between populations. Such analyses can provide insights into population structure, migration, adaptation, and the history of species or populations, as well as help identify genetic factors associated with diseases or traits in specific populations.

5. Structure and molecular interaction: Structure and molecular interaction in bioinformatics pertains to the study and prediction of the three-dimensional structures of biomolecules (like proteins and nucleic acids) and how they interact with one another. By using computational methods to model and analyze these structures, scientists can gain insights into the molecular mechanisms underlying biological processes. This understanding can be pivotal for drug design, predicting protein functions, and understanding the effects of genetic mutations on molecular interactions and stability.

6. Web and graphical applications: Web and graphical applications in bioinformatics refer to the development and use of online tools and software with user-friendly graphical interfaces to analyze, visualize, and interpret biological data. These applications allow both experts and non-experts to interact with complex datasets, facilitating tasks like sequence alignment, gene annotation, pathway analysis, and more. By providing accessible platforms, these applications help democratize the analysis and understanding of vast and intricate bioinformatics data.

7. Assembly and sequence analysis: Assembly and sequence analysis in bioinformatics involves the process of taking raw sequence data, typically from next-generation sequencing technologies, and reconstructing the original DNA or RNA sequences. Assembly might mean piecing together short DNA reads into longer sequences or entire genomes. Once assembled, sequence analysis is used to identify genes, predict their functions, detect variations, and compare sequences across different organisms or individuals. This foundational process is key to many downstream analyses in genomics and personalized medicine.

8. Transcription and RNA-seq: Transcription and RNA-seq (RNA sequencing) in bioinformatics relate to the study of the transcriptome—the complete set of RNA molecules expressed from genes in a cell or tissue. RNA-seq is a technique that uses next-generation sequencing to capture and quantify the RNA in a sample, providing insights into which genes are active (or "expressed") under specific conditions. By analyzing RNA-seq data, researchers can uncover patterns of gene expression, identify novel RNA species, and study regulatory pathways, aiding in understanding cellular responses, disease mechanisms, and developmental processes.

All eight topics were present across our subset of 28 repositories from which we created the benchmark dataset. A detailed breakdown of which topics corresponded to which repositories is as follows:

| Repository | Topic Areas |
| --- | --- |
| AdmiralenOla/Scoary | Genetics and population analysis, Assembly and sequence analysis |
| biocore/deblur | Transcription and RNA-seq |
| CellProfiler/CellProfiler | Cancer and epigenetics, Proteomics and microscopy |
| CGATOxford/UMI-tools | Genetics and population analysis |
| choderalab/ensembler | Structure and molecular interaction |
| etal/cnvkit | Variant calling |
| gem-pasteur/macsyfinder | Structure and molecular interaction, Web and graphical applications |
| hangelwen/miR-PREFeR | Transcription and RNA-seq |
| jnktsj/DNApi | Web and graphical applications, Transcription and RNA-seq |
| juliema/aTRAM | Genetics and population analysis, Assembly and sequence analysis |
| karel-brinda/rnftools | Variant calling, Structure and molecular interaction |
| mad-lab/transit | Web and graphical applications |
| makson96/Dynamics | Structure and molecular interaction |
| MikkelSchubert/paleomix | Variant calling, Assembly and sequence analysis |
| msproteomicstools/msproteomicstools | Proteomics and microscopy |
| ODonnell-Lipidomics/LipidFinder | Structure and molecular interaction |
| pgxcentre/genipe | Proteomics and microscopy, Genetics and population analysis |
| ratschlab/spladder | Transcription and RNA-seq |
| SamStudio8/goldilocks | Web and graphical applications |
| simonvh/fluff | Web and graphical applications |
| sjspielman/pyvolve | Structure and molecular interaction, Web and graphical applications |
| SpatialTranscriptomicsResearch/st pipeline | Proteomics and microscopy, Web and graphical applications, Transcription and RNA-seq |
| ursgal/ursgal | Proteomics and microscopy |
| vals/umis | Transcription and RNA-seq |
| williamgilpin/pypdb | Proteomics and microscopy, Web and graphical applications |
| zhanglab/psamm | Structure and molecular interaction, Web and graphical applications |
| zstephens/neat-genreads | Variant calling, Assembly and sequence analysis |
| samtools/htsjdk | Assembly and sequence analysis, Transcription and RNA-seq |

## P   MODEL ERROR DISTRIBUTIONS

| Models | Failure: Syntax Error | Failure: Runtime Error | Failure: Timeout Error | Failure: Output Disagreement | Passed Tests |
|---|---|---|---|---|---|
| CodeGen-6B-Mono | 11268 | 8176 | 1 | 148 | 105 |
| CodeGen2-7B | 12687 | 6718 | 0 | 211 | 79 |
| GPT-3.5 Turbo | 9231 | 10603 | 0 | 5624 | 6643 |
| InCoder-6B | 11268 | 8176 | 1 | 148 | 105 |
| InstructCodeT5P-16B | 19667 | 33 | 0 | 0 | 0 |
| SantaCoder | 14391 | 4601 | 1 | 555 | 139 |
| StarCoder | 26233 | 10688 | 0 | 1660 | 808 |

## Q  LANGUAGE ANALYSIS

Our decision to include Java and Python was based on an empirical investigation into the prevalence of different programming languages across bioinformatics repositories. We computed the total byte sizes of various languages across all repositories we surveyed. Out of the 13 languages analyzed (Python, Bourne Shell, R, Perl, Cpp, C, Java, Bourne_Again_Shell, MATLAB, m4, SQL, Ruby, PHP), Java was the most prevalent, with a byte size of 242,318,285 bytes, followed closely by Python at 192,324,018 bytes. The presence of Java as the most extensive language indicates that it plays a crucial role in the bioinformatics community, perhaps in the form of underlying tools or frameworks that support high-level bioinformatics operations. A detailed breakdown of the byte size of the various languages in our repository is as follows:

| Programming Language | Number of Bytes |
|---|---|
| Java | 242,318,285 |
| Python | 192,324,018 |
| C | 184,967,144 |
| C++ | 184,694,473 |
| Perl | 129,213,485 |
| R | 40,708,273 |
| Bourne Shell | 35,495,605 |
| PHP | 33,876,889 |
| MATLAB | 28,889,990 |
| SQL | 15,630,061 |
| Ruby | 8,935,640 |
| m4 | 7,956,980 |
| Bourne Again Shell | 2,851,620 |

## R    TEST CASE EXAMPLE

```
import random                    Context file
import hashlib
import numpy
import skimage
import skimage.measure
import scipy.ndimage
import os
import logging
F_ANGLE = 'Angle'
LOGGER = logging.getLogger(__name__)
class UntangleWorms:
    def ncontrol_points(self):
        return <|int;range=3,10|>
    <<insert solution here>>

def main():
    print(UntangleWorms().angle_features())
if __name__ == "__main__":
    main()
```

```
def angle_features(self):
    """Return a list of angle feature names"""
    try:
        return ['_'.join((F_ANGLE, str(n)))
for n in range(1, self.
            ncontrol_points() - 1)]
    except:
        LOGGER.error(
            ''Failed to get # of control
points from training file. Unknown number of
angle measurements''
            , exc_info=True)        Golden Code
        return []
```

```
def angle_features(self):
    # Define angle feature names
    angle_feature_names = ['Feature1',
'Feature2', 'Feature3']        Generated
                                  Code
    return angle_feature_names
```

Figure 17: Test case for UntangleWorms example. The context file includes various import dependencies and a class definition with a method placeholder for the solution. The UntangleWorms class comes from a GitHub repository file (`https://github.com/CellProfiler/CellProfiler/blob/master/cellprofiler/modules/untangleworms.py`) that was scraped in our study. UntangleWorms is an image analysis tool that was initially part of the paper "An image analysis toolbox for high-throughput *C. elegans* assays."

# S    ERROR DISTRIBUTION CHART

The errors include: 'different output' where the generated code's output did not match the golden code's output; 'invalid syntax' where syntax errors in the generated code prevented code execution; 'function timed out' where code execution exceeded time limits; and 'runtime error' where the generated code compiled successfully but failed to run. The vast majority of the generated code samples tested encountered a sort syntax or runtime error without resulting in an output. See Figure 18 for more detail.

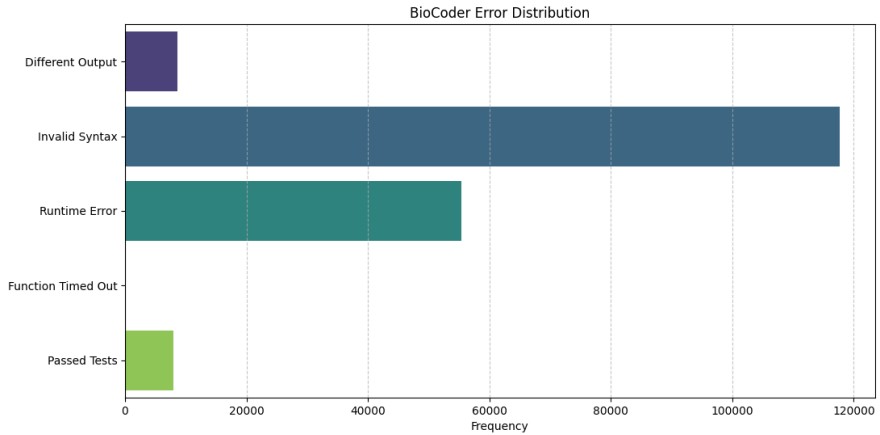

Figure 18: Error distribution aggregated over all models

## T  FUNCTION FILTERING PROCESS

When filtering the function, we were given a large baseline of functions. In order to decrease the manual workload of filtering through these functions, we initiated two rounds of automatic filtering. In the first round, every function and its associated comments underwent keyword filtering. The keywords were scraped from bioinformatics Wikipedia articles, with each function requiring at least 10 keyword matches to be considered relevant to bioinformatics. This round was critical to avoid any unnecessary OpenAI API calls, which are expensive. The remaining functions advanced to the second round, where we inputted each function into GPT-3.5, which assessed bioinformatics relevance. With the workload significantly reduced, we manually sorted through the remaining functions, leaving us with a final count of 1,026 Python functions and 1,243 Java functions (see Table 2). Note that in Java, there are no global variables, because all variables are encapsulated in classes. In Table 2, the "similar data" set includes an additional 157 Python problems and 50 Java problems, maintaining the same 253 Rosalind problem tally, thus reflecting the composition of the public data. The issues in this set were cherry-picked based on their L2 similarity norm of calculated statistics, ensuring close alignment to the mean distribution of the public data.

Indeed, knowledge of bioinformatics was an integral part of constructing this dataset and benchmark. We stressed how we used extensive bioinformatics-specific keywords and relied on the expertise of four students specializing in Computer Science and Computational Biology. Our team manually sifted through potential functions to build our final dataset.

It is important to note that our selections were made with a focus on bioinformatics-related coding problems. From the bioinformatics-related repositories, we obtained a vast number of functions. However, among these functions: some were highly specific to bioinformatics (like algorithms), while others were not directly relevant (like web page scripts).

Therefore, a key task in our process was to judiciously filter these functions. We had to strike a balance to ensure that our final dataset comprises functions that are truly bioinformatics-focused and applicable to our study. This filtering process was thorough and meticulous, undertaken by experts with knowledge in bioinformatics, thereby reaffirming the indispensable role of a bioinformatics understanding in this work.

To ensure relevance, we executed a recursive search in scraping the entire Wikipedia corpus using 100 seed words that are known to be relevant to bioinformatics. Eventually, we manually filtered all the collected keywords to ensure relevance, which resulted in a dictionary of bioinformatics/adjacent words.

We then employed a two-level filtering process: (1) In the first round, each function and corresponding comments passed through a keyword filter, requiring at least 10 keyword matches related to bioinformatics. (2) In the following round, we leveraged GPT3.5 to generate associated keywords for each function, which underwent manual review. This strategy minimized the manual workload and helped us retain high-quality, relevant functions for our dataset.

Thus, our effort to benchmark code generation, although general in nature, is deeply rooted in the context of bioinformatics, making use of carefully curated and filtered datasets based on bioinformatics problems. While an understanding of bioinformatics and biology may not be essential for using the benchmark, the benchmark, and the dataset were built to reflect the complexity and domain specifics of bioinformatics.

## U  SUMMARY AT BOTTOM RESULTS

Summary At Bottom results were omitted from Table 4 due to brevity reasons, and so are included here. More details about how Summary At Bottom prompts are constructed can be found in Appendix i.

| Model | Pass@1 | Pass@5 | Pass@10 | Pass@20 |
|---|---|---|---|---|
| InCoder-6B | 0.610 | 2.587 | 4.303 | 6.274 |
| SantaCoder-1.1B | 0.510 | 1.949 | 3.013 | 4.459 |
| StarCoder-15.5B | 6.465 | 13.824 | 16.746 | 19.076 |
| StarCoder-15.5B (finetuned) | \ | \ | \ | \ |
| StarCoder+ | 4.172 | 11.772 | 14.933 | 17.197 |
| CodeGen-6B-mono | 2.070 | 4.535 | 5.896 | 7.006 |
| CodeGen-16B-mono | 2.166 | 5.137 | 6.022 | 6.369 |
| CodeGen2-7B | 0.510 | 1.019 | 1.207 | 1.274 |
| GPT-3.5-Turbo | 13.439 | 20.040 | 22.460 | 25.478 |
| GPT-4 | 6.316 | 15.272 | 20.223 | 24.823 |

Table 11: Performance on "Summary at Bottom" rows for Python columns

## V    ERROR DISTRIBUTION GIVEN OVER ALL MODELS

The failure modes are similar between the models but not exactly the same.

| Models | Syntax Error | Runtime Error | Timeout Error | Output Disagreement | Passed |
|---|---|---|---|---|---|
| Overall | 117665 | 55351 | 4 | 8661 | 7982 |
| CodeGen | 11268 | 8176 | 1 | 148 | 105 |
| CodeGen2 | 12687 | 6718 | 0 | 211 | 79 |
| GPT3.5 | 9231 | 10603 | 0 | 5624 | 6643 |
| InCoder | 11268 | 8176 | 1 | 148 | 105 |
| InstructCodeT5P | 19667 | 33 | 0 | 0 | 0 |
| SantaCoder | 14391 | 4601 | 1 | 555 | 139 |
| StarCoder | 26233 | 10688 | 0 | 1660 | 808 |

Table 12: Error distribution per model.

Below is the failure breakdown for each model:

```
CodeGen-6B-Mono

Failure: Syntax Error = 11268
Failure: Runtime Error = 8176
Failure: Timeout Error = 1
Failure: Output Disagreement = 148
Passed Tests = 105

CodeGen2-7B

Failure: Syntax Error = 12687
Failure: Runtime Error = 6718
Failure: Output Disagreement = 211
Passed Tests = 79

GPT3.5-Turbo

Failure: Syntax Error = 9231
Failure: Runtime Error = 10603
Failure: Output Disagreement = 5624
Passed Tests = 6643

InCoder-6B

Failure: Syntax Error = 12777
Failure: Runtime Error = 6502
Failure: Timeout Error = 2
Failure: Output Disagreement = 309
Passed Tests = 100

InstructCodeT5P-16B

Failure: Syntax Error = 19667
Failure: Runtime Error = 33

SantaCoder

Failure: Syntax Error = 14391
Failure: Runtime Error = 4604
Failure: Timeout Error = 1
Failure: Output Disagreement = 555
Passed Tests = 139

StarCoder
```

```
46
47 Failure: Syntax Error = 26233
48 Failure: Runtime Error = 10688
49 Failure: Output Disagreement = 1660
50 Passed Tests = 808
```

## W    MORE MODELS

Here we aim to answer the question of why we didn't mention ChatGPT/Codex/Github copilot as baselines.

ChatGPT/Codex/Github copilot are all powered by a GPT model. In our paper, we provide results for GPT-3.5 and GPT-4, which was the model that ChatGPT and Github Copilot both used. As for Codex, it uses a weaker GPT 3.0 model and was deprecated by OpenAI at the time of testing, so we did not include it in the benchmark.

## X    IMPORTANCE OF THE CONTEXT

Imports and classes are predefined and are included in the context because, as we are testing function-level code generation, we are not prompting the model nor expecting the model to generate the classes it needs to pass the tests. Instead, we are testing the model's ability to extract the pertinent imports and classes from the context to use in the generated function.

To further illustrate how the prompt needs to contain information on the context to correctly generate the golden code, we provide an example below. You'll observe that the golden-code function uses the global variable "IGNORE_GENE_NAMES". We include this information in the prompt because it embeds the task's necessary foundations. Essentially, we ensure the prompt has enough context for an informed human to reproduce the function, implying that the function generated ought to incorporate the environment's existing resources (imports & classes) rather than generating new ones.

```
1  PROMPT:
2
3  This is in python.
4  Write a function called "unpipe_name" that takes in a string parameter
       called "name". The function takes a string containing multiple gene
       names separated by pipes, and returns a single gene name devoid of
       duplicates and pipe characters. The function checks if there are any
       duplicates, and removes any meaningless target names. If there are
       multiple gene names present, the function takes the longest name as
       the final name. If there are any ambiguous gene names, the function
       logs a warning and selects the longest name as the final name. The
       function should also import "cnvlib.params".
5
6  def unpipe_name(name):
7
8  Here are the imports:
9  import numpy as np
10 import logging
11 from . import params
12 from skgenome import tabio
13 Here are the global variables:
14 MIN_REF_COVERAGE = -5.0
15 MAX_REF_SPREAD = 1.0
16 NULL_LOG2_COVERAGE = -20.0
17 GC_MIN_FRACTION = 0.3
18 GC_MAX_FRACTION = 0.7
19 INSERT_SIZE = 250
20 IGNORE_GENE_NAMES = '-', '.', 'CGH'
21 ANTITARGET_NAME = 'Antitarget'
22 ANTITARGET_ALIASES = ANTITARGET_NAME, 'Background'
23 Here are the class declarations:
24 Here are the additional function declarations:
25 def do_import_picard(fname, too_many_no_coverage):
26     summary: Reads a file in 'picardhs' format, processes the data, and
       returns a modified dataframe.
27     param: fname (string) - the file name/path to be read in 'picardhs'
       format.
28     param: too_many_no_coverage (int) - if the number of bins with no
       coverage is greater than this value, a warning message is logged.
       Default is 100.
29     return: garr (pandas dataframe) - a modified dataframe with added
       columns 'gene' and 'log2' based on the original dataframe read from
       the input file.
30 def unpipe_name(name):
31     summary: Remove duplicate gene names and pipe characters from a given
        string.
32     param: name (str) - the string containing the gene names.
33     return: new_name (str) - the cleaned gene name string.
34 def do_import_theta(segarr, theta_results_fname, ploidy):
```

```
35      summary: A function for importing theta results and estimating copy
        number and log2 ratios of segments.
36      param: segarr (numpy array) - array of segments
37      param: theta_results_fname (str) - name of theta results file
38      param: ploidy (int) - ploidy of genome (default is 2)
39      return: generator of numpy arrays - array of segments with estimated
        copy number and log2 ratios.
40  def parse_theta_results(fname):
41      summary: Parses THetA results into a data structure with NLL, mu, C,
        and p* columns.
42      param: fname (str) - name of the file to parse the results from
43      return: (dict) - a dictionary containing the NLL, mu_normal,
        mu_tumors, C, and p* values
44  Here are the comments and the specs:
45  Write a function called "unpipe_name" that takes in a string parameter
        called "name". The function takes a string containing multiple gene
        names separated by pipes, and returns a single gene name devoid of
        duplicates and pipe characters. The function checks if there are any
        duplicates, and removes any meaningless target names. If there are
        multiple gene names present, the function takes the longest name as
        the final name. If there are any ambiguous gene names, the function
        logs a warning and selects the longest name as the final name. The
        function should also import "cnvlib.params".
46
47  def unpipe_name(name):
48  GOLDEN CODE:
49
50  def unpipe_name(name):
51      """Fix the duplicated gene names Picard spits out.
52
53      Return a string containing the single gene name, sans duplications
        and pipe
54      characters.
55
56      Picard CalculateHsMetrics combines the labels of overlapping
        intervals
57      by joining all labels with '|', e.g. 'BRAF|BRAF' -- no two distinct
58      targeted genes actually overlap, though, so these dupes are redundant
        .
59      Meaningless target names are dropped, e.g. 'CGH|FOO|-' resolves as '
        FOO'.
60      In case of ambiguity, the longest name is taken, e.g. "TERT|TERT
        Promoter"
61      resolves as "TERT Promoter".
62      """
63      if '|' not in name:
64          return name
65      gene_names = set(name.split('|'))
66      if len(gene_names) == 1:
67          return gene_names.pop()
68      cleaned_names = gene_names.difference(IGNORE_GENE_NAMES)
69      if cleaned_names:
70          gene_names = cleaned_names
71      new_name = sorted(gene_names, key=len, reverse=True)[0]
72      if len(gene_names) > 1:
73          logging.warning('WARNING: Ambiguous gene name %r; using %r', name
        ,
74              new_name)
75      return new_name
```

## Y    POTENTIAL CONCERNS ABOUT CHANGES IN PACKAGES

In addressing potential concerns about code generation issues due to changes in packages, we have taken steps to ensure stability and consistency in our testing environment. Our testing environment utilizes standardized and version-locked packages that guarantee stable results for our "golden" code samples. As such, we do not need to worry about package updates or feature deprecation.

Furthermore, when giving prompts, we have taken care to specifically mention the packages that need to be used, which guides the LLM to correctly utilize the expected versions of these packages. Any remaining discrepancies will be highlighted in our error analysis for the respective LLMs.

In the prompts we provide, we clearly specify which packages and functions should be used, including any external dependencies. This way, the LLM knows exactly which packages to use.

Take, for example, this dataset-generated prompt in Python:

```
#This is in python

#write a function called "binary_erosion" that takes in two parameters: "
    x_data" and "structuring_element". The function should first check if
    the "structuring_element" is a 2D array or not. If it is, and "
    x_data" is not a 2D array, the function should create a new array
    called "y_data" that is the same size as "x_data" and contains only
    zeros. The function should then loop through each slice of "x_data"
    and apply the skimage.morphology.binary_erosion function using the "
    structuring_element" as the structuring element. The result of each
    slice should be added to the corresponding slice of "y_data". Finally
    , the function should return "y_data". If the "structuring_element"
    is not 2D and "x_data" is 2D, the function should raise a
    NotImplementedError with the message "A 3D structuring element cannot
    be applied to a 2D image." Otherwise, the function should simply
    apply the skimage.morphology.binary_erosion function to "x_data"
    using "structuring_element" as the structuring element and return the
    result.
#
#def binary_erosion(x_data, structuring_element):
```

You can observe how the prompt meticulously guides the LLM to employ the "binary_erosion" function derived from the "skimage.morphology package". Any inaccuracies in using the specified packages, likely due to the model training on outdated or more advanced versions, would deem its output incorrect, leading us to categorize it as a failed generation. A paramount facet we are evaluating is the proficiency of the LLM in accurately implementing specified external packages as mentioned in the prompts.

## Z    DIFFERENCES BETWEEN PUBLIC, HIDDEN, AND SIMILAR SETS

While constructing the datasets, we observed the need for a categorization that simultaneously caters to present-day language learning models (LLMs) and future, more capable versions. Consequently, we delineated our public and hidden datasets, which admittedly, resulted in a considerable divergence between their summary statistics.

The public test set encapsulates relatively moderate challenges an LLM might encounter, keeping in mind the token limits of current-generation code LLMs, which range from 2,048 to 4,096 tokens. This constraint dictated that the public test set should comprise smaller and simpler functions.

Conversely, the hidden dataset was configured to house a broader, more challenging set of problems, aiming to benchmark future models, for instance, GPT4 with its anticipated 32K token limit. This two-tiered framework ensures the longevity of this benchmark as advancements unfold in the field of LLMs and facilitates additional investigations into domain-specific generative models.

Moreover, we have devised a "Similar Dataset", which is essentially a subset of the hidden dataset harboring statistics comparable to the public dataset. This new dataset ensures direct, fair comparisons between the public and the hidden tests.

The "public data" represents datasets with crafted specific contexts and corresponding test cases. The "hidden data" encompasses a wider array of intricate issues. Furthermore, the "similar data" is a subset of the hidden data, curated to mimic the overall distribution of the public data. This set includes an additional 157 Python problems and 50 Java problems, maintaining the same 253 Rosalind problem tally, thus reflecting the composition of the public data. The issues in this set were cherry-picked based on their L2 similarity norm of calculated statistics, ensuring close alignment to the mean distribution of the public data.

In addition, the reason why the number of hidden tests is much larger than public tests is that we aim to ensure the integrity of our benchmark. Thus, we divided our dataset into a much larger hidden set rather than a public set. We believe that access to too many public test cases might lead to potential overfitting while fine-tuning m odels, which would fundamentally undermine the benchmark.

## AA    PROMPT STRUCTURE ANALYSIS

As demonstrated by the scatterplots in Appendix k, we can see that there is a general negative correlation between the length of prompt and the performance of the model. We can also plot the number of models that perform the best for each prompt type as follows:

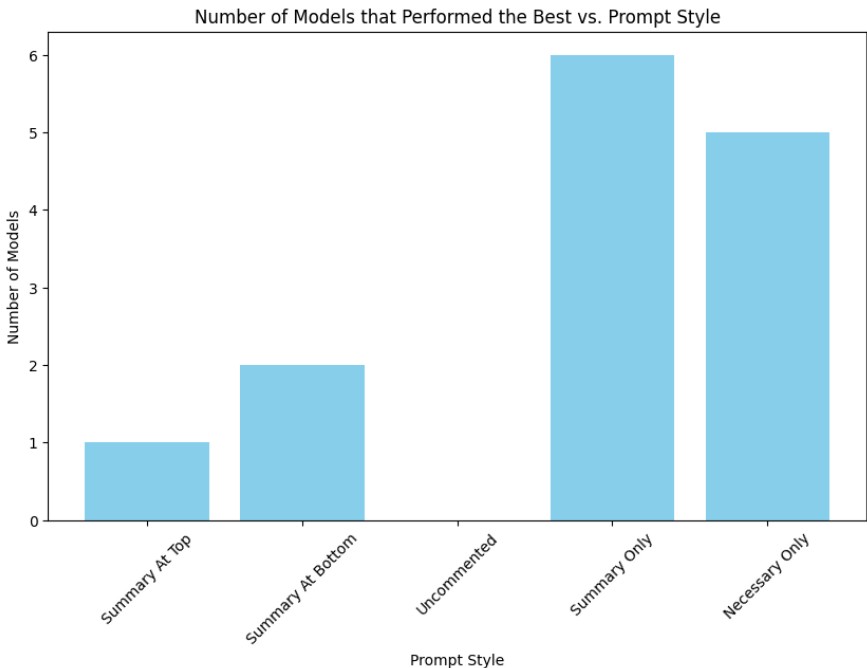

Figure 19: Number of best performing models over all prompt styles

Judging from Figure 19, we can see that overall, the prompts with the fewest tokens, i.e. the "Summary Only" and "Necessary Only" prompt styles, had the most models that performed the best, however there are some models that performed better with the longer "Summary At Top" and "Summary At Bottom" prompts. In this section, we will analyze some of the possible reasons for these discrepancies and variations in the results by looking at prompt structure. Consider the following "Summary Only" and "Necessary Only" prompts for the following "binary_erosion" function:

```
def binary_erosion(x_data, structuring_element):
    is_strel_2d = structuring_element.ndim == 2
    is_img_2d = x_data.ndim == 2
    if is_strel_2d and not is_img_2d:
        y_data = numpy.zeros_like(x_data)
        for index, plane in enumerate(x_data):
            y_data[index] = skimage.morphology.binary_erosion(plane,
                structuring_element)
        return y_data
    if not is_strel_2d and is_img_2d:
        raise NotImplementedError(
            'A 3D structuring element cannot be applied to a 2D image.')
    y_data = skimage.morphology.binary_erosion(x_data,
    structuring_element)
    return y_data
```

Here is the "Summary Only" prompt:

```
#This is in python
#write a function called "binary_erosion" that takes in two parameters: "
    x_data" and "structuring_element". The function should first check if
     the "structuring_element" is a 2D array or not. If it is, and "
```

```
      x_data" is not a 2D array, the function should create a new array
      called "y_data" that is the same size as "x_data" and contains only
      zeros. The function should then loop through each slice of "x_data"
      and apply the skimage.morphology.binary_erosion function using the "
      structuring_element" as the structuring element. The result of each
      slice should be added to the corresponding slice of "y_data". Finally
      , the function should return "y_data". If the "structuring_element"
      is not 2D and "x_data" is 2D, the function should raise a
      NotImplementedError with the message "A 3D structuring element cannot
       be applied to a 2D image." Otherwise, the function should simply
      apply the skimage.morphology.binary_erosion function to "x_data"
      using "structuring_element" as the structuring element and return the
       result.
3 #
4 #def binary_erosion(x_data, structuring_element):
```

and here is the "Necessary Only" prompt

```
1 Write a function with the following specs:
2 --specs begin here--
3 #write a function called "binary_erosion" that takes in two parameters: "
      x_data" and "structuring_element". The function should first check if
       the "structuring_element" is a 2D array or not. If it is, and "
      x_data" is not a 2D array, the function should create a new array
      called "y_data" that is the same size as "x_data" and contains only
      zeros. The function should then loop through each slice of "x_data"
      and apply the skimage.morphology.binary_erosion function using the "
      structuring_element" as the structuring element. The result of each
      slice should be added to the corresponding slice of "y_data". Finally
      , the function should return "y_data". If the "structuring_element"
      is not 2D and "x_data" is 2D, the function should raise a
      NotImplementedError with the message "A 3D structuring element cannot
       be applied to a 2D image." Otherwise, the function should simply
      apply the skimage.morphology.binary_erosion function to "x_data"
      using "structuring_element" as the structuring element and return the
       result.
4 param: x_data (numpy.ndarray) - input data to be eroded.
5 param: structuring_element (numpy.ndarray) - structuring element for
      erosion.
6 return: y_data (numpy.ndarray) - eroded data.
7 --specs end here--
8 Note the function will be embedded in the following context
9 --context begins here--
10 import numpy
11 import skimage.morphology
12 import os
13 numpy.random.seed(<|int;range=0,100|>)
14 <<insert solution here>>
15 def main():
16     x_data = numpy.random.randint(2, size=(10, 10))
17     structuring_element = skimage.morphology.square(3)
18     print(binary_erosion(x_data, structuring_element))
19 if __name__ == "__main__":
20     main()
21 --context ends here--
22 Make sure to only generate the function and not any of the context. Make
      sure you are generating valid, runnable code. Begin your solution
      with:
23 def binary_erosion(x_data, structuring_element):
24 MAKE SURE TO INDENT THE BODY OF YOUR FUNCTION BY A TAB
```

From the structure of these prompts, we can hypothesize why some of the simpler models perform better with the "Summary Only" prompts, while the GPT models perform better generally with the "Necessary Only" prompts. Since the "Necessary Only" prompts includes the GPT-generated summary of the function embedded inside the prompt, while including additional context, it is

reasonable to hypothesize that the simpler models such as InCoder, SantaCoder, StarCoder, or CodeGen are simply unable to digest the context information embedded inside the "Necessary Only" prompts, and instead the context information ends up being regarded as noise that just confuses the simpler models resulting in poorer performance. However, the larger models, such as the GPT-3.5 and GPT-4 models released by OpenAI, are able to properly incorporate the context information provided in the "Necessary Only" prompts, and as a result, they perform significantly better than their smaller counterparts with the additional context information.

It is also worth looking at the structure of the "Uncommented" prompt style, to see why these prompts perform so poorly when compared to the commented version of the prompts. For instance here is the "Uncommented" promppt style for the "binary_erosion" problem.

```
1  This is in python
2  write a function called "binary_erosion" that takes in two parameters: "
       x_data" and "structuring_element". The function should first check if
        the "structuring_element" is a 2D array or not. If it is, and "
       x_data" is not a 2D array, the function should create a new array
       called "y_data" that is the same size as "x_data" and contains only
       zeros. The function should then loop through each slice of "x_data"
       and apply the skimage.morphology.binary_erosion function using the "
       structuring_element" as the structuring element. The result of each
       slice should be added to the corresponding slice of "y_data". Finally
       , the function should return "y_data". If the "structuring_element"
       is not 2D and "x_data" is 2D, the function should raise a
       NotImplementedError with the message "A 3D structuring element cannot
        be applied to a 2D image." Otherwise, the function should simply
       apply the skimage.morphology.binary_erosion function to "x_data"
       using "structuring_element" as the structuring element and return the
        result.
3
4  def binary_erosion(x_data, structuring_element):
5
6  Here are the imports:
7  import skimage.morphology
8  import scipy.ndimage
9  import numpy
10 Here are the global variables:
11 Here are the class declarations:
12 Here are the additional function declarations:
13 def dilation(x_data, structuring_element):
14     summary: Performs dilation on input image data with a structuring
       element
15     param: x_data (numpy array) - input image data to perform dilation on
16     param: structuring_element (numpy array) - structuring element for
       the dilation operation
17     return: y_data (numpy array) - resulting dilated image data
18 def erosion(x_data, structuring_element):
19     summary: Performs erosion operation on input data using a structuring
        element.
20     param: x_data (numpy.ndarray) - input data to apply erosion on.
21     param: structuring_element (numpy.ndarray) - structuring element used
        for erosion operation.
22     return: y_data (numpy.ndarray) - erosion result as a 2D or 3D numpy
       array.
23 def binary_erosion(x_data, structuring_element):
24     summary: Performs binary erosion on input data using a structuring
       element.
25     param: x_data (numpy.ndarray) - input data to be eroded.
26     param: structuring_element (numpy.ndarray) - structuring element for
       erosion.
27     return: y_data (numpy.ndarray) - eroded data.
28 def morphological_gradient(x_data, structuring_element):
29     summary: Computes the morphological gradient of an image.
30     param: x_data (numpy array) - input data.
```

```
31      param: structuring_element (numpy array) - structuring element used
         for the operation.
32      return: y_data (numpy array) - output data.
33 Here are the comments and the specs:
34 write a function called "binary_erosion" that takes in two parameters: "
         x_data" and "structuring_element". The function should first check if
          the "structuring_element" is a 2D array or not. If it is, and "
         x_data" is not a 2D array, the function should create a new array
         called "y_data" that is the same size as "x_data" and contains only
         zeros. The function should then loop through each slice of "x_data"
         and apply the skimage.morphology.binary_erosion function using the "
         structuring_element" as the structuring element. The result of each
         slice should be added to the corresponding slice of "y_data". Finally
         , the function should return "y_data". If the "structuring_element"
         is not 2D and "x_data" is 2D, the function should raise a
         NotImplementedError with the message "A 3D structuring element cannot
          be applied to a 2D image." Otherwise, the function should simply
         apply the skimage.morphology.binary_erosion function to "x_data"
         using "structuring_element" as the structuring element and return the
          result.
35 def binary_erosion(x_data, structuring_element):
```

and here is the corresponding commented version of the prompt:

```
1 #This is in python
2 #write a function called "binary_erosion" that takes in two parameters: "
         x_data" and "structuring_element". The function should first check if
          the "structuring_element" is a 2D array or not. If it is, and "
         x_data" is not a 2D array, the function should create a new array
         called "y_data" that is the same size as "x_data" and contains only
         zeros. The function should then loop through each slice of "x_data"
         and apply the skimage.morphology.binary_erosion function using the "
         structuring_element" as the structuring element. The result of each
         slice should be added to the corresponding slice of "y_data". Finally
         , the function should return "y_data". If the "structuring_element"
         is not 2D and "x_data" is 2D, the function should raise a
         NotImplementedError with the message "A 3D structuring element cannot
          be applied to a 2D image." Otherwise, the function should simply
         apply the skimage.morphology.binary_erosion function to "x_data"
         using "structuring_element" as the structuring element and return the
          result.
3 #
4 #def binary_erosion(x_data, structuring_element):
5 #
6 #Here are the imports:
7 #import skimage.morphology
8 #import scipy.ndimage
9 #import numpy
10 #Here are the global variables:
11 #Here are the class declarations:
12 #Here are the additional function declarations:
13 #def dilation(x_data, structuring_element):
14 #    summary: Performs dilation on input image data with a structuring
         element
15 #    param: x_data (numpy array) - input image data to perform dilation on
16 #    param: structuring_element (numpy array) - structuring element for
         the dilation operation
17 #    return: y_data (numpy array) - resulting dilated image data
18 #def erosion(x_data, structuring_element):
19 #    summary: Performs erosion operation on input data using a structuring
          element.
20 #    param: x_data (numpy.ndarray) - input data to apply erosion on.
21 #    param: structuring_element (numpy.ndarray) - structuring element used
          for erosion operation.
```

```
22  #    return: y_data (numpy.ndarray) - erosion result as a 2D or 3D numpy
        array.
23  #def binary_erosion(x_data, structuring_element):
24  #    summary: Performs binary erosion on input data using a structuring
        element.
25  #    param: x_data (numpy.ndarray) - input data to be eroded.
26  #    param: structuring_element (numpy.ndarray) - structuring element for
        erosion.
27  #    return: y_data (numpy.ndarray) - eroded data.
28  #def morphological_gradient(x_data, structuring_element):
29  #    summary: Computes the morphological gradient of an image.
30  #    param: x_data (numpy array) - input data.
31  #    param: structuring_element (numpy array) - structuring element used
        for the operation.
32  #    return: y_data (numpy array) - output data.
33  #Here are the comments and the specs:
34  #write a function called "binary_erosion" that takes in two parameters: "
        x_data" and "structuring_element". The function should first check if
         the "structuring_element" is a 2D array or not. If it is, and "
        x_data" is not a 2D array, the function should create a new array
        called "y_data" that is the same size as "x_data" and contains only
        zeros. The function should then loop through each slice of "x_data"
        and apply the skimage.morphology.binary_erosion function using the "
        structuring_element" as the structuring element. The result of each
        slice should be added to the corresponding slice of "y_data". Finally
        , the function should return "y_data". If the "structuring_element"
        is not 2D and "x_data" is 2D, the function should raise a
        NotImplementedError with the message "A 3D structuring element cannot
         be applied to a 2D image." Otherwise, the function should simply
        apply the skimage.morphology.binary_erosion function to "x_data"
        using "structuring_element" as the structuring element and return the
         result.
35  #def binary_erosion(x_data, structuring_element):
```

Comparing the two prompt styles, we can see the only difference is the inclusion of the "#" symbol in front of each line in the commented version of the line. These "#" symbols seem to have a large effect on LLM code generation. For instance, here is the a sample ChatGPT code generation after being passsed in a "Uncommented" prompt:

```
1   Here is the Python function as you described:
2
3   ‘‘‘Python
4   import numpy as np
5   from skimage.morphology import binary_erosion
6
7   def binary_erosion(x_data, structuring_element):
8
9       # Check if structuring_element is 2D
10      if len(structuring_element.shape) == 2:
11          # Check if x_data is not 2D
12          if len(x_data.shape) != 2:
13              # Create y_data (same shape as x_data, filled with zeros)
14              y_data = np.zeros_like(x_data)
15
16              # Apply binary_erosion slice by slice
17              for idx in range(x_data.shape[0]):
18                  y_data[idx] = binary_erosion(x_data[idx],
        structuring_element)
19
20              return y_data
21
22          else:
23              # Apply binary_erosion directly
24              return binary_erosion(x_data, structuring_element)
25
```

```
26      else:
27          # Check if x_data is 2D
28          if len(x_data.shape) == 2:
29              # Raise error
30              raise NotImplementedError("A 3D structuring element cannot be
    applied to a 2D image.")
31
32          else:
33              # Apply binary_erosion directly
34              return binary_erosion(x_data, structuring_element)
35  '''
36  If 2D structuring element is provided and x_data isn't 2D, function
        applies the skimage.morphology.binary_erosion function to each slice
        of x_data separately. Then adds result to y_data. Otherwise, if
        structuring_element isn't 2D and x_data is 2D, it raises an error. If
         both conditions don't apply, it directly applies the binary_erosion
        function to x_data.
```

and here is a sample ChatGPT code generation after being passed in a corresponding commented prompt:

```
1  import skimage.morphology
2  import numpy as np
3
4  def binary_erosion(x_data, structuring_element):
5      if len(structuring_element.shape) == 2:
6          if len(x_data.shape) != 2:
7              y_data = np.zeros_like(x_data)
8              for i in range(x_data.shape[0]):
9                  y_data[i] = skimage.morphology.binary_erosion(x_data[i],
    structuring_element)
10             return y_data
11
12     elif len(structuring_element.shape) != 2 and len(x_data.shape) == 2:
13             raise NotImplementedError("A 3D structuring element cannot be
    applied to a 2D image.")
14
15     else:
16         return skimage.morphology.binary_erosion(x_data,
    structuring_element)
```

We can see that the models respond far better to the commented prompt, only returning runnable Python code without wrapping it unnecessary text that may result in runtime errors in the testing environment. Overall, it is not just prompt length that has a tangible effect on LLM code generation performance. For instance, the slightly longer "Necessary Only" prompts can outperform the shorter "Summary Only" prompts for larger more complex models that can properly digest and understand the given code context. Furthermore, the inclusion of certain symbols, such as the presence of a "#" at the beginning of every line of the commented prompts, can help prompt models to generate more desirable code outputs.

