# OpenReview forum: "BioCoder: A Benchmark for Bioinformatics Code Generation with Contextual Pragmatic Knowledge"
_ICLR.cc/2024/Conference — Submitted to ICLR 2024_

### Official Review · Reviewer_Adqr · 2023-10-30

**Soundness:** 2 fair
**Presentation:** 1 poor
**Contribution:** 2 fair
**Rating:** 5
**Confidence:** 4

**Summary:**

## Summary
Authors construct a dataset for code generation targeting the bioinformatics domain.

They use a combination of automated fetching of code from GitHub combined with manual inspection of code, and creating test cases.

They ran existing popular LLM models on their dataset and reported how they perform.
This adds additional value to the paper, showcasing that all LLMs struggle on the task they created.

**Strengths:**

+ **Important problem**. I like the problem being tackled. There is a great challenge in enabling code generation for domain specific tasks.
+ **Combination of automated and manual inspection**. Manual inspection can ensure better quality.
+ Authors created a repository with code and thus are enabling **knowledge sharing**. I suppose they plan to open source it later as well.

**Weaknesses:**

- **Paper writing**. Can be significantly improved. Flow of the paper is inconsistent, sometimes containing phrases which are not previously explained.
- **Inconsistent results and motivation**. The results shown in the evaluation don't testify much about the usefulness of the constructed dataset for the challenges authors initially propose tackling.

**Questions:**

## Other Comments
- Abstract. I would expect that in the second part of the abstract (after you motivated the problem of difficulty of correctly generating code for domain specific areas such as bioinformatics) you provide some evidence on why your dataset is either: (a) useful to help train models that achieve better performance for the given domain or (b) it provides a good measure to estimate whether an LLM is effective for the code generation in the field of bioinformatics (or specific subarea of bioinformatics coding domains).

- Abstract. Sentence "In relation to function-code generation ..." is unclear given the previous sentences. Reformulate.

- Abstract. Sentence "It incorporates 1026 functions and 1243 methods in Python and Java from GitHub". You probably need to add "respectively" after the word "Java".

- Introduction. I would like to see some discussion or evidence of why the dataset you constructed is an effective dataset for the field of bioinformatics. Some of the questions that are still left in my mind are:
    - What is the range of problems/areas bioinformatics code typically encompasses?
    - What position/importance in the field of bioinformatics is occupied by the tasks you include in your dataset? I see some information in the appendix, but ultimately this is a crucial factor in the paper, so there needs to be a good evidence about it in the main paper.
    - Ultimately, what is the value that a bioinformatics programmer, who wants to use an LLM to help writing his/her code, will gain by evaluating LLMs on your dataset? Can he/she gain some level of confidence that LLM that works well on your dataset, will perform better on his/her problems?

- Introduction. You mention that you create "a new high-quality dataset". It may be a high-quality dataset, but it would be good to add some specific numbers or evidence on why it is so.

- It would be great if in Figure 2 (describing the overall process of constructing the dataset) you indicate which parts of the process are done manually, and which part is automated.

- Related Work. "We ensure each function demands a certain level of domain expertise in bioinformatics". Can you add briefly how? It can be as simple as "via manual inspection". But, later on you need to elaborate further how the manual inspection was performed to ensure that.

- "golden code file". While one can guess what the meaning of the golden code file is, it would be good to explain. This phrase is used only at a single place in the paper.

- "Summary Only prompts". When encountering this phrase in the paper for the first time, it's not obvious what this means. One can look at Figure 3 and infer, but it would be good to have a natural flow in the main body of the paper itself.
    - If your concern is space, I would rather remove results of certain prompt types from the main paper, and put them into the appendix. Then, for a single version of the prompt I would explain in detail in the main paper how it is constructed.
    - Also, Figure 3 is not elaborated inside of the paper.

- Table 4. "Results are in %". I would write something like: results are expressed in percentage.

- I would add line numbers for the review purposes so reviewers can easily refer to a given part of the paper in their comments.

- Table 1. Spell out the abbreviations somewhere for P.C., P.L., C.C., C.L.

- Table 6. I would suggest using percentage instead of raw numbers for showing distribution of test error types. You can also include information about total number of test cases vs number of failed cases. You could possibly restructure the table to remove repetitive "Failure Reason".

---

> ### Author Response · Authors · 2023-11-16
> **Response to Reviewer Adqr:**
>
> Thank you for your constructive review. We have addressed your comments and questions and made respective changes in our revised manuscript.

---

> ### Author Response · Authors · 2023-11-16
> **Improvements to Abstract for Enhanced Clarity**
>
> > Abstract. I would expect that in the second part of the abstract (after you motivated the problem of difficulty of correctly generating code for domain specific areas such as bioinformatics) you provide some evidence on why your dataset is either: (a) useful to help train models that achieve better performance for the given domain or (b) it provides a good measure to estimate whether an LLM is effective for the code generation in the field of bioinformatics (or specific subarea of bioinformatics coding domains).
>
> > Abstract. Sentence "In relation to function-code generation ..." is unclear given the previous sentences. Reformulate.
>
> > Abstract. Sentence "It incorporates 1026 functions and 1243 methods in Python and Java from GitHub". You probably need to add "respectively" after the word "Java".
>
> Thank you for all your good suggestions on our abstract. We found it really useful, and we have edited our abstract based on them.
>
> Current abstract:
>
> >> Pre-trained large language models have significantly improved code generation. As these models scale up, there is an increasing need for the output to handle more intricate tasks and to be appropriately specialized to particular domains. Bioinformatics provides an important domain. In this field generating functional programs poses additional notable challenges due to the amount of specialized domain knowledge, the need for complicated data operations, and intricate functional dependencies between the operations. Here, we present BioCoder, a benchmark developed to evaluate existing pre-trained models in generating bioinformatics code. In relation to function-code generation, BioCoder covers potential package dependencies, class declarations, and global variables. It incorporates 1026 functions and 1243 methods in Python and Java from GitHub and 253 examples from the Rosalind Project. BioCoder incorporates a fuzz-testing framework for evaluation, and we have applied it to evaluate many models including InCoder, CodeGen, CodeGen2, SantaCoder, StarCoder, StarCoder+, InstructCodeT5+, GPT-3.5, and GPT-4. The results highlight two key aspects of successful models: 1) that they contain specific domain knowledge of bioinformatics (beyond just coding knowledge); 2) that they accommodate a long prompt with full context (i.e. functional dependencies).
>
> Based on your comment, we could edit our abstract to:
>
> >> Pre-trained large language models have significantly improved code generation. As these models scale up, there is an increasing need for the output to handle more intricate tasks and to be appropriately specialized to particular domains. Here, we target the bioinformatics domain due to the amount of specialized domain knowledge, algorithms, and data operations. We present BioCoder, a benchmark developed to evaluate large language models (LLMs) in generating this bioinformatics-specific code. BioCoder spans a broad spectrum of the field and covers cross-file dependencies (e.g., package dependencies), class declarations, and global variables. It incorporates 1026 Python functions and 1243 Java methods extracted from GitHub, along with 253 examples from the Rosalind Project, all pertaining to bioinformatics. Using topic modeling we show that overall coverage of the included code is representative of the full spectrum of bioinformatics calculations. BioCoder incorporates a fuzz-testing framework for evaluation. We have applied it to evaluate many models including InCoder, CodeGen, CodeGen2, SantaCoder, StarCoder, StarCoder+, InstructCodeT5+, GPT-3.5, and GPT-4.  Furthermore, we finetuned StarCoder, demonstrating how our benchmark dataset can effectively enhance the performance of an LLM. The results highlight two key aspects of successful models: (1) that they contain specific domain knowledge of bioinformatics (beyond just coding knowledge); (2) that they accommodate a long prompt with full context (i.e. functional dependencies).

---

> > ### Author Response · Authors · 2023-11-16
> > **Enhancements to the Introduction**
> >
> > > Introduction. I would like to see some discussion or evidence of why the dataset you constructed is an effective dataset for the field of bioinformatics. Some of the questions that are still left in my mind are: What is the range of problems/areas bioinformatics code typically encompasses?
> >
> > Our benchmark data encapsulates the majority of daily tasks a bioinformatician encounters in data analysis.  In Appendix O, Using topic modeling we show that overall coverage of the included code is representative of the full spectrum of bioinformatics calculations.
> >
> > Bioinformatics code typically encompasses a wide array of problems targeted at understanding biological data. This includes genomic sequence analysis, predicting protein structure and function, analyzing gene expression, and protein-protein interactions, and developing algorithms and statistical models to interpret biological data, among others.
> >
> >
> > > What position/importance in the field of bioinformatics is occupied by the tasks you include in your dataset? I see some information in the appendix, but ultimately this is a crucial factor in the paper, so there needs to be a good evidence about it in the main paper.
> >
> > The tasks covered in the \textsc{BioCoder} dataset were carefully extracted from popular bioinformatics packages of published research literature. We strived to extract a representative sample from these repositories included in survey papers, therefore ensuring that we cover as much of the bioinformatics field as possible. While we intentionally do not focus on a specific topic, we have ensured functions from high-profile repositories, such as those that deal with DNA sequencing, protein structure analysis, and gene expression analysis, are included. Therefore, we ensure a broad range of topics, guaranteeing that the most important tasks in the field of bioinformatics are covered. This ensures that our benchmark reflects real-world scenarios across a broad spectrum of bioinformatics tasks.
> >
> > > Ultimately, what is the value that a bioinformatics programmer, who wants to use an LLM to help writing his/her code, will gain by evaluating LLMs on your dataset? Can he/she gain some level of confidence that LLM that works well on your dataset, will perform better on his/her problems?
> >
> > BioCoder acts as a gauge of the potential performance of LLMs in the bioinformatics domain. If an LLM performs well on this dataset, it indicates that it has a good grasp of bioinformatics programming tasks and, thus, is likely to perform well on similar real-world tasks. This is compounded by the fact that our dataset places a special emphasis on cross-file dependencies, reflecting the development experience in a real-world scenario. This can help bioinformatics programmers decide which models to use, and how to use them, saving time and resources.
> >
> > > Introduction. You mention that you create "a new high-quality dataset". It may be a high-quality dataset, but it would be good to add some specific numbers or evidence on why it is so.
> >
> > Thank you for your suggestion. While it is difficult to comment on the “quality” of the data with numbers and metrics, it is important to note the results in the preliminary human study, in which human annotators rated nearly all of the functions as high quality, and completable by a human. We promise to add finished results to the camera-ready version of the paper.
> > For now, we have updated the introduction in the manuscript accordingly.
> >
> > ORIGINAL:
> >
> > >> We create a new high-quality dataset for code generation, curated from 1,720 bioinformatics repositories referenced in peer-reviewed bioinformatics articles, aiding in the practical and realistic evaluation of code generation tasks.
> >
> > NEW:
> >
> > >> We create a new high-quality dataset for code generation, curated from 1,720 bioinformatics repositories referenced in peer-reviewed bioinformatics articles, aiding in the practical and realistic evaluation of code generation tasks.
> > To ensure we captured the highest quality functions from these repositories, we employed two rounds of filtering,  with both an automatic approach (using GPT) and a manual examination. In the first round, we used GPT to compare each candidate function to a list of bioinformatics keywords, and then of the functions that matched the closest to the list, we performed a manual inspection to ensure it meets our guidelines of being highly bioinformatics-related, the non-trivial function that relies on code context. For the second round, it was done with the expertise of four students specializing in computer science and computational biology.
> >
> > More information can be found in Section 3.1: DATASET FILTERING.

---

> > > ### Author Response · Authors · 2023-11-16
> > > **Refinements to Diagram Clarifications, Prompt Construction, and Table Presentation**
> > >
> > > > It would be great if in Figure 2 (describing the overall process of constructing the dataset) you indicate which parts of the process are done manually, and which part is automated.
> > >
> > > Thanks for pointing out potential ambiguities in our Figure 2 diagram. We have updated Figure 2 in the revision to clarify which parts of the process have been automated, and which parts were completed manually.
> > >
> > >
> > > > Related Work. "We ensure each function demands a certain level of domain expertise in bioinformatics". Can you add briefly how? It can be as simple as "via manual inspection". But, later on you need to elaborate further how the manual inspection was performed to ensure that.
> > >
> > >
> > > Thanks for your great suggestion. We have added a simple description to the paper and used an appendix to describe in detail how we performed manual inspection.
> > >
> > > In summary, we use a combination of automatic filtering, GPT-assisted filtering, and manual inspection on the finished dataset to ensure that the function demands a certain level of domain expertise.
> > >
> > > We utilize GPT to generate relevant keywords for each function, and compare them to a list of hundreds of keywords scraped from Wikipedia that are known to be relevant to the field of bioinformatics. Functions that do not satisfy this requirement likely are not closely related to bioinformatics, and are therefore scrapped. Out of the remaining functions, we relied on the expertise of four students specializing in the Computer Science and Computational Biology departments to perform a human inspection to ensure that they utilize a topic, technique, or algorithm specific to bioinformatics.
> > >
> > >
> > > > "golden code file". While one can guess what the meaning of the golden code file is, it would be good to explain. This phrase is used only at a single place in the paper.
> > >
> > > Thank you so much for the feedback. Sorry for the confusion, we should use “golden code” instead of “‘golden code file’”. Here “golden code” is the ground truth code that is guaranteed to be correct, producing the correct outputs when run with the test cases. We use the golden code outputs to compare with the generated outputs for Pass @K testing purposes. Golden code and golden code files are used interchangeably to reference the original code from the repository, as opposed to the generated code from LLMs. We have updated the revision to reflect this.
> > >
> > > > "Summary Only prompts". When encountering this phrase in the paper for the first time, it's not obvious what this means. One can look at Figure 3 and infer, but it would be good to have a natural flow in the main body of the paper itself.
> > > Thanks for your suggestion and we have updated Section 5.
> > > Summary-only prompts are shortened prompts utilized across all our LLM models to ensure that context-limited LLMs still receive all the information necessary to potentially generate functions. Within the summary-only prompts, we optimized our prompts to only contain the absolute minimum of necessary information, without including much of the additional context that provides detail regarding the functionality of other dependencies.
> > >
> > > > If your concern is space, I would rather remove results of certain prompt types from the main paper, and put them into the appendix. Then, for a single version of the prompt I would explain in detail in the main paper how it is constructed.
> > >
> > > Thanks for your suggestions. We tried moving a subset of the results of prompt types to the Appendix, however since we repeatedly cite the results across various prompt types in the results analysis section, we could not find a way to make it coherent. For now, we keep the results of all prompt types for this revision of the paper, but we will look into this further.
> > >
> > > Furthermore, we have put a brief description of the prompt construction in Section 4. MODELS AND RESULTS.
> > >
> > > > Also, Figure 3 is not elaborated inside of the paper.
> > >
> > > Thanks for pointing it out. We edited the paper with new sections, and have now referenced Figure 3 in Section 3.2 and the second paragraph of Section 4. MODELS AND RESULTS.
> > >
> > >
> > > > Table 4. "Results are in %". I would write something like: results are expressed in percentage.
> > >
> > > Thanks for pointing it out. We have updated it in the main text.
> > >
> > > > I would add line numbers for the review purposes so reviewers can easily refer to a given part of the paper in their comments.
> > >
> > > Thanks for suggesting it. We are using the ICLR official template, without line numbers.
> > >
> > > > Table 1. Spell out the abbreviations somewhere for P.C., P.L., C.C., C.L.
> > >
> > > Thank you so much for the feedback. We have updated the manuscript so that it more clearly indicates the meaning of the “summary only” prompts, and have optimized it so that it is much easier to follow.

---

> > ### Comment · Reviewer_Adqr · 2023-11-21
> > **Re: Improvements to Abstract for Enhanced Clarity**
> >
> > Modified abstract looks better. Other suggestions for the abstract:
> >
> > - "we finetuned StarCoder, demonstrating how our benchmark dataset can effectively enhance the performance of an LLM"
> > If possible add some numbers showing how much of improvements it brought to StarCoder.
> >
> > - You can expand on the last sentence, as it is the one that testifies about the usefulness of the dataset.
> > "The results highlight two key aspects of successful models: 1) that they contain specific domain knowledge of bioinformatics (beyond just coding knowledge); 2) that they accommodate a long prompt with full context (i.e. functional dependencies)."
> >
> > e.g., how you tested that successful models contain specific domain knowledge of bioinformatics.
> > How does your dataset help there?
> > How does your dataset help in estimating quality of the model?
> > In the abstract it's good to have some quantitative evidence as well.
> >
> > minor things:
> > "this bioinformatics-specific code" -> I don't like using word "this". You can rephrase it.
> >
> > Other minor comments:
> > Table 1:
> > - "Test" column is not explained.
> > - "Num." in the caption. There's no need for "." after it, you probably want to bold it also, so it's clear it refers to the column name.

---

> ### Author Response · Authors · 2023-11-16
> **Showing Test Error Types as Percentages**
>
> > Table 6. I would suggest using percentage instead of raw numbers for showing distribution of test error types. You can also include information about total number of te st cases vs number of failed cases. You could possibly restructure the table to remove repetitive "Failure Reason".
>
> Thank you for your suggestion. We have updated table 6 to show the distribution of failure reasons in percentages, and merged some of the tables accordingly. They are reflected in the revision of the paper.
>
> | Type                  | Quantity | Percentage |
> |-----------------------|----------|------------|
> | Mismatched output     | 8661     | 4.567%     |
> | Invalid syntax        | 117665   | 62.038%    |
> | Runtime error         | 55351    | 29.184%    |
> | Time out              | 4        | 0.002%     |
> | Successfully Passed   | 7982     | 4.209%     |
>
>
>
> Once again, we **appreciate** your feedback and believe that our revisions have addressed your concerns. Your insights have helped improve the quality and clarity of our work.

---

> > ### Author Response · Authors · 2023-11-21
> > **Follow-Up: Seeking Further Feedback**
> >
> > Dear Reviewer,
> > We hope you're doing well. Following up on our recent exchange regarding this paper, we wanted to check if there are any further concerns or feedback from your side. Your insights are invaluable to us, and we're keen to address any remaining issues.

---

> > ### Comment · Reviewer_Adqr · 2023-11-21
> > **Re: Showing Test Error Types as Percentages**
> >
> > Table 5:
> > Instead of "we have omitted the percentage symbol (%)", you can write something like: "Numbers represent percentages".
> > You can consider if you want to round numbers to 2 decimals and include percentage symbol (I suppose spacing is a problem). But, it is up to you.
> >
> > Table 6:
> > It's a minor thing, but you can consider rounding to 2 decimal places.
> >
> > When looking up again which model Table 6 is related to, I backtrack to the text, paragraph: "Table 6 provides an overview of the error statistics collected from our test runs". It is unclear which model is in question?
> >
> > I do like when Table/Figure captions are self explanatory. But, considering the space issues you might not have enough space to achieve that. In that case, I would suggest that the place where you refer to a Table/Figure is clear about what exactly it is about.

---

> ### Author Response · Authors · 2023-11-22
> **Responses and Revisions to the Second Round of Feedback**
>
> Thank you for your further comments on our manuscript. **Your feedback on the abstract is much appreciated**, and we agree that a more quantitative approach to highlighting the improvements of StarCoder would be beneficial. Therefore, we have now updated this to provide specific numbers that show the notable performance increase that was achieved following fine-tuning on our benchmark dataset.
>
> Our new abstract is as follows:
>
> >> Pre-trained large language models have significantly improved code generation. As these models scale up, there is an increasing need for the output to handle more intricate tasks and to be appropriately specialized to particular domains. Here, we target bioinformatics due to the amount of specialized domain knowledge, algorithms, and data operations this discipline requires. We present BIOCODER, a benchmark developed to evaluate large language models (LLMs) in generating bioinformatics-specific code. BIOCODER spans a broad spectrum of the field and covers cross-file dependencies, class declarations, and global variables. It incorporates 1026 Python functions and 1243 Java methods extracted from GitHub, along with 253 examples from the Rosalind Project, all pertaining to bioinformatics. Using topic modeling we show that overall coverage of the included code is representative of the full spectrum of bioinformatics calculations. BIOCODER incorporates a fuzz-testing framework for evaluation. We have applied it to evaluate many models including InCoder, CodeGen, CodeGen2, SantaCoder, StarCoder, StarCoder+, InstructCodeT5+, GPT-3.5, and GPT-4. **Furthermore, we finetuned StarCoder, demonstrating how our dataset can effectively enhance the performance of an LLM (by >15% in terms of Pass@K in certain contexts and always >3%)**. The results highlight two key aspects of successful models: **(1) Successful models accommodate a long prompt (> ~2600 tokens) with full context, for functional dependencies. (2) They contain specific domain knowledge of bioinformatics, beyond just general coding knowledge. This is evident from the performance gain of GPT-3.5/4 compared to the smaller models on the benchmark (50% vs up to ~25%).**
>
>
> In consideration of your minor comments, we have taken the liberty to make **the following revisions**:
>
> - The term "this bioinformatics-specific code" has now been replaced with "bioinformatics-specific code".
>
> - For **Table 1**, we have expanded the explanation for the "Test" column.
>
> - The caption under "Num." has been boldly made and the period following it has been removed, as suggested.
> For **Table 5**, instead of "we have omitted the percentage symbol (%)", it has now been rephrased to "Numbers represent the Pass@K in the form of percentages.”
>
> - Regarding **Table 6**, we have clarified that the statistics are aggregated across all models.
>
> - Alleviated space constraints by converting **Table 5** into a two-column format.
>
> While we were considering rounding decimals to two decimal places, it seems that it does not make a meaningful difference on the length of the manuscript. Therefore, we have kept the rounding to three decimal places, as we would like to maintain as much detail about the results as possible given the performance of many models is very low.
>
> Your insights have been invaluable in refining our paper. We appreciate the constructive critique.

---

> > ### Author Response · Authors · 2023-11-22
> > **Follow-Up: Seeking Further Feedback**
> >
> > Dear Reviewer Adqr,
> >
> > We would like to express our gratitude for your in-depth feedback which has been incredibly valuable for our revisions. Following your suggestions, we have made comprehensive modifications addressing all your query points.
> >
> > As we are approaching the end of the discussion period, we kindly request your further remarks or concerns, should you have any. Your additional insights would be greatly beneficial for us to further improve the quality and clarity of our paper.
> >
> > Thank you again for your time and support.

---

> > > ### Comment · Reviewer_Adqr · 2023-11-23
> > >
> > > Thanks for making the updates.
> > > I think this is as good as the paper can get for this submission.
> > >
> > > > demonstrating how our dataset can effectively enhance the performance of LLMs (by >15% in terms of Pass@K in certain contexts and always >3%).
> > > >> Is the performance increase visible on the (test) portion of your own dataset, or in some other contexts?
> > >
> > > I do believe that the paper has potential but requires a more significant rewrite to make it flow nicely.
> > > There are similar (small) comments like the ones I had previously throughout the paper; which in combination would make a big difference in how the paper reads.

---

> ### Author Response · Authors · 2023-11-23
> **Re: Official Comment by Reviewer Adqr**
>
> Dear Reviewer Adqr,
>
> We sincerely thank you for recognizing the potential of our work and for your positive remarks on the amendments made to the current submission.
>
> Thank you for your patience and continued engagement. We value our commitment to clarifying every ambiguity.
>
> Definitely, the improvements we observed were indeed on **the test portion** of our BioCoder dataset after fine-tuning StarCoder using the training set.
>
> When we referenced the performance enhancement (by >15% in terms of Pass@K in certain contexts and always >3%), the **'certain contexts'** indeed were associated with **different Prompt Configurations** in our BioCoder test set. To be clearer, we observed improvements of over 15% in **some Prompt Configurations** (```Uncommented``` & ```Summary Only``` & ```Necessary Only```) following the fine-tuning of StarCoder on our benchmark dataset. However, **all Prompt Configurations** revealed enhancements with a persistently significant increase of over 3%. This specifically highlights how effectively our BioCoder dataset aids in augmenting the performance of LLMs across variable prompt configurations.
>
> To clarify the data presentation from Table 4, we've further broken down the information into two subtables for Python and Java, respectively.
>
> ```Here are the results for Python:```
>
>
> |**StarCoder-15.5B (w/o finetuning)** | Pass@1 | Pass@5 | Pass@10 | Pass@20 |
> |---------|--------|--------|---------|---------|
> | _Summary at Top_ | 3.694 | 13.197 | 19.359 | 24.554 |
> | _Uncommented_ | 0.318 | 1.062 | 1.591 | 2.548 |
> | _Summary Only_ | 4.682 | 15.225 | 21.200 | 27.166 |
> | _Necessary Only_ | 0.127 | 0.603 | 1.123 | 1.911 |
>
> |  **StarCoder-15.5B (finetuned)**     | Pass@1 | Pass@5 | Pass@10 | Pass@20 |
> |---------|--------|--------|---------|---------|
> | _Summary at Top_ | 5.818 | 16.562 | 21.091 | 27.048 |
> |  _Uncommented_ | 3.312 | 9.073 | 12.574 | 17.536 |
> |  _Summary Only_ | 7.295 | 20.838 | 26.143 | 39.570 |
> |  _Necessary Only_ | 0.597 | 1.173 | 1.813 | 2.611 |
>
> ```Here are the results for Java:```
>
> |     **StarCoder-15.5B  (w/o finetuning)**     | Pass@1 | Pass@5 | Pass@10 | Pass@20 |
> |---------|--------|--------|---------|---------|
> |  _Summary at Top_ | 0 | 0 | 0 | 0 |
> |  _Uncommented_ | 0 | 0 | 0 | 0 |
> |  _Summary Only_ | 0 | 0 | 0 | 0 |
> |  _Necessary Only_ | 0 | 0 | 0 | 0 |
>
> |     **StarCoder-15.5B (finetuned)**    | Pass@1 | Pass@5 | Pass@10 | Pass@20 |
> |---------|--------|--------|---------|---------|
> | _Summary at Top_ | 0 | 0 | 0 | 0 |
> | _Uncommented_ | 0 | 0 | 0 | 0 |
> |  _Summary Only_ | 0.200 | 1.000 | 2.000 | 4.000 |
> |  _Necessary Only_ | 3.300 | 12.097 | 19.545 | 30.000 |
>
>
>
>
> We acknowledge that the term 'in certain contexts' could have been more explicit. We will modify it to 'in certain prompt configurations' in our subsequent communications and manuscript revisions to ensure better understanding. We trust this further elucidates your query.
>
>
> Current version:
>
> >> Furthermore, we finetuned StarCoder, demonstrating how our dataset can effectively enhance the performance of LLMs on our benchmark (by >15% in terms of Pass@K in certain prompt configurations and always >3%).
>
> What's noteworthy is that your invaluable suggestions are primarily oriented towards the presentation and writing aspects of the manuscript, rather than questioning the foundational contents or the contributions proposed by our work. For future versions, we strive to enhance the readability while remaining assured that the essence of its contributions to the field is aptly recognized.
>
> Once again, we thank you for your time and constructive feedback. It has been instrumental in refining our paper, and we will continue to polish it for the next version.
>
> Best regards,
>
> Authors

---

> ### Author Response · Authors · 2023-11-23
> **Request for reassessment**
>
> Dear Reviewer Adqr,
>
> We are deeply appreciative of your comprehensive feedback and the time you have taken to improve our manuscript. We have sincerely endeavored to incorporate your invaluable suggestions, primarily those concerned with enhancing the readability and presentation aspects of the paper.
>
> We understand that although we have made significant improvements, the paper could still benefit from further polishing. However, given our earnest efforts in addressing the writing concerns in the limited time of the rebuttal period, and the absence of substantial disputes regarding the foundational content and contributions of our work, we genuinely hope that these critical developments in our revision will reflect positively in your reassessment of our work.
>
> We are more than happy to address followup comments you may have, and we will continue refining the paper for future versions, striving for clarity and improved readability.
>
> Best regards,
>
> Authors

---

### Official Review · Reviewer_BxXn · 2023-10-30

**Soundness:** 3 good
**Presentation:** 3 good
**Contribution:** 3 good
**Rating:** 6
**Confidence:** 2

**Summary:**

This paper addresses the function-level code generation challenges within the field of bioinformatics and evaluate the effectiveness of current leading pre-trained  large models (including InCoder, CodeGen, CodeGen2, SantaCoder, StarCoder, StarCoder+, InstructCodeT5+, GPT-3.5, and GPT-4) in generating code in the realm of bioinformatics.  To accomplish this, the paper utilized web scraping techniques to extract data from 1743 bioinformatics-adjacent GitHub repositories on GitHub, constructing and presenting a benchmark dataset named BioCoder. This dataset offers a reliable evaluation standard for code generation tasks within the field of bioinformatics. The main contribution of this paper comprises code parsing tools, proposed dataset, docker environment and comprehensive evaluations.

**Strengths:**

The dataset, evaluation in this paper is particulary well-established. The implementation is very solid, and the presentation in this paper is easy to follow. It has the poteintial to become a standard evaluation benchmark for bioinformatic code generation.

**Weaknesses:**

The article includes an appendix that might be too long for the readers, and the content in the appendix is referred from the main body for multiple times. It is better to make the paper more self-contained if it is accepted to be published in conference proceedings. Moreover, the author should clearly point out the main technical contribution of this paper. I don't quite catch the challenges for benchmarking bioinformatics code generation compared to other domain specific languages.

**Questions:**

1. How does the benchmarking for bioinformatics code generation differs from the benchmarking for other domain specific languages?
2. What is the key technical contribution for the BioCoder, that is strongly related to benchmarking bioinformatics code generation?

---

> ### Author Response · Authors · 2023-11-16
> **Our response to Reviewer BxXn**
>
> Thank you for your insightful questions!
>
> > It is better to make the paper more self-contained if it is accepted to be published in conference proceedings.
>
> Thank you for your constructive feedback. We agree that the paper could benefit from being more self-contained. As per your suggestion, we have revisited the main text and revised it in a way so as to include more key details that were initially placed in the appendix, for instance, the definition of different versions of Prompts. We are confident, after these revisions, that the paper stands comprehensive on its own. Even in the absence of the appendix, the purpose, methodologies, and key findings of our research can be entirely understood from the main text. We appreciate your insights and believe these changes significantly enhance the paper's readability and comprehensibility.
>
>
> >  I don't quite catch the challenges for benchmarking bioinformatics code generation compared to other domain specific languages.
>
> The challenges of benchmarking bioinformatics code generation originate from both the characteristics of the domain and the nature of the tasks involved.
>
> Good bioinformatics coding requires a deep understanding of both biological concepts and advanced computation methods. Generating or evaluating code thus requires knowledge and experience in both. Bioinformatics involves numerous subfields and types of tasks, from sequence alignment, and phylogenetic analysis, to protein structure prediction, to name just a few.
>
> Hence, to generate high-quality and functional bioinformatics code, the model must be well-versed not only in programming but also in biological and computational concepts relevant to bioinformatics.
>
>
> > How does the benchmarking for bioinformatics code generation differs from the benchmarking for other domain specific languages?
>
> While the concept of Bioinformatics code does not differ significantly from domains such as chemistry, it differs a bit from more mathematical and computational heavy domains like engineering, physics, and others. In our findings, the code is a lot more dependent on knowledge within the domain, and it is nearly impossible to generate any of the functions without the required domain knowledge.
>
> > Moreover, the author should clearly point out the main technical contribution of this paper.
>
> > What is the key technical contribution for the BioCoder, that is strongly related to benchmarking bioinformatics code generation?
>
> Thank you for both questions. We will answer them at the same time.
>
> The technical contributions of this paper are multifold, primarily focusing on the development of tools, datasets, models, and testing environments to enhance bioinformatics code generation:
>
> Code Parsing Tools – We've developed comprehensive tools for parsing large-scale code repositories, which include AST (Abstract Syntax Tree) parsers and utilities to extract code attributes. These tools facilitate efficient and systematic extraction, transformation, and storage of coding tasks from raw source codes. Furthermore, they also feature techniques to correct minor errors that arise during generation.
>
> Data Processing Scripts - We have developed methodical procedures and scripts for data processing, which are both robust and transparent. These are provided so that others within the community may utilize and adapt them for their own tasks, lending methodological insights and fostering further research within the field.
>
> Testing and Docker – We've established a robust testing environment that includes Docker configurations, required dependencies, and an array of fuzz test cases. The aim is to create a reproducible testing setup that not only imitates realistic project scenarios but also ensures the scalability and transferability of our approach.
>
> Model Zoo - In our effort to promote applicability and ease-of-use, we have created a 'Model Zoo.' This feature provides immediate access to numerous pre-implemented code-generation large language models, such as SantaCoder, StarCoder, and CodeGen alongside APIs based on OpenAI. This enables other researchers or developers to effortlessly test their tasks on many different models, without the overhead of implementing these models from scratch. We firmly believe that our model hub significantly contributes engineering-wise to the entire domain.

---

> > ### Comment · Reviewer_BxXn · 2023-11-23
> >
> > Thanks for the response on the questions and updates on the submission. I believe the contribution of the paper and the implementation of the benchmark would have enough potiential influence. Moreover, I do appreciate the work on code parsing, data prrcessing, docker environment and model zoo.
> >
> > In addition, concerning the biology domain knowledge, the authors name several tasks like sequence alignment, phylogenetic analysis and protein strucutre prediction, and claim that the domain knowledge alters the benchmark (at least from the dataset selection or prompt generation). It is still not enough for the users of the benchmark to analyze a model on its understanding of domian specific knowledge. For instance, is it possible for BioCoder to make a distinction on the ability of domain specific knowledge and general code gerneation capability when contribution to a task of generating a specific bioinformatical program？
> >
> > If the answer is possitive, it would be more convincible on the technical novelty of the paper.

---

> ### Author Response · Authors · 2023-11-23
> **Re: Official Comment by Reviewer BxXn**
>
> Dear Reviewer BxXn,
>
> Thank you for your further inquiries and valuable insights.
>
> Indeed, BioCoder tasks inherently interweave domain-specific content with general coding competencies. While these strands are tangled, our BioCoder benchmark primarily emphasizes the importance of domain knowledge by manually selecting data with algorithms specifically used within the bioinformatics domain. Hence, purely generic code samples are not available in our dataset.
>
> We concur with your sentiment that disentangling domain-specific knowledge from general code generation skills is a challenging undertaking. In our BioCoder context, learning to generate bioinformatical programs necessitates an understanding of the underlying coding logic, syntax, and structure, which forms the bedrock of these tasks. Yet, the distinctive layer of biological context in our use cases shapes the models' nuanced understanding of specific domain knowledge.
>
> Significantly, our fine-tuning of StarCoder, which originally was pre-trained purely on coding data and thus devoid of any bioinformatics domain knowledge, demonstrated an essential improvement (>15% in terms of Pass@K). This significant enhancement affirms that exposure to our training data, which possesses a proportion of tasks necessitating biological understanding, successfully imparts some degree of domain knowledge to the model.
>
> We acknowledge the opportunity and necessity for future research to delve deeper into distinguishing and evaluating a model's domain-specific knowledge and its general code generation capability. Understanding these clearly could significantly further the interpretability and analysis of language models across various domains.
>
> We hope we have addressed your queries satisfactorily. We extend our heartfelt gratitude for your invaluable feedback that has driven us to look beyond and identify potential areas of investigation.
>
> Best regards,
>
> Authors

---

### Official Review · Reviewer_r6TX · 2023-11-01

**Soundness:** 3 good
**Presentation:** 3 good
**Contribution:** 3 good
**Rating:** 6
**Confidence:** 4

**Summary:**

This paper presents a large-scale benchmark, BioCoder, which is devoted to assessing the capability of LLMs regarding code generation, specifically in the field of bioinformatics. The authors collect 1026 functions and 1243 methods in two programming languages (Python and Java) from GitHub and 253 examples from the Rosalind project to form a relatively intricate dataset to evaluate the LLMs' code generation abilities from various aspects. The authors conduct multiple steps to ensure the validity and unbiasedness of the constructed benchmark. In particular, 10 different LLMs (including the fine-tuned one) are evaluated on BioCoder, and the experiments reveal what factors can potentially affect the performance of LLMs while tackling challenging code generation tasks.

**Strengths:**

1. Timely and vital problem.
2. A valuable large-scale benchmark for code generation, including specialized domain knowledge.
3. Comprehensive evaluation with multiple LLMs.
4. The presentation is in a good manner, and the paper is easy to follow.

**Weaknesses:**

I appreciate that this paper has provided a valuable benchmark to the communities, as the new benchmark can potentially help researchers and practitioners in this direction. However, there are several concerns regarding the methodology and evaluation of this paper, which I will elaborate on below:


1. My biggest concern is the lack of justifications for the testing framework for Python and Java functions in BioCoder. The authors state, "... we employ a custom syntax to indicate the insertion points for custom randomly generated test cases." How are the test cases "randomly generated" and inserted into the context files? I did not find any detailed explanations in the main paper or in Appendix L and Y. In particular, Appendix L only briefly introduces the pipeline of the testing framework, how does such an approach deliver "a secure and efficient testing framework, promising robustness in the evaluation of generated code"? The authors need to clarify more about the generation of the test cases.

2. In addition to the previous point, for Rosalind functions, the authors mentioned, "...the output of this execution is compared with the cached golden code output." Why and how are the generated codes compared with the gold code outputs? I do not find any experiment results that illustrate the comparison outcomes.

3. Another concern is the implementation of correction mechanisms which rectify minor syntax and style errors. What kind of syntax and style errors can be considered "minor" with no impact on the functionality of the generated programs? As the authors take invalid syntax and runtime error as two major failure reasons in the following error distribution analysis, I recommend further justifying such correction mechanisms, which may affect the validity of the analysis results.

4. Table 4 summarizes the performance of the studied LLMs on BioCoder w.r.t 4 different types of prompt formats. However, the explanations of the different prompt versions are placed in Appendix I, which makes Table 4 hard to understand. Moreover, Appendix I only gives explanations with examples of the prompts in each version; nevertheless, I am looking for some high-level guidelines for the prompt design. Namely, how the five prompt versions are proposed? Are they from existing lectures or experimental experience? What are the characteristics of different prompt formats?

5. The discussion of the experiment results seems shallow to me. In section 5, the authors consider there is an inverse relationship between the length of the input prompts and the performance of the generated codes. However, from Table 4 and Appendix I, the Necessary only prompts have relatively shorter prompts but lower passing rates compared to uncommented and Summary at Top/Bottom in most of the studied LLMs. The author may elaborate more on the perspectives of prompt structures and contents instead of just the length of the prompts.


Minor Comments

1. The "Summary At Bottom" results illustrated in Appendix U seem incomplete (no row for GPT-4).

2. From section 3.4, "Our testing framework starts with a manual review of selected functions, leading to the creation of a context file and a golden code file for each problem (see Figure 3)". I do not find how Figure 3 is correlated with the testing framework, Figure 17 in Appendix R may be a better example.

**Questions:**

1. The details of the testing framework and the corresponding effectiveness should be discussed.

2. For Rosalind functions, why and how are the generated codes compared with the gold code outputs?

3. What are the guidelines while designing the 5 different prompt styles for the subject LLMs?

---

> ### Author Response · Authors · 2023-11-16
> **Our response to Reviewer r6TX**
>
> Thank you for your comments. We have addressed each one as follows:
>
> > My biggest concern is the lack of justifications for the testing framework for Python and Java functions in BioCoder. The authors state, "... we employ a custom syntax to indicate the insertion points for custom randomly generated test cases." How are the test cases "randomly generated" and inserted into the context files? I did not find any detailed explanations in the main paper or in Appendix L and Y. In particular, Appendix L only briefly introduces the pipeline of the testing framework, how does such an approach deliver "a secure and efficient testing framework, promising robustness in the evaluation of generated code"? The authors need to clarify more about the generation of the test cases.
>
> Apologies for the oversight, as we did not provide enough information about the testing framework in Appendix L. We provided a brief overview of the pipeline in Appendix L, but we will make sure to explain the details and importance of the framework in more depth.
>
> Our fuzz testing framework was created based on the need to accurately compare ground truth and generated code samples, while optimizing the test case writing pipeline. A detailed explanation of the entire process, as well as our motivations, is written in the postceding paragraph. Our test cases can be thought of as typical unit tests with randomly generated parameters, hence why we are “inserting” randomly generated test cases into the code (more details given later). Our entire process, including the use of docker containers, virtualization, and consistent testing environments, allows us to ensure the security and robustness of our testing framework.
>
> Here is a more detailed explanation:
>
> We decided to utilize concepts from fuzz testing, as fuzz testing is widely used in the industry to capture bugs, crashes, security vulnerabilities, etc. in functions. However, in these cases, they do not have access to a “correct” version of the function; instead, they are merely creating inputs to intentionally try to crash the program, find out-of-bounds memory accesses, etc. Our situation is unique because we have the “golden code”, or the ground truth version of the code, so given input, we definitely know what the expected output should be, which is not something that’s usually available in typical fuzz testing frameworks. Therefore, our situation could be considered a mixture of both unit testing and fuzz testing.
>
> Given this requirement, and the goal of large-scale prompt generation, we decided to implement our own framework. We set out to accomplish two things: make the annotation process a lot easier for human editors, and support our feature set that combines both elements from unit testing and elements from fuzz testing. We believe that our resulting pipeline is more intuitive than piecing together other testing frameworks, and in our annotation process, it proved to make things efficient, enabling larger-scale annotation, as the goal of the paper.
>
> Furthermore, note that while handwritten test cases would likely target edge cases of a program (eg. branch coverage, conditional coverage), the probability of our fuzz testing framework hitting all of the same edge cases is high given 1000 iterations of randomly generated inputs. This means that we can save a significant amount of time building the dataset, as we only need to write an outline of a test case, and let the framework handle the rest. Therefore, we can think of the framework as thousands of “random unit tests,” with a high probability that these unit tests would include handwritten test cases, if we had written them.
>
> In terms of variable generation, we replace the <|var_type;parameter|> syntax with random values each iteration, for an unlimited number of iterations. These parameters are modifiable, and we implemented this system to be flexible, so that we can target specific scopes for fuzz testing. We check correctness by substituting the exact same variables in the original code, and checking if the outputs of the two functions match. This indicates identical functionality to the original code.

---

> ### Author Response · Authors · 2023-11-16
> **Examples of random integer and numpy array generation**
>
> Here is an example of random integer and numpy array generation:
>
> ```
> import numpy
> import skimage.morphology
> import os
> <<insert solution here>>
> def main():
>     numpy.random.seed(<|int;range=0,100|>)
>     labels = numpy.random.randint(2, size=(3, 3))
>     diameter = <|int;range=2,10|>
>     print(fill_object_holes(labels, diameter))
> if __name__ == "__main__":
>     main()
> ```
>
> Example of random string generation:
>
> ```
> import random
> [IMPORTS REDACTED FOR CONCISENESS]
> import warnings
> from textwrap import wrap
> import string
> import zlib
> import io
> from os.path import isfile
> class GenipeError(Exception):
>         pass
> _CHECK_STRING = b'GENIPE INDEX FILE'
> def dosage_from_probs(homo_probs, hetero_probs, scale=2):
>     """Computes dosage from probability matrix (for the minor allele).
>     Args:
>         homo_probs (numpy.array): the probabilities for the homozygous genotype
>         hetero_probs (numpy.array): the probabilities for the heterozygous
>                                     genotype
>         scale (int): the scale value
>     Returns:
>         numpy.array: the dosage computed from the probabilities
>     """
>     return (homo_probs + (hetero_probs / 2)) * scale
> <<insert solution here>>
> def main():
>     np.random.seed(<|int;range=0,100|>)
>     prob_matrix = np.random.rand(10, 10)
>     a1 = <|string|>
>     a2 = <|string|>
>     print(maf_dosage_from_probs(prob_matrix, a1, a2))
> if __name__ == "__main__":
>     main()
> ```
>
> Let’s break apart the integer syntax:
> <|int|> denotes an integer. If left without parameters, then in one iteration of the program, this will be replaced with a random integer between INT_MIN and INT_MAX before compile time (or in this case, before the Python file is executed). There are parameters that can be passed in, that include range, even/odd, etc.
> Similarly, for <|string|> this generates a random ASCII string of any type. It can be further narrowed down into ASCII strings only, lowercase only, specific characters only, etc. by passing in the relevant parameters.
> These random inserted values can be manipulated to become part of a larger data structure, for example, a Numpy array, or a mock Python object.
>
> When these files are executed, we replace <<insert solution here>> with the golden code on one iteration, and the error-corrected generated code on a corresponding iteration. The fuzzing framework is designed so that the same inputs will be passed to this pair of iterations, meaning that we should be getting the same output (none of the functions have a non-deterministic component to them). Therefore this supports one aspect of the “secure” testing framework, as we have created an environment where all else is equal, except for the generated/golden code.
>
> Furthermore, the “security” is also established by our usage of Docker containers to enable a secure (malicious code cannot affect the host system), consistent testing environment for all testing iterations. All of these aspects come together to enable us to efficiently and robustly test the generated functions.
>
> We have updated this in the Appendix as well.

---

> ### Author Response · Authors · 2023-11-16
> **Clarification on Code Comparison and Testing Process in Rosalind Functions**
>
> > In addition to the previous point, for Rosalind functions,
> the authors mentioned, "...the output of this execution is compared with the cached golden code output."
> Why and how are the generated codes compared with the gold code outputs? I do not find any experiment results that illustrate the comparison outcomes.
>
>
> Thank you for the clarifying question. In the current paper, we have: “For Rosalind functions, the process is simpler and more efficient as the functions are less complex. The golden code’s output is generated and cached ahead of time. During testing, the tester executes the generated code within the corresponding context, and the output of this execution is compared with the cached golden code output.”
>
> We admit that this is not clear. The “cached golden code output” refers to an optimization specific to Rosalind. We utilize the fact that we are given the same input for all test cases, and only run the golden code once, instead of running it 20 times in all 20 pairs of generated-golden code pairs. For the next revision, we will likely remove this in the name of clarity, and because it is an implementation detail that does not affect any results.
>
> Rosalind (https://rosalind.info/) is an educational resource and web project for learning bioinformatics through problem-solving and computer programming.
>
> In Rosalind Online Judge, they declare a code sample as “correct” using just one test case. We define “golden code” as the original code, which is guaranteed to be correct. Therefore, for our evaluation, we also use one test case, and use it as input for both the original golden code and the generated code. Then, the (typically string) outputs are compared, and declared “correct” if and only if it is an exact match.
>
> In the current paper, the “cached golden code output” refers to the optimization where we used the same test case on each pair of golden and generated code. We used a web scraper to automatically download the desired “test cases” from the Rosalind website. Therefore, the correct output should always be the same between all executions of a problem, so we only need to execute the golden code once, and “cache” the output.
>
> Based on this we can revise the paper to better explain the Rosalind testing process, and the similarities to the other testing process:
>
> “For Rosalind functions, the process is simpler and more efficient as the functions are less complex. During testing, the tester executes the generated code and golden code within the corresponding context with the same input, and the output of the pair of executions is considered correct if and only if they are identical.”
>
> Golden Code:
> count the number of times that “A” appears in the string
>
> ```
> def count_A(input_str):
>     return input_str.count(“A”)
>     print(count)
> ```
>
> Generated code:
> count the number of times that “A” appears in the string
>
> ```
> def count_A(input_str):
>     count=0
>     for character in input_str:
>         if character==’A’: count+=1
>     print(count)
> ```
>
> In these simplified examples, we could pass in the string “ABCAABC@” into the function as input_str. Note that both functions would correctly print “3” as the response, so we can mark it as correct. Here is another example of a generated sample:
>
> count the number of times that “A” appears in the string
>
> ```
> def count_A(input_str):
>     count=0
>     for character in input_str:
>         if “A” in input_str: count+=1
>     print(count)
> ```
>
> In this example, if we pass in the same string, then it would print “8”, which is clearly not the same as what the golden code printed, and therefore we would mark it as incorrect.

---

> > ### Author Response · Authors · 2023-11-16
> > **Justification for the Implementation of Correction Mechanisms in Error Analysis**
> >
> > > Another concern is the implementation of correction mechanisms which rectify minor syntax and style errors. What kind of syntax and style errors can be considered "minor" with no impact on the functionality of the generated programs? As the authors take invalid syntax and runtime error as two major failure reasons in the following error distribution analysis, I recommend further justifying such correction mechanisms, which may affect the validity of the analysis results.
> >
> > Thank you for the questions and suggestions. We admit that we did not describe the implementation details of this correction mechanism clearly.
> >
> > We define “minor syntax errors” as errors that we are able to correct automatically with our heuristics. For example, StarCoder often “over-generates” code, so during one generation (even with dynamic generation limits), it could generate 3-4 functions at once, while we would only need the first one. In other words, the model doesn’t know when to stop generating.
> >
> > In this case, we perform dynamic code analysis to potentially determine any relationships between functions, and retain the functions that are utilized in the target function, and remove all extraneous data. In other cases, we automatically correct mismatched brackets, missing semicolons, incorrect indentations, etc. so when inserted back into the context, it adapts accordingly.
> >
> > The logic behind this is that in real-world scenarios, if we are able to deterministically correct model outputs, then we should do so. We believe this method is justified as it doesn’t make sense to reduce resulting model performance by something that can be adjusted without human intervention.
> >
> > We do acknowledge that there could be more optimizations and heuristics we could implement to further enhance this error correction process, however, we did our best in generalizing the most common model output errors we could.

---

> ### Author Response · Authors · 2023-11-16
> **Clarification on Prompt Designs**
>
> > What are the guidelines while designing the 5 different prompt styles for the subject LLMs? (*)
>
> Thank you for your question. We would be happy to go into some more detail as to how we designed the 5 different prompt styles described in the paper.
>
> When designing the prompts, we wanted to make sure the prompts had sufficient and necessary information to capture the essence of the function that we were trying to generate. Specifically, we wanted to design prompts that contained enough code context so that an experienced programmer would have the necessary tools to complete the function.
> Using these guidelines, we tried hundreds of different ideas for prompt structures, manually testing the performance of various structures using the OpenAI API, as well as locally run models.
>
> Eventually, we ended up with 5 main types of prompts, each logically motivated by some observation we made during our experiments.
>
> We begin with the “summary at bottom” prompts, which is the format used by most papers (we based it off of CoderEval, but it seems to be used in many other papers as well). From here, we also follow the instruction model, putting the summary at the top.
> For the context, it’s important to note that models typically give higher “priority” to context later in the prompt. Therefore, the inter-file context needs to go before the intra-file context, which is more likely to be referenced by the target function. However, this context is not displayed in correct syntax, so it might have been important to make a distinction between the inter-file and intra-file context, so we commented out the inter-file ones.
> During testing, we observed that this would cause the model to generate an excessive amount of comments, sometimes generating the entire function as a comment. We couldn’t figure out a way to consistently handle this during the dynamic analysis/error correction process, so we uncommented these, and most models seemed to behave more or less “normally”
> The “Necessary context only” is human annotated context, which has been fully cleaned so that only the necessary information is included, so it is logical to test these. This is in contrast to the other prompt types, which are almost entirely automatically generated from a repository.
> For the “summary only” prompts, we purely wanted to see whether the function would be completable without any context whatsoever.
>
> We feel that these prompts represent a logical progression in our pursuit to extract the highest possible performance out of these models. While we do acknowledge that these may not be the perfect prompts for every model tested, our testing shows that they are some of the best out of the hundreds we have tried, and therefore should be sufficient to prompt LLMs for code generation.
>
>
> >  However, the explanations of the different prompt versions are placed in Appendix I, which makes Table 4 hard to understand
>
> We have introduced a paragraph in Section 4. MODELS AND RESULTS briefly discuss our prompt versions. We hope it provides an adequate amount of information to explain Table 4, while the more lengthy explanation is in Appendix I.
>
> > Moreover, Appendix I only gives explanations with examples of the prompts in each version; nevertheless, I am looking for some high-level guidelines for the prompt design. Namely, how the five prompt versions are proposed?
>
> Thank you for your question. Our answer to your question “What are the guidelines while designing the 5 different prompt styles for the subject LLMs?” includes an answer for this.
>
>
> > Are they from existing lectures or experimental experience?
>
> > What are the characteristics of different prompt formats?
>
> Thank you for both questions. We will answer them at the same time.
>
> We created the prompt formats based on experimental experience. We set out to create prompt structures to test the specific performance characteristics of each model. We first decided on a set of characters to target, as shown in the following prompt types:
> Uncommented/Summary at top/Summary at bottom - target performance on deciding which context to use, as there is intentionally some extraneous context not required for function
> Summary Only - gather metrics on the level of dependency on context
> Necessary Only - isolate purely the logical reasoning abilities of models, as we assume that all context is utilized.
> For each of the goals, we experimented with slight modifications of each prompt structure until we found one that achieved the highest performance, best representing the performance of that specific characteristic.

---

> > ### Author Response · Authors · 2023-11-16
> > **Comprehensive Analysis of Experiment Results**
> >
> > > The discussion of the experiment results seems shallow to me. In section 5, the authors consider there is an inverse relationship between the length of the input prompts and the performance of the generated codes. However, from Table 4 and Appendix I, the Necessary only prompts have relatively shorter prompts but lower passing rates compared to uncommented and Summary at Top/Bottom in most of the studied LLMs. The author may elaborate more on the perspectives of prompt structures and contents instead of just the length of the prompts.
> >
> > Thank you for your suggestion. In response, we have added a new appendix to the manuscript discussing more analysis of the performance with the perspective of prompt structures and contents. Please refer to the updated manuscript for details on the content of the appendix. We have also updated the “Analysis and Discussion” section in the manuscript. Initially, we wrote:
> >
> > >> As shown by the scatterplots in Appendix J, it seems that on models with an average Pass@K score of at least 2%, there is an inverse relationship between the number of tokens in the prompt and the Pass@K score. Furthermore, for models such as SantaCoder and GPT, the performance fell sharply after around 500 tokens. This could be due to the massive amount of context “confusing” the models. When we look at the Necessary Only prompts, a similar trend occurs for longer contexts, but the phenomenon is not as pronounced, as on average these prompts are significantly shorter. Further discussion on the results of each model can be found in Appendix I.
> >
> > We have now updated it to read as follows in the manuscript:
> >
> > >> As shown by the scatterplots in Appendix k, on models with an average Pass@K score of at least 2%, there is an inverse relationship between the number of tokens in the prompt and the Pass@K score. Furthermore, for models such as SantaCoder and GPT models, the performance fell sharply after around 500 tokens. Despite this, model performance can not only be attributed to prompt length.
> > We can see that even though the "Necessary Only prompts are relatively shorter when compared to the "Summary at Top" or "Uncommented" prompts, the Pass@k performance of the "Uncommented" prompts is in fact worse for many of the models. For more analysis regarding this and prompt structure in general, please refer to Appendix aa.

---

> ### Author Response · Authors · 2023-11-16
> **Inclusion of Detailed Analysis on Prompt Structures and Updates on GPT-4 Results**
>
> > Add more analysis and elaborate more on the perspectives of prompt structures and contents
>
> Thank you for your suggestion. In response, we have added a new appendix AA to the manuscript discussing more analysis on the performance with the perspective of prompt structures and contents. We have also updated the “Analysis and Discussion” section in the manuscript.
>
>
>
> > The "Summary At Bottom" results illustrated in Appendix U seem incomplete (no row for GPT-4).
>
> We have updated the results of GPT-4 in Appendix U.
>
> > “From section 3.4, "Our testing framework starts with a manual review of selected functions, leading to the creation of a context file and a golden code file for each problem (see Figure 3)". I do not find how Figure 3 is correlated with the testing framework, Figure 17 in Appendix R may be a better example.”
>
> Thank you for your feedback. We have corrected the reference in the revised paper.

---

> ### Author Response · Authors · 2023-11-22
> **Follow-Up: Seeking Further Feedback**
>
> Dear Reviewer, We hope you're doing well. Following up on our recent exchange regarding this paper, we wanted to check if there are any further concerns or feedback from your side. Your insights are invaluable to us, and we're keen to address any remaining issues.

---

### Official Review · Reviewer_p2gV · 2023-11-03

**Soundness:** 4 excellent
**Presentation:** 4 excellent
**Contribution:** 4 excellent
**Rating:** 8
**Confidence:** 4

**Summary:**

The authors have introduced a benchmark named BioCoder for code generation in bioinformatics. BioCoder covers codes in Python and Java, featuring examples from the Rosalind project. The benchmark creation process is detailed, encompassing preprocessing, evaluation metrics, and baselines. Additionally, several state-of-the-art models are evaluated on the benchmark, and their performance is reported highlighting the superiority of black-box models over open LLMs for code.

**Strengths:**

* The proposed benchmark is an essential collaboration in the field of Bioinformatics.
  * The processes for dataset creation, preprocessing, and other steps are very well described, including examples and explanations.
  * The authors test most of the state-of-the-art models for code generation on the proposed benchmark, reviewing each and also fine-tuning one of them.
  * Every prompt is exemplified in the Appendix with a code snippet.
  * Another interesting comparison is the one between BioCoder and CoderEval.
  * Every model is analyzed and discussed (very large Appendix).

**Weaknesses:**

* Explanations of the prompt configurations, shown in Table 4, should come in the Table description or somewhere in the main text, not only in the Appendix.
* It Would be interesting to have a human evaluation or experiment considering the descriptions as a way to bring more validation to the GPT3.5 creation.

**Questions:**

-

---

> ### Author Response · Authors · 2023-11-16
> **Our Response to Reviewer p2gV**
>
> Thank you for your insightful suggestions.
>
> > Explanations of the prompt configurations, shown in Table 4, should come in the Table description or somewhere in the main text, not only in the Appendix.
>
> We have modified the paper accordingly to better organize a brief description of the prompt configurations into the main sections of the paper, specifically, in the second paragraph of Section 4. MODELS AND RESULTS.
>
> > It would be interesting to have a human evaluation or experiment considering the descriptions as a way to bring more validation to the GPT3.5 creation.
>
> Furthermore, we have conducted a preliminary human evaluation study on the quality of the GPT-generated summaries. We have randomly sampled 50 summaries that we generated and have had 5 undergraduate students majoring in Computer Science classify each summary, and determine whether there are any errors in their description of the function.
>
> In this preliminary study, we have found that 48/50 summaries are specific enough to complete a function without much ambiguity in functionality. All summaries accurately reflect the intention of the function, tested by asking whether they could write the code themselves, after reading the summary. This demonstrates the accuracy of the summaries. We do acknowledge that our sample size of 50 summaries is somewhat small, but we wanted to provide a timely response, and we believe that our sample is representative of the larger dataset of summaries.

---

### Author Response · Authors · 2023-11-22
**General Responses to All Reviewers and ACs.**

Dear Reviewers and ACs,

We would like to **appreciate the time and effort that all the reviewers and ACs spent during reviewing**. Your constructive comments have significantly contributed to enhancing the quality of our submission.

Your positive recognition of the motivation and the problem being tackled by our benchmark is particularly encouraging  (all reviewers including p2gV, r6TX, BxXn, and Adqr). Reviewer p2gV highlighted the detailed benchmark creation process, while r6TX, Adqr, and BxXn praised the novel and essential contribution of our benchmark. In addition, they recognized our testing of state-of-the-art models on the proposed benchmark.

Based on the feedback, we have made the following major updates to the paper:

- We've revised the paper to enhance the organization and readability, and restructured the content for added clarity. Key information such as prompt configurations, testing strategies, fuzz testing framework, results, and explanations previously in the appendix has been expanded upon, and relocated to the main text (as suggested by p2gV, r6TX, Adqr).

- Considering some critiques on paper writing, we have edited multiple sections throughout the paper, focusing on the relevance and objects in the Abstract and Introduction, methodology in Sections 3 and 4, and expanded explanations in all sections of our research (as commented by Aqdr, p2gV)

- We have clarified and reformatted diagrams (Adqr) and tables throughout the paper (as commented by Adqr and r6TX)

- We have added more discussion and evidence in the introduction to explain the effectiveness of our dataset in the field of bioinformatics. We have provided information on the range of problems/areas typically encompassed by bioinformatics code and the importance of the tasks included in our dataset. We have also highlighted the value of such a benchmark in real-world applications (as suggested by Adqr, BxXn).

- Furthermore, we have clarified the meaning of the terms “golden code” and "golden code file" and explained their usage in the paper (r6TX, Adqr)

We hope the questions and concerns from all the reviewers can be addressed by adding more discussion, description, and evidence (Reviewer r6TX and Adqr), explanations of the prompt configurations (Reviewer p2gV, r6TX, Adqr), and revision on paper writing (Reviewer Adqr, p2gV, r6TX, and BxXn).

We also update our submission according to the suggestions from all the reviewers. Please contact us if you have further questions.

Best,
Authors

---

### Meta-Review · Area_Chair_Tqhq · 2023-12-19

**Metareview:**

This paper introduces a new dataset for bioinformatics code generation, which was generated by collecting and filtering bioinformatics GitHub repositories. The paper provides some basic statistics of the composition of the dataset and evaluates various code-generation models on it.

While most reviewers agree that benchmarks like this one are important for the community, there were several issued raised with respect to the presentation of the paper, and the motivation and generation of the dataset. While the former are important (especially for a dataset/benchmark paper), it is relatively salvageable. The latter, however, seem much more critical. Indeed, the paper lacks *detailed motivation* (what makes code for bioinformatics particularly difficult/different from “standard” code to justify the creation of a specialized benchmark) and *construct validity* (it is not clear whether this benchmark measures what the authors set out to measure: the capability of LLMs for code generation in bioinformatics). In particular:

* **The paper lacks extrinsic quality validation**. While the authors claim that this is a high-quality dataset, all evaluation is intrinsic (i.e. performance of models trained on the dataset itself), which makes it impossible to assess the actual quality of the dataset. It should be noted that most of the dataset papers cited by this work do have some form of extrinsic validation (e.g., DS-1000 [Lai et al. 2022; Section 2.5]). During the rebuttal, the authors conducted a “preliminary human evaluation” of very limited scale, which is nevertheless sparsely discussed in the rebuttal and not incorporated into the manuscript. Such a fundamental aspect of validation should have been included in the submitted version and cannot be appended as an afterthought.
* **Lack of evidence of generality**. The paper does not convincingly provide evidence to support its claim that the dataset provides a good measure of whether an LLM is effective at bioinformatics code generation *beyond this specific dataset* (a point raised by Adqr when discussing the abstract and introduction, but which the authors interpreted in a very narrow sense of “writing suggestion” rather than of a valid criticism of the paper’s justification).
* There are various **design choices that lack justification** or explanation: generation of test cases, syntax correction, prompt design. Again, the authors provide some justification for these in the response, but this discussion was added hastily in the manuscript, and several questions remain (as evidenced by the discussion with reviewer r6TX).
* **The analysis of model performance is shallow**. Besides generic and predictable observations (“larger models perform better”), there is little in the way of insights about what makes this dataset hard, failure modes of the tested models, and —most importantly— how much does training on this dataset improve the model (ideally, measure with respect to an extrinsic criterion) with respect to training on generic code. This, ultimately, is the reason why someone would train/evaluate their model on this benchmark, but the paper provides very little evidence in this regard.

Overall, I tend to agree with Reviewer Adqr’s comment that “this is as good as the paper can get for this submission”. While the authors have made several changes to the paper during the discussion, this phase is not intended for a complete rewriting of the paper. When fundamental issues in a paper need to be ‘patched’ during rebuttal, it indicates that the paper would probably benefit from another round of in-depth reviewing.

**Justification For Why Not Higher Score:**

A dataset/benchmark paper must convincingly: (i) justify why a certain type of dataset/benchmark is needed and (ii) demonstrate that this particular dataset (including its design choices, collection, filtering, etc) is representative of what is sought. This paper does not convincingly address either of these.

**Justification For Why Not Lower Score:**

N/A

---

### Decision · Program_Chairs · 2024-01-16

Reject